# Turning Internal Gap into Self-Improvement: Promoting the Generation-Understanding Unification in MLLMs

Yujin Han[1,8]*, Hao Chen[2], Andi Han[3,4], Zhiheng Wang[5,6], Xinyu Liu[7]
Yingya Zhang[8], Shiwei Zhang[8]†, Difan Zou[1]†

[1]The University of Hong Kong, [2]Carnegie Mellon University, [3]University of Sydney, [4]RIKEN AIP,
[5]Shanghai Artificial Intelligence Laboratory, [6]Shanghai Jiao Tong University, [7]Hong Kong University of
Science and Technology, [8]Alibaba Group

## Abstract

Although unified MLLMs aim to unify generation and understanding, they are considered to exhibit an internal gap, with understanding outperforming generation. Through large-scale evaluation across multiple MLLMs and tasks, we confirm the widespread non-unification of MLLMs, and demonstrate that it indeed stems from weak generation rather than misunderstanding. This finding motivates us to propose a simple yet effective internal gap-based self-improvement framework, which mitigates internal gaps by leveraging stronger understanding to guide weaker generation without relying on any external signals. We validate this strategy through comprehensive experiments: scoring generations with understanding to construct image data for post-training (e.g., SFT and DPO) significantly improves generation while promoting unification. Furthermore, we empirically discover a co-improvement effect of such self-improvement, a phenomenon well known in pre-training but underexplored in post-training. Specifically, as generation improves, understanding becomes more effective at detecting false positives that were previously misclassified as prompt-aligned. To explain this effect, we extend learning dynamic theory to the MLLM setting, showing that the shared empirical neural tangent kernel between generation and understanding encourages aligned learning dynamics, thereby driving co-improvement. This interplay between generation and understanding further motivates a curriculum learning approach for stronger self-improvement: progressively enhanced understanding and generation revisit samples underutilized by pre-trained MLLMs, dynamically expanding post-training data and leading to improved performance and unification.

## 1 Introduction

Unified Multimodal Large Language Models (MLLMs) have attracted growing attention for their capability to conduct both generation and understanding (Xie et al., 2024; Wu et al., 2024a; Wang et al., 2024; Team, 2025a; Zhou et al., 2024; Chen et al., 2025a). However, an emerging consensus is that, despite being designed to unify both generation and understanding, they are not truly unified in performance (Yang et al., 2025; Mao et al., 2025; Hong et al., 2025; Yan et al., 2025), where understanding typically outperforms generation (Yang et al., 2025). For example, Fig. 1 shows, an MLLM's generation may be judged as prompt-misaligned by its own understanding branch, revealing an internal generation–understanding gap. A natural question arises:

*Can the internal gap in MLLMs be leveraged as a free bonus, with the stronger branch guiding the weaker one to improve the model's performance and mitigate non-unification?*

---

*Email to: Yujin Han (yujinhan@connect.hku.hk).
†Correspondence to: Shiwei Zhang (zhangjin.zsw@alibaba-inc.com) and Difan Zou (dzou@cs.hku.hk).

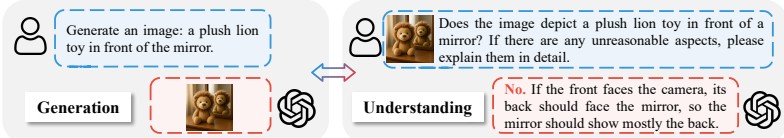

Figure 1: Illustration of MLLMs' internal gap. We examine a challenging case (Han et al., 2025) involving implicit physical principles using ChatGPT o3 (OpenAI, 2024) and find: images produced by generation branch are identified as incorrect by understanding branch, showing non-unification.

Prior works have discussed the internal gap in unified MLLMs, but their mitigation methods often rely on external reward models (Yang et al., 2025) or additional supervised datasets (Mao et al., 2025), or focus solely on improving a single task, e.g., generation (Jiang et al., 2025; Yan et al., 2025; Xie et al., 2025), without emphasizing generation–understanding alignment. In this paper, we explore the potential of mitigating MLLMs' non-unification without any external signals, and propose a simple yet effective internal gap-based self-improvement framework. We further provide a detailed analysis of the dynamic interplay between generation and understanding during self-improvement, offering a strong complement to existing studies.

We begin by validating the generation–understanding gap across multiple MLLMs and tasks. We first introduce an internal metric, *non-unification score*, defined as the proportion of cases where the understanding branch judges the generation as prompt-misaligned. Unlike previous unification metrics that rely on an external estimator (Yang et al., 2025; Mao et al., 2025), our metric directly quantifies the internal consistency between two branches, avoiding biases from external assessment. Comprehensive evaluation on six unified MLLMs and tasks of three difficulty levels shows that non-unification is pervasive, with non-unification score reaching up to 60%. Further quantitative analysis attributes most misalignments (60–100%) to *weak generation* rather than misunderstanding, consistent with prior findings on single tasks (Yang et al., 2025) and single models (Mao et al., 2025).

After confirming widespread non-unification and stronger understanding, we propose an **internal gap-based self-improvement framework** that aligns MLLMs by leveraging stronger understanding to guide the weaker generation. We validate its effectiveness on mainstream MLLMs such as Janus-Pro-7B (Chen et al., 2025b): using the understanding branch to score generations and construct post-training data for generation, standard pipelines, e.g., SFT (Brown et al., 2020; Radford et al., 2021) and DPO (Rafailov et al., 2024), significantly boost generation (up to +20% on T2I-CompBench++ (Huang et al., 2025)) and reduce the internal gap (non-unification score by as much as −16%), surpassing even baselines with multiple external reward models such as T2I-R1 (Jiang et al., 2025).

Furthermore, we empirically observe a *co-improvement* effect: the generation-targeted self-improvement method also enhances understanding. Specifically, self-improved MLLMs better detect false positives, i.e., samples previously misidentified as prompt-aligned. While co-improvement is well-known in pre-training (Tong et al., 2024; Wu et al., 2025a; Deng et al., 2025; Zhang et al., 2025; Wu et al., 2025b), it remains underexplored in post-training (Yang et al., 2025; Mao et al., 2025). To explain it, we extend learning dynamic theory (Ren & Sutherland, 2025) to multimodal settings and formalize joint evolution of generation and understanding during self-improvement. Our theory reveals a shared empirical neural tangent kernel (eNTK) facilitates consistent learning dynamics across generation and understanding. Consequently, aligned dynamics reduce misaligned generations and enhance misalignment detection, thus leading to co-improvement effect observed.

Finally, motivated by the co-improvement effect, we further demonstrate that *curriculum learning* (Elman, 1993; Bengio et al., 2009) can be incorporated into self-improvement by gradually introducing harder samples that were initially excluded due to limited capabilities in generation or understanding. Experiments show that curriculum learning enables self-improvement to dynamically expand post-training data, further enhancing both the performance and unification of MLLMs.

Through a systematic exploration of MLLMs' internal gap, our contributions are as follows:

- We first introduce the non-unification score, an internal consistency metric to measure MLLMs' internal gap. Extensive evaluations across diverse models and tasks confirm pervasive non-unification phenomenon, which is primarily caused by weak generation.

- Motivated by non-unification in MLLMs, we then propose a simple yet effective internal gap-based self-improvement framework, which leverages stronger understanding capability to guide

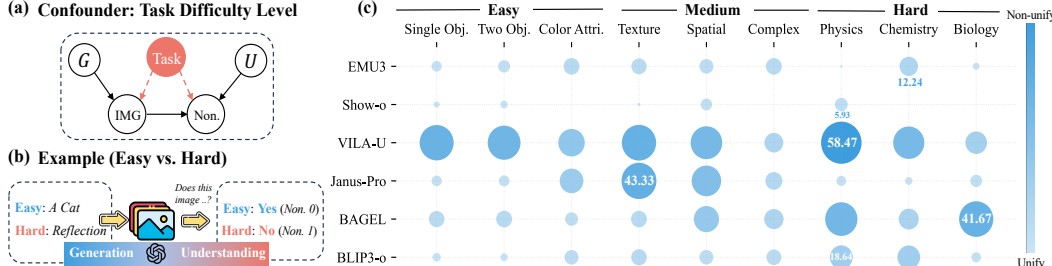

Figure 2: Verification of internal gaps. (a) and (b) identify task difficulty as a confounder in measuring non-unification score (*Non.*): easy tasks may underestimate the gap, while hard tasks risk overestimation. Stratifying by task difficulty (Easy–Medium–Hard) yields a more reliable estimation. (c) Evaluation of six MLLMs across three difficulty levels shows unified MLLMs remain non-unified, with non-unification scores approaching 60%. More details are provided in Appendix A.1.

the weaker generation. Extensive experiments show the proposed self-improvement significantly boosts both generation and unification without external signals.

- In self-improvement, we empirically identify a co-improvement effect, where understanding better detects prompt-misaligned generations. Extending learning dynamics to MLLMs, we attribute this effect to shared eNTK between generation and understanding.

- Finally, co-improvement effect inspires a curriculum-based self-improvement strategy: progressively strengthen understanding and generation enable reusing underutilized samples, thereby expanding post-training data and boosting both performance and unification.

## 2 RELATED WORK

**Non-unification of MLLMs.** There are works showing internal gap of MLLMs, typically with understanding outperforming generation (Yang et al., 2025; Mao et al., 2025; Hong et al., 2025; Yan et al., 2025; Yang et al., 2025). However, existing studies *lack systematic quantification* of such gap across multiple MLLMs and tasks, with conclusions often confined to single models (Mao et al., 2025) or single tasks (Yang et al., 2025). Additionally, their measurements of internal gap rely on external models, e.g., ChatGPT (Yang et al., 2025; Mao et al., 2025) *instead of measuring internal consistency*, which potentially makes biased estimation by external evaluators. Therefore, we first focus on introducing non-unification metric and performing large-scale verification.

**Mitigating Non-unification of MLLMs.** Several studies attempt to mitigate internal gap within MLLMs, but they rely on external models (Jiang et al., 2025; Yang et al., 2025) or additional data (Mao et al., 2025). For example, Hermesflow (Yang et al., 2025) leverages external Bert (Devlin et al., 2019) for understanding, self-critique and VQA (Antol et al., 2015) models for generation, to improve both branches. Other works (Jiang et al., 2025; Duan et al., 2025) enhance weaker generation by introducing multiple external reward models, e.g., BLIP (Li et al., 2022) and HPMs (Wu et al., 2023; Xu et al., 2023). In contrast, we focus on mitigating internal gap purely through self-improvement *without any external signals*. Importantly, self-improvement does not conflict with existing approaches: once achieved, external signals can be incorporated to further boost MLLMs.

**Co-improvement of MLLMs.** Co-improvement in unified MLLMs often refers to one branch improving when the other is improved, such as understanding gains from adding more generation data (Tong et al., 2024; Wu et al., 2025a). This phenomenon has been widely observed in pre-training (Tong et al., 2024; Wu et al., 2025a; Deng et al., 2025; Zhang et al., 2025; Wu et al., 2025b), yet it has *not been sufficiently highlighted* or *thoroughly analyzed* in post-training (Yang et al., 2025; Mao et al., 2025; Hong et al., 2025). Our work provides a learning-dynamics perspective on it, offering insights into interplay between understanding and generation in unified MLLMs.

## 3 PHENOMENON VERIFICATION: THE NON-UNIFICATION IN MLLMS

While prior work suggests internal imbalances in MLLMs, this claim remains unverified through evaluation across diverse models and tasks (see Section 2). We therefore take the large-scale empirical verification of non-unification as the starting point of our study.

We first propose a self-consistency metric to quantify the generation-understanding gap, termed the *non-unification score*. Specifically, consider an MLLM $\pi_\theta$, a prompt $\mathbf{y}$ and the generated image $\mathbf{x} = \pi_\theta^{\mathrm{gen}}(q(\mathbf{y}))$. We form an image–question pair $(\mathbf{x}, q(\mathbf{y}))$, where $q(\mathbf{y}) \coloneqq$ "Does this image describe $\mathbf{y}$?". This pair is processed by the understanding branch $\pi_\theta^{\mathrm{und}}(\cdot)$, yielding a binary decision: 1 if $\mathbf{x}$ is aligned with $\mathbf{y}$, and 0 otherwise. The *non-unification score* is the proportion of decisions equal to 0,

$$\text{Non-unification score} \coloneqq \mathbb{E}_{(\mathbf{x},\mathbf{y})} \mathbb{I}\left[\pi_\theta^{\mathrm{und}}\left(\mathbf{x},\ q(\mathbf{y})\right) = 0\right]. \tag{1}$$

Intuitively, unified MLLMs should have a near-zero non-unification score: generation renders the prompt as an image and understanding verifies the image matches the prompt.

We evaluate multiple MLLMs (Wang et al., 2024; Xie et al., 2024; Wu et al., 2024b; Chen et al., 2025b; Deng et al., 2025; Chen et al., 2025a), across tasks of varying difficulty. We emphasize that task difficulty is a confounder affecting both generation and understanding, thereby biasing the non-unification score (see Fig. 2(a)). For example, as shown in Fig. 2(b), a simple prompt like *generate a cat* makes both generation and understanding easy, so the non-unification score is close to zero and may underestimate the internal gap. In contrast, for harder tasks such as *generate a mirror reflection*, where the generation branch may fail to capture latent physical rules and the score may be overestimated. Therefore, stratifying by task difficulty provides a more reliable way to estimate the internal gap. Specifically, we construct nine subtasks of increasing difficulty from

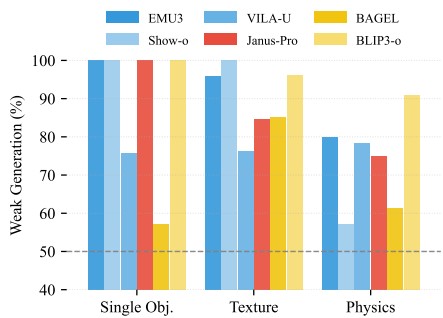

Figure 3: Weak-generation (Qwen-checked) above 50% (even 100%) indicate internal gap mainly stems from weak generation. Fig. 11 provides human check, showing conclusions consistent with Qwen. Appendix A.2 reports more Weak Generation results.

three benchmarks (Ghosh et al., 2023; Huang et al., 2025; Li et al., 2025), ranging from simple case (e.g., `a cat`) to complex prompts with implicit rules (e.g., `ice at 60°C`). Detailed tasks and MLLMs are shown in Appendix A.1.

**Results.** Fig. 2(c) demonstrates that all six evaluated models exhibit larger non-unification scores on hard (5/6) and medium (1/6) tasks. In contrast, the small non-unification scores observed on easy tasks may not indicate the absence (or near absence) of an internal gap, but rather that the tasks are too simple. On VILA-U (Wu et al., 2024b), it even reaches 58.47%, meaning that nearly 60% of generations are rejected (prompt misaligned) by understanding. More discussion on non-unification is provided in Appendix A.2.

To further distinguish whether non-unification comes from weak generation or misunderstanding, we use a stronger external model, Qwen2.5-VL-72B-Instruct (Bai et al., 2025), to check the accuracy of the understanding scores. Define Weak Generation as the probability that, when the MLLM's understanding branch rejects an output, its judgment agrees with Qwen, i.e.,

$$\text{Weak Generation} \coloneqq \mathbb{P}\left(\pi_\theta^{\mathrm{und}}(\mathbf{x},\ q(\mathbf{y})) = \pi_{\mathrm{Qwen}}^{\mathrm{und}}(\mathbf{x},\ q(\mathbf{y})) \,\middle|\, \pi_\theta^{\mathrm{und}}(\mathbf{x},\ q(\mathbf{y})) = 0\right).$$

Fig. 3 shows, across different task difficulties, all MLLMs achieve over 50% and up to 100% Weak Generation, indicating that the internal gap mainly stems from poor generation rather than misjudgments of understanding which well align with prior findings (Yang et al., 2025). Additionally, Appendix A.2 provides weak generation score computed using Gemini-Pro-2.5 (Team, 2025b) yielding conclusions consistent with Qwen: the internal gap primarily stems from weaker generation.

## 4 MITIGATING NON-UNIFICATION: A SELF-IMPROVEMENT FRAMEWORK

### 4.1 METHOD: INTERNAL GAP-BASED SELF-IMPROVEMENT

The observation that understanding consistently outperforms generation then motivates our *internal gap-based self-improvement* framework to promote unification of unified MLLMs, which leverages stronger understanding to enhance the weaker generation. Specifically, we adopt standard post-training strategies such as Direct Preference Optimization (DPO) and Supervised Fine-Tuning (SFT). Given an image generation prompt $\mathbf{y}$, the MLLM $\pi_\theta$ produces $N$ candidate images, i.e., $\{\mathbf{x}_i\}_{i=1}^N = \pi_\theta(\mathbf{y})$. Each candidate $\mathbf{x}_i$ is paired with the question as $q(\mathbf{y}) \coloneqq$

**Algorithm 1:** Self-Improvement (SFT)

**Input:** $\pi_\theta$, prompts $\mathcal{P}$, image candidates $N$, epochs $T$
**Data:** $\mathcal{D}_{\text{SFT}} \leftarrow \emptyset$, discard pool $\mathcal{B} \leftarrow \emptyset$
**for** $\mathbf{y} \in \mathcal{P}$ **do**
$\quad \{\mathbf{x}_i\}_{i=1}^N \leftarrow \pi_\theta^{\text{gen}}(\mathbf{y})$;
$\quad s_i \leftarrow \pi_\theta^{\text{und}}(\mathbf{x}_i, q(\mathbf{y})) \in \{0, 1\}$;
$\quad \mathcal{C} \leftarrow \{\mathbf{x}_i : s_i = 1\}$; **if** $|\mathcal{C}| = 0$ **then**
$\quad\quad \mathcal{B} \leftarrow \mathcal{B} \cup \{\mathbf{y}\}$
$\quad$ **else**
$\quad\quad \mathcal{D}_{\text{SFT}} \leftarrow \mathcal{D}_{\text{SFT}} \cup \{(\mathbf{y}, \mathbf{x}_{\text{chosen}}) \,|\, \mathbf{x}_{\text{chosen}} \in \mathcal{C}\}$

**for** $t = 1$ **to** $T$ **do**
$\quad \theta \leftarrow \theta - \eta \, \nabla_\theta \mathcal{L}_{\text{gen}}(\theta; \mathcal{D}_{\text{SFT}})$ ;

**Algorithm 2:** Curriculum Replay

**Input:** $\pi_\theta$, discard pool $\mathcal{B}$, image candidates $N$, curriculum epochs $\mathcal{E}_{cur}$
**Data:** $\mathcal{D}_{\text{SFT}}$ (shared with Alg. 1)

**for** $t \in \mathcal{E}_{cur}$ **do**
$\quad$ **for** $\mathbf{y} \in \mathcal{B}$ **do**
$\quad\quad \{\tilde{\mathbf{x}}_j\}_{j=1}^N \leftarrow \pi_\theta^{\text{gen}}(\mathbf{y})$;
$\quad\quad \tilde{s}_j \leftarrow \pi_\theta^{\text{und}}(\tilde{\mathbf{x}}_j, q(\mathbf{y}))$;
$\quad\quad \tilde{\mathcal{C}} \leftarrow \{\tilde{\mathbf{x}}_j : \tilde{s}_j = 1\}$; **if** $|\tilde{\mathcal{C}}| > 0$ **then**
$\quad\quad\quad \mathcal{D}_{\text{SFT}} \leftarrow \mathcal{D}_{\text{SFT}} \cup \{(\mathbf{y}, \mathbf{x}) \,|\, \mathbf{x} \in \tilde{\mathcal{C}}\}$;
$\quad\quad\quad$ remove $\mathbf{y}$ from $\mathcal{B}$

"Does this image describe $\mathbf{y}$?" and processed by understanding branch $\pi_\theta^{\text{und}}$. Images judged (most likely) as aligned with the prompt are labeled as *chosen*, while those judged (most likely) as misaligned are labeled as *rejected*, forming preference data $(\mathbf{y}, \mathbf{x}_{\text{chosen}}, \mathbf{x}_{\text{rejected}})$ for DPO and supervision pairs $(\mathbf{y}, \mathbf{x}_{\text{chosen}})$ for SFT on the generation branch. Appendix B.1 provides further details on post-training data construction, and Alg. 1 outlines the SFT-based self-improvement procedure.

## 4.2 EXPERIMENT: EFFECTIVENESS OF SELF-IMPROVEMENT ON MLLMS

We then show effectiveness of proposed self-improvement through following experiments.

### 4.2.1 SETUP

**Baseline and Data.** To validate self-improvement, we apply it to two baselines: Janus-Pro-7B (Chen et al., 2025b) and Show-o (Xie et al., 2024). We ablate which MLLM components to optimize (e.g., the LLM and vision aligner) and find that updating only the shared LLM yields substantial gains. Further details are in Appendix F.2. Experiments are conducted on T2I-CompBench++ (Huang et al., 2025), which provides about 6000 text prompts as post-training candidates. After data construction, classical post-training strategies, SFT and DPO, are applied for generation-focused self-improvement. Further implementation details are in Appendix B.

**Evaluation.** We compare self-improved and pre-trained [1] MLLMs on generation, unification and understanding. For generation, we follow T2I-CompBench++ metrics and measure unification by non-unification score. For understanding, we use win rate (excluding ties) (Zheng et al., 2023; Chen et al., 2024): given validation text prompts with images generated by pre-trained MLLMs, models judge prompt–image alignment. Win rate is the proportion of cases where the self-improved MLLM disagrees with the pre-trained one but agrees with the stronger external judge, e.g., Qwen2.5-VL-72B-Instruct. For example, if the models disagree on three samples and the self-improved model matches Qwen on two, win rate is $2/3$. Pre- and post-trained models with comparable understanding achieve a win rate of $0.5$. Win rate enables tracking changes in understanding and generation on the same task, facilitating analysis of two branches. Appendix B.1 includes additional metric descriptions, as well as win rates obtained using additional external judges, e.g., Gemini-Pro-2.5.

### 4.2.2 RESULTS

We summarize key findings under SFT as follows. The corresponding DPO results, largely consistent with SFT, are provided in Appendix B.2.

**Finding 1: Internal gap-based self-improvement effectively improves generation and promotes MLLM unification.** Fig. 4 shows self-improved MLLMs can achieve up to 20% gains in generation and up to 16% in unification, validating effectiveness of proposed method. Moreover, we find improvements in generation are significantly correlated with unification ($\rho_{\Delta,\text{Non.}} = 0.53$). Specially, for model level, Janus-Pro, with a larger internal gap (see Fig. 2), achieves greater gains than

---

[1]For clarity, we name MLLMs without self-improvement as pre-trained MLLMs, even if they may undergo post-training phases during training.

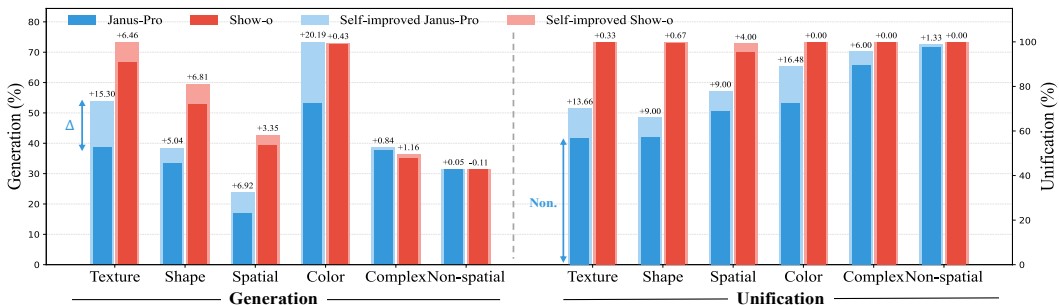

Figure 4: Self-improvement enhances generation and unification, with gains up to 20% and 16% (1−non-unification score). Furthermore, improvements correlate with the internal gap (correlation coefficient $\rho_{\Delta,\text{Non.}} = 0.53$): models and subtasks with larger gaps benefit more.

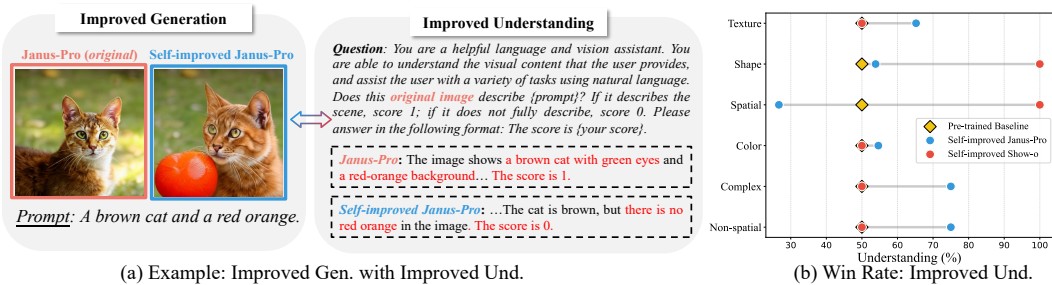

(a) Example: Improved Gen. with Improved Und.                (b) Win Rate: Improved Und.

Figure 5: The Co-improvement Effect. (a) illustrates an example where self-improved Janus-Pro generates prompt-aligned images and correctly scores the original as mismatched (see more cases in Appendix B); (b) reports win rates mostly above 50%, showing self-improved MLLMs judge prompt–image alignment more accurately than pre-trained ones.

Show-o with a smaller gap. For task level, subtasks with lower unification (e.g., `Texture`) benefit more. We attribute this to internal gap–based method encouraging more post-training samples from larger-gap subtasks, thereby enabling greater improvements. Fig. 12 further confirms this by showing post-training data contain a higher proportion of samples from larger-gap subtasks.

**Finding 2: Generation-targeted self-improvement also enhances understanding, showing a co-improvement effect.** Fig. 5(a) shows an example that, in addition to generating more prompt-aligned images, the self-improved MLLM also better detects mismatches between the original image and the prompt. Fig. 5(b) further reports high win rates for Janus-Pro and Show-o across six subtasks. For instance, the self-improved Janus-Pro achieves a win rate above 50% on 5 of 6 subtasks, indicating higher accuracy than its pre-trained counterpart in judging prompt–image alignment. Additionally, we also provide results on standard understanding benchmarks in Table 8, where self-improved MLLM consistently outperforms the pre-trained model. Furthermore, in Appendix F.1, we discuss additional unified MLLMs, such as BAGEL (Deng et al., 2025), which adopt different architectures (e.g., MoT rather than a shared LLM), and present their self-improvement results.

## 5 UNDERSTANDING CO-IMPROVEMENT IN SELF-IMPROVEMENT

Section 4.2.2 reveals a co-improvement effect in self-improvement, an underexplored phenomenon in unified MLLMs (see Section 2). Understanding this effect is crucial, as it highlights the unique interplay between generation and understanding and may inspire more effective self-improvement.

### 5.1 LEARNING DYNAMICS OF GENERATION AND UNDERSTANDING

We extend the learning dynamics framework (Ren & Sutherland, 2025) to the multimodal setting, as it provides a principled way to analyze how MLLMs $\pi_\theta$ evolve after self-improvement on post-training data $(\mathbf{y}_u, \mathbf{x}_u)$. Specifically, the framework helps to answer: (1) *Generation*: given a text input $\mathbf{y}_0$, how generated images of the self-improved model differs from that of the base model;

(2) *Understanding*: given an image input $\mathbf{x}_0$, how understanding output of the self-improved model differs from that of the base model.

Suppose $\mathbf{x}_0$ (from the pre-trained MLLM) and $\mathbf{y}_0$ are misaligned. If generation and understanding share aligned learning dynamics, e.g., jointly decreasing incorrect generation $\pi_\theta(\mathbf{x}_0|\mathbf{y}_0)$ and misunderstanding $\pi_\theta(\mathbf{y}_0|\mathbf{x}_0)$, the co-improvement occurs.

**Settings.** We first consider the setting where generation and understanding *share the same tokenizer*, as in Show-o and EMU3 (Wang et al., 2024). This contrasts with decoupled designs (e.g., Janus-Pro) that use separate tokenizers. Nevertheless, our later analysis in Section 5.2 indicates that the conclusions drawn under the shared-tokenizer assumption also apply to decoupled architectures. Additionally, our theoretical framework can also be extended to MLLMs that employ diffusion models for modeling continuous image tokens (Xie et al., 2024; Zhou et al., 2024). Then, we denote $\mathcal{V}$ as the unified vocabulary of text and image tokens with size $V = |\mathcal{V}|$. Given a validation example $(\mathbf{y}_0, \mathbf{x}_0)$, with image token sequence $\mathbf{x}_0 = (x_{0,1}, \ldots, x_{0,M})$ of length $M$ and text token sequence $\mathbf{y}_0 = (y_{0,1}, \ldots, y_{0,L})$ of length $L$, our goal is to analyze how MLLM's generation and understanding outputs on $(\mathbf{x}_0, \mathbf{y}_0)$ change after self-improvement on the post-training sample $(\mathbf{y}_u, \mathbf{x}_u)^2$.

Following Ren & Sutherland (2025), we adopt standard causal masking in MLLMs (Wu et al., 2024a; Wang et al., 2024; Wu et al., 2025b) and define the input to generation branch as $\mathcal{Y}_0 = [\,\mathbf{y}_0 \mid \mathbf{x}_0\,] \in \mathbb{R}^{d \times (M+L)}$, and input to understanding branch as $\mathcal{X}_0 = [\,\mathbf{x}_0 \mid \mathbf{y}_0\,] \in \mathbb{R}^{d \times (M+L)3}$. We denote the logit network as $h_\theta$, which outputs understanding and generation logits $\mathbf{z}^0_{\text{und}} := h_\theta(\mathcal{X}_0)_{[:,\, M+1:M+L]}$ and $\mathbf{z}^0_{\text{gen}} := h_\theta(\mathcal{Y}_0)_{[:,\, L+1:L+M]}$ respectively. We define the likelihood of sample $(\mathbf{y}_0, \mathbf{x}_0)$ under generation and understanding branch with *shared LLM* as

$$\pi_\theta(\mathbf{x}_0 \mid \mathcal{Y}_0) = \prod_{k=1}^{M} \pi_\theta(x_{0,k} \mid \mathbf{y}_0, \mathbf{x}_{0,<k}) = \prod_{k=1}^{M} \left[\text{softmax}(\mathbf{z}^0_{\text{und}})\right]_{x_{0,k},k} \qquad \text{(Generation)}$$

$$\pi_\theta(\mathbf{y}_0 \mid \mathcal{X}_0) = \prod_{\ell=1}^{L} \pi_\theta(y_{0,\ell} \mid \mathbf{x}_0, \mathbf{y}_{0,<\ell}) = \prod_{\ell=1}^{L} \left[\text{softmax}(\mathbf{z}^0_{\text{gen}})\right]_{y_{0,\ell},\ell} \qquad \text{(Understanding)}$$

where the softmax is applied column-wise.

**One-step learning dynamics.** At epoch $t$, we define the one-step learning dynamics of evaluation data pair $(\mathbf{y}_0, \mathbf{x}_0)$ likelihood after training one-step on post-training data $(\mathbf{y}_u, \mathbf{x}_u)$ as $\Delta G_t(\mathbf{x}_0 \mid \mathcal{Y}_0) := \log \pi_{\theta_{t+1}}(\mathbf{x}_0 \mid \mathcal{Y}_0) - \log \pi_{\theta_t}(\mathbf{x}_0 \mid \mathcal{Y}_0)$ for generation branch and $\Delta U_t(\mathbf{y}_0 \mid \mathcal{X}_0) := \log \pi_{\theta_{t+1}}(\mathbf{y}_0 \mid \mathcal{X}_0) - \log \pi_{\theta_t}(\mathbf{y}_0 \mid \mathcal{X}_0)$ for understanding branch. We consider self-improvement with SFT and relate the dynamics of understanding and generation in the following proposition. Self-improvement with DPO are analyzed in Appendix D.2.

**Proposition 1** (Learning Dynamics of Generation and Understanding under SFT). *Consider self-improvement proposed in Section 4 with SFT and at epoch $t$.*

*The one-step learning dynamics of **generation** is*

$$\Delta G_t(\mathbf{x}_0 \mid \mathcal{Y}_0) = -\eta \sum_{k=1}^{M} \sum_{r=1}^{M} (\mathbf{e}_{x_{0,k}} - \pi^0_k)^\top \mathcal{K}^t_{k,r}(\mathcal{Y}_0, \mathcal{Y}_u)(\pi^u_r - \mathbf{e}_{x_{u,r}}) + \mathcal{O}(\eta^2), \qquad (2)$$

*where $\pi^u_r = \text{softmax}(\mathbf{z}^u_r)$ and $\mathbf{z}^u_r = [h_\theta(\mathcal{Y}_u)]_r$ are the logits at position $r$ obtained by running $h_\theta$ on $\mathcal{Y}_u$ and $\mathcal{K}^t_{k,r}(\mathcal{Y}_0, \mathcal{Y}_u) := (\nabla_{\theta_t}\mathbf{z}^0_k)(\nabla_{\theta_t}\mathbf{z}^u_r)^\top \in \mathbb{R}^{V \times V}$ is empirical neural tangent kernel (eNTK).*

*The one-step learning dynamics of **understanding** is*

$$\Delta U_t(\mathbf{y}_0 \mid \mathcal{X}_0) = -\eta \sum_{k=1}^{M} \sum_{r=1}^{M} \sum_{\mathbf{y}_i \neq \mathbf{y}_0} w_{\theta_t}(\mathbf{y}_i \mid \mathbf{x}_0) \left( (\mathbf{e}_{x_{0,k}} - \pi^0_k)^\top \mathcal{K}^t_{k,r}(\mathcal{Y}_0, \mathcal{Y}_u) - (\mathbf{e}_{x_{0,k}} - \pi^i_k)^\top \mathcal{K}^t_{k,r}(\mathcal{Y}_i, \mathcal{Y}_u) \right)(\pi^u_r - \mathbf{e}_{x_{u,r}})$$
$$+ \mathcal{O}(\eta^2)$$
$$(3)$$

*where $w_{\theta_t}(\mathbf{y} \mid \mathbf{x}_0) := \frac{\pi_{\theta_t}(\mathbf{x}_0|\mathbf{y})}{\sum_{\mathbf{y}'} \pi_{\theta_t}(\mathbf{x}_0|\mathbf{y}')}$ and $\mathcal{Y}_i$ denotes the concatenation of prompt $\mathbf{y}_i \neq \mathbf{y}_0$ and $\mathbf{x}_0$.*

---

²Appendix E provides more detailed preliminaries.

³We omit potential special tokens (e.g., [SOI]) for simplicity.

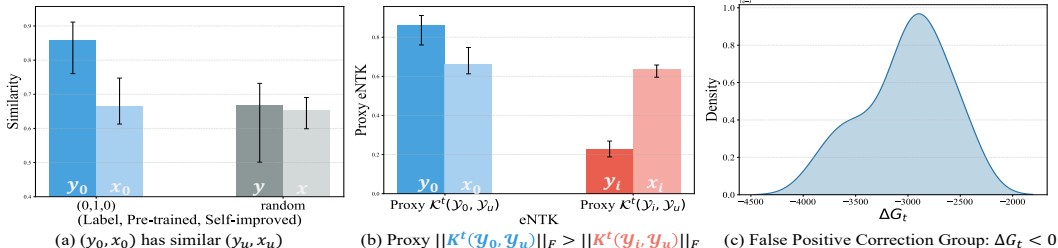

(a) $(y_0, x_0)$ has similar $(y_u, x_u)$

(b) Proxy $\|K^t(\mathcal{y_0}, \mathcal{y_u})\|_F > \|K^t(\mathcal{y_i}, \mathcal{y_u})\|_F$

(c) False Positive Correction Group: $\Delta G_t < 0$

Figure 7: Empirical Evidence from Self-improved Janus-Pro with SFT. (a) Compared to random samples, $(\mathbf{y}_0, \mathbf{x}_0)$ in the false positive correction group are more likely to be matched with highly similar post-training pairs $(\mathbf{y}_u, \mathbf{x}_u)$. (b) Such high data similarity makes $\mathcal{K}_{k,r}^t(\mathcal{Y}_0, \mathcal{Y}_u)$ be the dominant term in Equation (3), thereby promoting aligned learning dynamics $\Delta G_t$ and $\Delta U_t$. (c) With aligned dynamics, $\Delta G_t < 0$ implies $\Delta U_t < 0$: both the probability of mis-generation $\pi_\theta(\mathbf{x}_0 \mid \mathbf{y}_0)$ and misjudging $\pi_\theta(\mathbf{y}_0 \mid \mathbf{x}_0)$, are reduced, i.e., false positive correction and co-improvement occur.

Proposition 1 shows the learning dynamics of generation ($\Delta G_t$ in Equation (2)) and understanding ($\Delta U_t$ in Equation (3)) are similar. The key difference is that $\Delta U_t$ includes an additional eNTK term, $\mathcal{K}_{k,r}^t(\mathcal{Y}_i, \mathcal{Y}_u)$, which measures alignment between $\mathcal{Y}_i$ ($i \neq 0$) and the post-training data $\mathcal{Y}_u$.

*We therefore hypothesize*: for co-improved pair $(\mathbf{y}_0, \mathbf{x}_0)$, there likely exist post-training samples $(\mathbf{y}_u, \mathbf{x}_u)$ that are highly similar, leading to $\|\mathcal{K}_{k,r}^t(\mathcal{Y}_0, \mathcal{Y}_u)\|_F \geq \|\mathcal{K}_{k,r}^t(\mathcal{Y}_i, \mathcal{Y}_u)\|_F$. Hence, understanding update $\Delta U_t$ in Equation (3) is dominated by $\mathcal{K}_{k,r}^t(\mathcal{Y}_0, \mathcal{Y}_u)$, which is a *shared eNTK* term with the generation update $\Delta G_t$ in Equation (2). Aligned updates between generation and understanding, i.e., aligned $\Delta G_t$ and $\Delta U_t$, can jointly reduce the probabilities of mis-generation $\pi_\theta(\mathbf{x}_0 \mid \mathbf{y}_0)$ and misunderstanding $\pi_\theta(\mathbf{y}_0 \mid \mathbf{x}_0)$, thereby yielding co-improvement.

To test this hypothesis, we combine empirical evidences from Section 4.2 with theoretical results in Proposition 1, and empirically examine: for a sample $(\mathbf{y}_0, \mathbf{x}_0)$ of which understanding improves, whether there exist similar post-training samples $(\mathbf{y}_u, \mathbf{x}_u)$. Such similarity may render $\Delta U_t$ dominated by the eNTK term $\mathcal{K}_{k,r}^t(\mathcal{Y}_0, \mathcal{Y}_u)$, which is shared by both generation and understanding branch, thereby aligning the updates of the two branches, i.e., aligned $\Delta G_t$ and $\Delta U_t$.

## 5.2 EMPIRICAL EVIDENCE

First, samples where understanding improves can be classified into two cases: (1) *False Positive Correction*: when image $\mathbf{x}_0$ and text $\mathbf{y}_0$ are actually misaligned (Qwen label = 0), pre-trained MLLMs incorrectly judge them as aligned (score = 1), while self-improved MLLMs correctly predict misalignment (score = 0); (2) *False Negative Correction*: when $\mathbf{x}_0$ and $\mathbf{y}_0$ are aligned (Qwen label = 1), pre-trained MLLMs incorrectly predict misalignment (score = 0), while self-improved MLLMs correctly judge alignment (score = 1). Using self-improvement with SFT on Janus-Pro as an example, Figure 6 shows approximately 80% of the understanding improvement originates from case (1), i.e., false positive correction.

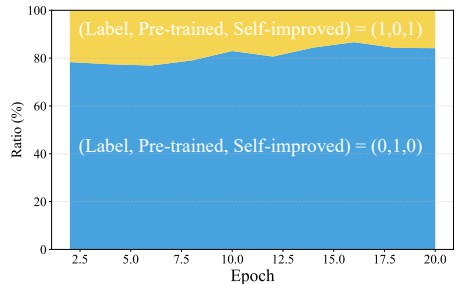

Figure 6: On T2I-CompBench++, understanding gains primarily (80%) arise from false positive correction. See Appendix D for results on additional MLLMs.

Our verification below mainly focuses on *false positive correction*. Specifically, consider prompt $\mathbf{y}_0$ and misaligned image $\mathbf{x}_0$ (generated by pre-trained MLLMs).

- Fig. 7(a) identifies, for each $(\mathbf{y}_0, \mathbf{x}_0)$ in the false-positive correction group, its most similar counterpart in the post-training data, showing that these samples typically have higher-similarity post-training pairs $(\mathbf{y}_u, \mathbf{x}_u)$. In particular, the prompt $\mathbf{y}_0$ attains an average similarity of about 0.8, significantly higher than a randomly sampled reference.

| Model | ES | IS | CL[4] | Texture | | | Shape | | | Spatial | | | Color | | | Complex | | | Non-spatial | | |
|---|---|---|---|---|---|---|---|---|---|---|---|---|---|---|---|---|---|---|---|---|---|
| | | | | Gen.↑ | Und.↑ | Non.↓ | Gen.↑ | Und.↑ | Non.↓ | Gen.↑ | Und.↑ | Non.↓ | Gen.↑ | Und.↑ | Non.↓ | Gen.↑ | Und.↑ | Non.↓ | Gen.↑ | Und.↑ | Non.↓ |
| *Gen. only* | | | | | | | | | | | | | | | | | | | | | |
| StrucDiffusion (Feng et al., 2022) | ✗ | ✗ | ✗ | 49.00 | – | – | 42.18 | – | – | 13.86 | – | – | 49.90 | – | – | 33.55 | – | – | 31.11 | – | – |
| CompDiffusion (Liu et al., 2022) | ✗ | ✗ | ✗ | 36.45 | – | – | 32.99 | – | – | 8.00 | – | – | 40.63 | – | – | 28.98 | – | – | 29.80 | – | – |
| Attend&Excite (Chefer et al., 2023) | ✗ | ✗ | ✗ | 59.63 | – | – | 45.17 | – | – | 14.55 | – | – | 64.00 | – | – | 34.01 | – | – | 31.09 | – | – |
| PixArt-$\alpha$ (Chen et al., 2023) | ✗ | ✗ | ✗ | 64.77 | – | – | 49.27 | – | – | 20.64 | – | – | 66.90 | – | – | 34.33 | – | – | **31.97** | – | – |
| CoMat (Jiang et al., 2024) | ✗ | ✗ | ✗ | 64.68 | – | – | 53.29 | – | – | 24.28 | – | – | **78.27** | – | – | 36.80 | – | – | 31.87 | – | – |
| SDv1.5 (Rombach et al., 2022) | ✗ | ✗ | ✗ | 41.86 | – | – | 37.13 | – | – | 11.65 | – | – | 37.58 | – | – | 30.47 | – | – | 31.12 | – | – |
| SD-XL-base-1.0 (Podell et al., 2023) | ✗ | ✗ | ✗ | 52.99 | – | – | 46.87 | – | – | 21.31 | – | – | 58.79 | – | – | 32.37 | – | – | 31.19 | – | – |
| FLUX.1 (Labs, 2024) | ✗ | ✗ | ✗ | 69.22 | – | – | 57.18 | – | – | 28.63 | – | – | 74.07 | – | – | 37.03 | – | – | 31.27 | – | – |
| *Gen. and Und.* | | | | | | | | | | | | | | | | | | | | | |
| Janus-Pro-7B (Chen et al., 2025b) | ✗ | ✗ | ✗ | 38.63 | 50.00 | 43.33 | 33.49 | 50.00 | 43.00 | 16.81 | 50.00 | 31.00 | 53.22 | 50.00 | 27.33 | 37.73 | 50.00 | 10.33 | 31.40 | 50.00 | 2.33 |
| T2I-R1 (Jiang et al., 2025)[5] | ✓ | ✗ | ✗ | 50.91 | 52.50 | 34.67 | 37.80 | 53.49 | 36.00 | 24.22 | 45.00 | 23.67 | 70.47 | 35.29 | 11.33 | 38.53 | 72.73 | 3.33 | 31.38 | 75.00 | 1.00 |
| *Self-improved Janus-Pro-7B* | | | | | | | | | | | | | | | | | | | | | |
| + *SFT* | ✗ | ✓ | ✗ | 53.93 | 65.22 | 29.67 | 38.63 | 53.85 | 34.00 | 23.73 | 26.67 | 22.00 | 73.41 | 54.62 | 10.85 | 38.57 | 75.00 | 4.33 | 31.45 | 75.00 | 1.00 |
| + *C-SFT* | ✗ | ✓ | ✓ | 56.38 | 66.67 | 28.33 | 39.86 | 64.52 | 33.67 | 24.87 | 38.46 | 21.67 | 73.77 | 52.14 | 12.20 | **38.78** | 70.00 | 3.33 | 31.44 | 75.00 | 2.33 |
| *Gen. and Und.* | | | | | | | | | | | | | | | | | | | | | |
| Show-o (Xie et al., 2024) | ✗ | ✗ | ✗ | 66.80 | 50.00 | 0.33 | 52.72 | 50.00 | 0.67 | 39.31 | 50.00 | 4.67 | 72.50 | 50.00 | **0.00** | 35.17 | 50.00 | **0.00** | 31.43 | 50.00 | **0.00** |
| Hermesflow (Yang et al., 2025) | ✓ | ✓ | ✗ | 67.96 | 50.00 | 0.33 | 51.81 | 50.00 | 0.33 | 38.45 | 0.00 | 4.00 | 72.96 | 50.00 | 0.34 | 35.28 | 50.00 | **0.00** | 31.42 | 50.00 | **0.00** |
| *Self-improved Show-o* | | | | | | | | | | | | | | | | | | | | | |
| + *SFT* | ✗ | ✓ | ✗ | 73.26 | 50.00 | **0.00** | 59.53 | 100.00 | **0.00** | 42.66 | 100.00 | 0.67 | 72.93 | 50.00 | **0.00** | 36.33 | 50.00 | **0.00** | 31.32 | 50.00 | **0.00** |
| + *C-SFT* | ✗ | ✓ | ✓ | **74.11** | 50.00 | **0.00** | **59.75** | 100.00 | **0.00** | **42.70** | 100.00 | 0.33 | 72.38 | 50.00 | **0.00** | 36.42 | 50.00 | **0.00** | 31.53 | 50.00 | **0.00** |

Table 1: Curriculum learning-based self-improvement (*C-SFT*) yields better generation (higher Gen.) and understanding (higher Und.), and alleviates non-unification (lower Non.). which even surpasses baselines rely on external reward models, such as T2I-R1 (built on Janus-Pro-7B) and HermesFlow (built on Show-o). Additional post-training strategy, e.g., DPO, and evaluations on more benchmarks are provided in Appendix C.2.

- Fig. 7(b) supports that, the understanding branch of data in false positive correction group is dominated by the eNTK term $\mathcal{K}_{k,r}^{t}(\mathcal{Y}_0, \mathcal{Y}_u)$. Since $\mathcal{Y}_0$ is the concatenation of $\mathbf{y}_0$ and $\mathbf{x}_0$, we use the similarity between $(\mathbf{y}_0, \mathbf{x}_0)$ and its nearest post-training counterpart $(\mathbf{y}_u, \mathbf{x}_u)$ as a proxy for the eNTK. Based on proxies, we consistently observe $\|\mathcal{K}_{k,r}^{t}(\mathcal{Y}_0, \mathcal{Y}_u)\|_F \geq \|\mathcal{K}_{k,r}^{t}(\mathcal{Y}_i, \mathcal{Y}_u)\|_F$.

- Fig. 7(c) shows, for false positive correction samples, the generation update satisfies $\Delta G_t < 0$, i.e., the mis-generation probability $\pi_\theta(\mathbf{x}_0 \mid \mathbf{y}_0)$ decreases. Combined with Fig. 7(a)(b), this further implies $\Delta U_t < 0$, meaning the misunderstanding probability $\pi_\theta(\mathbf{y}_0 \mid \mathbf{x}_0)$ also decreases.

The above empirical evidence supports the hypothesis derived from Proposition 1, explaining both the emergence of false positive correction and co-improvement. We provide details on how each empirical result, e.g., the proxy of eNTK, was obtained and interpreted in Appendix D.

# 6 CURRICULUM LEARNING FOR STRONGER SELF-IMPROVEMENT

Co-improvement effect motivates a curriculum learning (Elman, 1993; Bengio et al., 2009) approach for stronger self-improvement: as generation and understanding improve together, difficult samples that pre-trained MLLMs could not previously utilize (due to weak generation or inaccurate understanding) can be incorporated later, forming an adaptive data expansion process based on prompt complexity (Li & Zhang, 2025). To demonstrate co-improvement incorporates more unused prompts, we compare two settings: (1) jointly improving generation and understanding, and (2) enhancing only a single branch (e.g., generation). As shown in Table 2, co-improvement contributes about 1000 additional samples from discard pool $\mathcal{B}$ (defined in Alg. 1) versus roughly 600 for single-branch enhancement, supporting our motivation. Alg. 2 shows details of curriculum learning.

**Setup.** Following the experimental setup in Section 4.2.1, we adopt self-improvement with curriculum learning strategy. For Janus-Pro and Show-o, curriculum learning is introduced at epoch 10, during which the models regenerate and rescore previously unused prompts to produce additional post-training samples. Evaluation follows the same metrics in Section 4.2.1. We provide more implementation details in Appendix C and ablation study in Appendix F.4.

| | Und. | Self-improved Und. |
|---|---|---|
| **Gen.** | 0 | 649 |
| **Self-improved Gen.** | 603 | **1091** |

Table 2: Co-improvement (self-improved both Und. and Gen.) adds 1091 samples from discard pool $\mathcal{B}$, compared to roughly 600 when improving only a single branch.

**Baseline.** Apart from generation-only models, e.g., SDv1.5 (Rombach et al., 2022), we consider two unified MLLM baselines: T2I-R1 (Jiang et al., 2025) improves generation of Janus-Pro-7B by using multiple external reward models and provides comparison for Janus-Pro-7B-based self-improvement. And Hermesflow (Yang et al., 2025) similarly employs external reward models, e.g., Bert (Devlin et al., 2019), to enhance Show-o, serving as a reference for Show-o-based approach.

**Results.** We report only SFT-based self-improvement with curriculum learning (denoted as *C-SFT*) on T2I-CompBench++ evaluation set. Results for DPO-based method and additional benchmarks, such as GenEval (Ghosh et al., 2023) and Science-T2I (Li et al., 2025), are provided in Appendix C.2. As shown in Table 1, incorporating curriculum learning enables unified MLLMs to achieve stronger self-improvement: compared with standard self-improvement, *C-SFT* delivers consistent gains in generation, understanding, and unification across most tasks, even surpassing baselines that rely on external rewards, such as T2I-R1 and Hermesflow. These results confirm the effectiveness of incorporating curriculum learning into the self-improvement process.

## 7 CONCLUSION AND LIMITATION

This paper systematically investigates generation–understanding gap in MLLMs through empirical validation, mitigation, mechanistic analysis and improved method design, showing gap-based self-improvement mitigates non-unification and induces co-improvement between the two branches.

This work has the following limitations. First, our exploration is restricted to limited unified MLLMs, such as Janus-Pro and Show-o. Second, we attribute the observed co-improvement to shared eNTK between generation and understanding. A deeper question, however, is why such NTK sharing arises in unified MLLMs, which calls for further investigation into model's mechanisms.

## REPRODUCIBILITY STATEMENT

We provide experimental details in Appendix A, Appendix B, and Appendix C, including evaluation tasks and hyperparameters, to ensure the reproducibility of experiments in Section 3, Section 4 and Section 6. The proof derivations are presented in Appendix D and Appendix E to guarantee the reproducibility of theoretical results in Section 5.

## ETHICS STATEMENT

This work aims to explore and mitigate the internal generation-understanding gap in unified MLLMs. All experiments are conducted on publicly available datasets and open-source models, ensuring that no private or sensitive data are involved. Our study focuses on explaining the phenomenon and developing mitigation methods, and does not directly deploy downstream applications that could raise ethical concerns.

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

# Appendix

CONTENTS

## A  ADDITIONAL DETAILS AND FULL RESULTS ON INTERNAL GAP

### A.1  ADDITIONAL DETAILS

In this section, we provide an overview of MLLMs and tasks evaluated in Section 3. Unified MLLMs aim to integrate generation and understanding, with common approaches including extending understanding MLLMs with external diffusion models for generation (Dong et al., 2023; Tong et al., 2024; Ge et al., 2024; Yang et al., 2024; Tian et al., 2024; Chen et al., 2025a; Xie et al., 2024), or representing both images and text as discrete tokens and training unified transformers under autoregressive paradigm (Team, 2025a; Zhou et al., 2024; Qu et al., 2024; Chen et al., 2025b; Wang et al., 2024). Despite aiming to unify tasks, most MLLMs emphasize single-task SOTA performance while overlooking models' internal alignment. Intuitively, truly unified MLLMs should maintain internal consistency between generation and understanding. Therefore, we first quantify at scale the non-unification problem in unified MLLMs.

**Evaluated MLLMs**   Our evaluation covers the following MLLMs:

- **EMU3** (Wang et al., 2024) is a unified model for both generation and understanding, which converts multiple modalities such as images, text, and video into discrete tokens, and performs next-token prediction in mixed multimodal sequences based on an LLM-style transformer architecture. EMU3 pursues maximal architectural unification between generation and understanding, sharing the same image tokenizer for both tasks and employing a common LLM backbone for generation and understanding.

- **Show-o** (Xie et al., 2024) also follows an LLM-style transformer architecture and an autoregressive paradigm. In its default setting, generation and understanding share the same visual understanding/generation encoder and LLM component. A distinctive feature of Show-o is that it adopts different attention mechanisms for text and image tokens: causal attention for the former and full attention for the latter. Moreover, for image tokens during training, it is modeled using discrete diffusion and incorporates a mask token prediction mechanism similar to that of MaskGIT (Chang et al., 2022).

- **VILA-U** (Wu et al., 2024b) also adopts a shared LLM and a unified next-token prediction paradigm to integrate generation and understanding tasks. To better learn the discrete token sequences resulting from concatenated images and text, VILA-U innovatively trains a unified foundation vision tower by applying a CLIP-like contrastive loss (Radford et al., 2021) between visual and textual tokens, while simultaneously enforcing accurate reconstruction of images after the decoder. This design promotes the performance of unified MLLMs.

- **Janus-Pro** (Chen et al., 2025b) differs slightly from the above models. While continuing to follow the LLM-style shared transformer and autoregressive paradigm, it emphasizes decoupling generation and understanding tasks at the tokenizer stage. By employing separate image tokenizers for the two tasks, Janus-Pro aims to mitigate conflicts arising from using a single unified tokenizer to serve tasks which require different representations.

- **BAGEL** (Deng et al., 2025), in contrast, adopts an architecture that explicitly separates generation and understanding. Inspired by the Mixture-of-Transformers (MoT) paradigm (Liang et al., 2025), BAGEL employs two dedicated transformer experts to handle the two types of information, respectively. The only point of interaction between the tasks is through the self-attention mechanism within each transformer block, while other components, such as visual tokenizers and FFN, are fully decoupled by task.

- **BLIP3-o** (Chen et al., 2025a), compared with the aforementioned models, adopts an even more decoupled design by combining an autoregressive paradigm with diffusion models. Specifically, BLIP3-o follows an understand-then-generate pipeline: it first performs image understanding using an pre-trained understanding MLLM (e.g., Qwen2.5-VL) to produce visual features that serve as semantic-level conditions for the subsequent image generation task. Then, leveraging these semantic conditions, DiT (Peebles & Xie, 2023) learn the distribution of the original image representations in the CLIP (Radford et al., 2021) embedding space via flow matching. During inference, a diffusion-based visual decoder will reconstruct pixel-level images from the CLIP representations generated by the DiT.

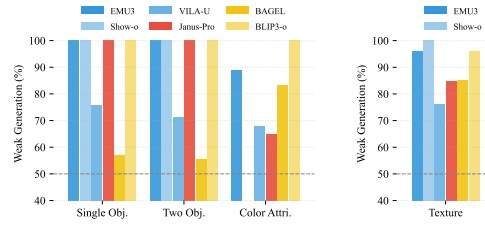 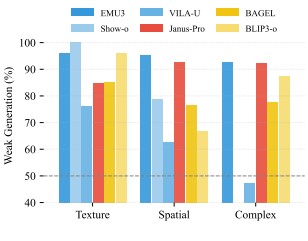 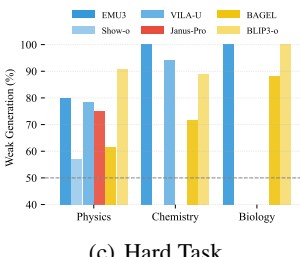

|   (a) Easy Task   |   (b) Medium Task   |   (c) Hard Task   |
|---|---|---|

Figure 8: Full Results on Weak Generation based on Qwen2.5-VL-72B-Instruct. Our evaluation across six MLLMs and nine tasks indicates that the primary cause of non-unification is weak generation, as reflected by weak generation scores exceeding 50% on the majority of tasks.

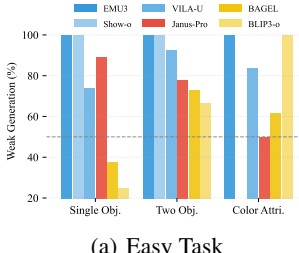 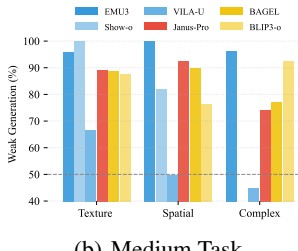 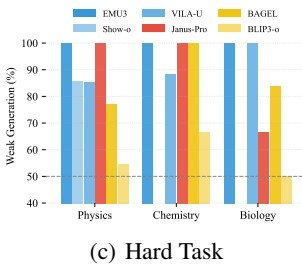

|   (a) Easy Task   |   (b) Medium Task   |   (c) Hard Task   |
|---|---|---|

Figure 9: Full Results on Weak Generation based on Gemini-Pro-2.5. Our evaluation across six MLLMs and nine tasks indicates that the primary cause of non-unification is weak generation, as reflected by weak generation scores exceeding 50% on the majority of tasks.

**Evaluated Task** We select nine subtasks from three benchmarks: GenEval (Ghosh et al., 2023), T2I-CompBench++ (Huang et al., 2025), and Science-T2I (Li et al., 2025). We then categorize subtasks into three difficulty levels (Easy, Medium, Hard) according to the complexity of generation and understanding required. Table 3 provides a detailed description of each subtask. We observe that Easy subtasks focus on the generation and understanding of *simple single objects*, e.g., `a cat`. Medium subtasks introduce relatively complex understanding such as spatial relationships (e.g., `on the top of`) that are typically made *explicit* in prompts, and Hard subtasks involve *implicit* reasoning not stated in the prompt, e.g., `tree in winter`, requiring MLLMs to leverage strong prior knowledge about physics, chemistry, and biology.

## A.2 FULL RESULTS

**Full Results.** Following the non-unification score defined in Section 3, we evaluate six MLLMs on subtasks across three difficulty levels and observe the widespread presence of the internal gap, as shown in Figure 2. In addition, we find substantial variation in non-unification across MLLMs. Show-o and EMU3 exhibit relatively small internal gaps, whereas recent models such as BAGEL and BLIP3-o have larger gaps but stronger performance (Deng et al., 2025; Chen et al., 2025a). It should be noted that the absolute performance of an MLLM is independent of its non-unification score. First, non-unification measures only the relative discrepancy between generation and understanding, rather than an MLLM's absolute performance on each task. Moreover, differences in training configurations, such as data scale and pipeline design, can make comparisons between absolute performance and the relative gap across models unreliable.

**Stronger Understanding and Human Check.** As described in Section 3, we use stronger external models, such as Qwen2.5-VL-72B-Instruct, to evaluate the scores given by the understanding branch in order to identify the source of the internal capability imbalance in MLLMs, i.e., the internal gap. Figure 8 presents the weak generation rates across nine subtasks based on Qwen's judgments, where we observe that most models exhibit more than 50% weak generation on the majority of tasks.

---

[4]Notation: external signals (ES), internal signals (IS) and curriculum learning (CL).

[5]For fair comparison, we generate images for T2I-R1 directly from original prompts, without using the understanding branch for prompt expansion.

| Difficulty | Task | Evaluation Size | Prompt Example | Source |
|---|---|---|---|---|
| Easy | Single Obj. | 80 | `a photo of a cat` | GenEval |
| | Two Obj. | 99 | `a photo of a stop sign and a dog` | |
| | Color Attri. | 100 | `a photo of a red cake and a purple chair` | |
| Medium | Texture | 300 | `fluffy clouds and a glass table` | T2I-CompBench++ |
| | Spatial | 300 | `a cat on the top of a sofa` | |
| | Complex | 300 | `The prickly green cactus contrasted with the smooth white walls.` | |
| Hard | Physics | 118 | `A ice block at sixty degrees Celsius, clear, simple and realistic.` | Science-T2I-S |
| | Chemistry | 49 | `A iron ball that has been exposed to oxygen for decades, simple, clear and realistic.` | |
| | Biology | 60 | `A sweetgum tree in winter with high realism.` | |

Table 3: Subtasks categorized by difficulty level. As shown in Table 3, we select nine subtasks from three benchmarks to construct evaluation data with progressively increasing generation and understanding difficulty. Easy tasks involve only object generation, while Medium prompts require both generation and reasoning over spatial relations, colors, and textures. Hard tasks contain implicit reasoning, requiring MLLMs to possess accurate prior knowledge.

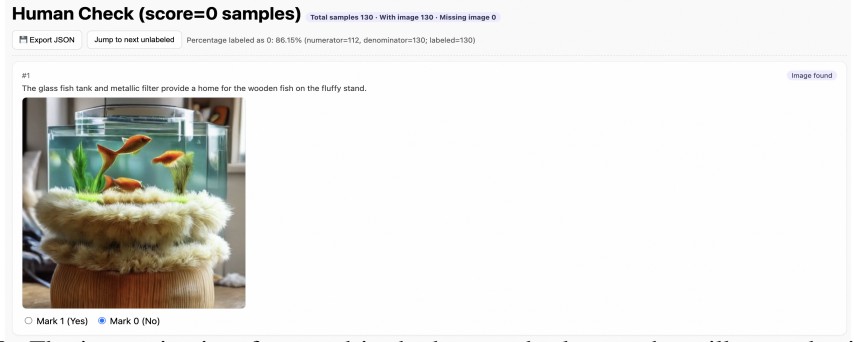

Figure 10: The interactive interface used in the human-check procedure, illustrated using Janus-Pro's evaluation on the texture subtask as an example. Human annotators are asked to determine whether the image–prompt pairs flagged by the understanding branch as misaligned (score = 0) are indeed aligned with the prompt.

It should be noted that a weak generation rate of zero may arise partly from misjudgments of the understanding branch, e.g., Janus-Pro and VILA-U in Biology have nearly zero weak generation, and in other cases, e.g., Show-o, from a non-unification score of zero for that task, which naturally leads to a weak generation rate of zero.

To ensure the accuracy of our evaluation, we further introduce a human check. The full procedure is as follows. (1) Obtain the understanding-branch scores. Following the definition of the non-unification score in Section 3, we first compute the understanding branch's judgment of whether each generated image satisfies its prompt. (2) Select samples predicted as incorrect (score = 0). We then collect all samples for

Figure 11: Human-evaluated weak generation aligns with Qwen-based results, confirming weak generation as the primary cause of non-unification and supporting the use of Qwen as external judges in win rate.

which the understanding branch outputs score = 0 (misaligned image–prompt pairs). (3) Perform human re-evaluation of these score = 0 samples. All selected samples are manually inspected, where

annotators decide whether each image satisfies the corresponding prompt (assigning a score of 0 or 1). The evaluation is conducted by two PhD-level annotators: one performs the initial annotation, and the other verifies it, ensuring accurate understanding of both prompts and images. (4) Compute the human-evaluated weak generation score. We the calculate the human-evaluated weak generation score which measures the probability that humans agree with the understanding branch conditioned on the branch predicting score = 0, i.e.,

$$\text{Human-evaluated Weak Generation} := \mathbb{P}\big(\pi_\theta^{\text{und}}(\mathbf{x},\ q(\mathbf{y})) = S_{\text{human}}(\mathbf{x},\ q(\mathbf{y})) \,\big|\, \pi_\theta^{\text{und}}(\mathbf{x},\ q(\mathbf{y})) = 0\big).$$

where $S_{\text{human}}$ denotes the human score.

Figure 11 further presents weak generation results based on human evaluation, which yield consistent findings: MLLMs achieve weak generation scores exceeding 50% on the majority of tasks, further emphasizing that non-unification primarily stems from weak generation rather than misunderstanding. Moreover, the weak generation scores obtained from human evaluation are closely aligned with those derived from Qwen-based evaluation, with an average score difference of 1.01% for Easy tasks, 8.21% for Medium tasks, and 19.67% for Hard tasks. The relatively larger discrepancy for Hard tasks may indicate that Qwen also faces limitations in understanding images involving implicit reasoning. Nevertheless, the overall agreement between human evaluation and Qwen in assessing MLLMs supports the continued use of Qwen as an external judge in subsequent studies, such as evaluating the win rate for understanding in Section 4.2.1.

# B  ADDITIONAL DETAILS AND FULL RESULTS ON SELF-IMPROVEMENT

## B.1  ADDITIONAL DETAILS

**Data Details.**   We use approximately 6000 prompts from T2I-CompBench++ (Huang et al., 2025) as our training data, where we strictly follow the official data split defined in T2I-CompBench++ to obtain the training and evaluation sets.

**Implementation Details.**   During the construction of SFT and DPO datasets, we feed each input image together with its corresponding question:

> **Question**
>
> You are a helpful language and vision assistant. You are able to understand the visual content that the user provides, and assist the user with a variety of tasks using natural language. Does this original image describe {prompt}? If it describes the scene, score 1; if it does not fully describe, score 0. Please answer in the following format: The score is {your score}.

We record the prediction probability from the understanding branch and select the image with the highest predicted probability of {*your score*} = 1 as the chosen sample, and the image with the highest predicted probability of {*your score*} = 0 as the rejected sample. The chosen images are used both as positive samples for DPO and as SFT samples, whereas the rejected images are used as negative samples for DPO. It is worth noting that, for DPO, we adopt the common practice of applying the negative log-likelihood (NLL) loss (Pang et al., 2024; Dubey et al., 2024) over the preferred response in each pair, in order to enhance DPO. We conduct self-improvement on Janus-Pro-7B and Show-o (option (a) and $512 \times 512$) using four 80 GB NVIDIA A800 GPUs, with self-improvement epochs set to 20 for SFT and 30 for DPO, respectively. Self-improvement requires approximately 7–8 hours. The detailed hyperparameter configurations are presented in Table 4.

**Evaluation.**   In addition to evaluating the self-improved MLLMs on the validation set of T2I-CompBench++, we also conduct evaluations on GenEval and Science-T2I. As introduced in Appendix A.1, GenEval is a relatively simple benchmark focusing on object and its basic attributes, whereas Science-T2I involves more complex prompts that require implicit reasoning. For image generation metrics, we follow the evaluation protocols and metric definitions specified by each

---

[6] Our implementation is based on https://github.com/PKU-Alignment/align-anything.

[7] Our implementation is based on https://github.com/ZiyuGuo99/Image-Generation-CoT.

| Hyperparameter | Janus-Pro-7B [6] | Show-o [7] |
|---|---|---|
| *Optimization* | | |
| Optimizer | Adam | AdamW |
| Learning rate | $1 \times 10^{-7}$ | $1 \times 10^{-5}$ |
| Adam(W) $\beta$ | $[0.9, 0.95]$ | $[0.9, 0.999]$ |
| Weight decay | 0.05 | 0.01 |
| Warmup steps (Ratio) | 0.03 | 0.1 |
| Epoch | 20 (SFT) / 30 (DPO) | 20 (SFT) / 30 (DPO) |
| Grad. accumulation | 1 | 1 |
| Per-GPU batch size | 1 | 1 |
| *Trainable modules* | | |
| Trainable parts | LLM | LLM |
| Full Fine-tuning | ✓ | ✓ |
| *Loss weights* | | |
| DPO $\beta$ | 0.01 | 0.01 |
| Weight NLL | 0.1 | 0.1 |
| CFG Weight | 5 | 5 |
| *Data Construction* | | |
| Image Size | $384 \times 384$ | $512 \times 512$ |
| Images per Prompt | 10 | 10 |
| Data Size | 1326 | 226 |

Table 4: Hyperparameter configurations in self-improvement. For trainable parts, we only consider the LLM components shared by generation and understanding, which are sufficient to promote MLLMs. Additional trainable modules are discussed in Appendix F.2.

| Model | Texture | | | Shape | | | Color | | | Spatial | | | Non-Spatial | | | Complex | | | Overall | | |
|---|---|---|---|---|---|---|---|---|---|---|---|---|---|---|---|---|---|---|---|---|---|
| | Gen.↑ | Und.↑ | Non.↓ | Gen.↑ | Und.↑ | Non.↓ | Gen.↑ | Und.↑ | Non.↓ | Gen.↑ | Und.↑ | Non.↓ | Gen.↑ | Und.↑ | Non.↓ | Gen.↑ | Und.↑ | Non.↓ | Gen.↑ | Und.↑ | Non.↓ |
| *Gen. and Und.* | | | | | | | | | | | | | | | | | | | | | |
| Janus-Pro-7B(Baseline) | 38.63 | 50.00 | 43.33 | 33.49 | 50.00 | 43.00 | 53.22 | 50.00 | 27.33 | 16.81 | 50.00 | 31.00 | 31.40 | 50.00 | 2.33 | 37.73 | 50.00 | 10.33 | 35.21 | 50.00 | 26.22 |
| + SFT | 53.93 | 65.22 | 29.67 | 38.63 | 53.85 | 34.00 | 73.41 | 54.62 | 10.85 | 23.73 | 26.67 | 22.00 | 31.45 | 75.00 | 1.00 | 38.57 | 75.00 | 4.33 | 43.29 | 58.39 | 16.98 |
| + C-SFT | 56.38 | 66.67 | 28.33 | 39.86 | 64.52 | 33.67 | 73.77 | 52.14 | 12.20 | 24.87 | 38.46 | 21.67 | 31.44 | 75.00 | 2.33 | 38.78 | 70.00 | 3.33 | 44.18 | 61.13 | 16.92 |
| + DPO | 40.98 | 53.85 | 43.00 | 33.49 | 57.89 | 47.00 | 51.72 | 63.64 | 27.12 | 16.49 | 41.67 | 30.00 | 31.32 | 66.67 | 2.00 | 38.61 | 50.00 | 6.67 | 35.44 | 55.62 | 25.97 |
| + C-DPO | 42.13 | 53.33 | 45.33 | 33.46 | 55.56 | 40.00 | 53.17 | 55.71 | 28.81 | 15.74 | 42.86 | 32.33 | 31.38 | 50.00 | 2.00 | 37.98 | 78.57 | 6.33 | 35.64 | 56.00 | 25.80 |
| T2I-R1(External) | 50.91 | 52.50 | 34.67 | 37.80 | 53.49 | 36.00 | 70.47 | 35.29 | 11.33 | 24.22 | 45.00 | 23.67 | 31.38 | 75.00 | 1.00 | 38.53 | 72.73 | 3.33 | 42.22 | 55.67 | 18.33 |
| *Gen. and Und.* | | | | | | | | | | | | | | | | | | | | | |
| Show-o(Baseline) | 66.80 | 50.00 | 0.33 | 52.72 | 50.00 | 0.67 | 72.50 | 50.00 | 0.00 | 39.31 | 50.00 | 4.67 | 31.43 | 50.00 | 0.00 | 35.17 | 50.00 | 0.00 | 49.66 | 50.00 | 0.95 |
| + SFT | 73.26 | 50.00 | 0.00 | 59.53 | 100.00 | 0.00 | 72.93 | 50.00 | 0.00 | 42.66 | 100.00 | 0.67 | 31.32 | 50.00 | 0.00 | 36.33 | 50.00 | 0.00 | 52.67 | 66.67 | 0.11 |
| + C-SFT | 74.11 | 50.00 | 0.00 | 59.75 | 100.00 | 0.00 | 72.38 | 50.00 | 0.00 | 42.70 | 100.00 | 0.33 | 31.53 | 50.00 | 0.00 | 36.42 | 50.00 | 0.00 | 52.82 | 66.67 | 0.06 |
| + DPO | 69.97 | 50.00 | 0.33 | 55.45 | 50.00 | 0.00 | 73.67 | 50.00 | 0.34 | 42.59 | 66.67 | 2.00 | 31.61 | 50.00 | 0.00 | 35.71 | 50.00 | 0.00 | 51.50 | 52.78 | 0.45 |
| + C-DPO | 70.32 | 50.00 | 0.00 | 57.32 | 50.00 | 1.00 | 75.39 | 50.00 | 0.00 | 44.55 | 100.00 | 1.33 | 31.52 | 50.00 | 0.00 | 35.47 | 50.00 | 0.00 | 52.43 | 58.33 | 0.39 |
| Hermsflow(External) | 67.96 | 50.00 | 0.33 | 51.81 | 50.00 | 0.33 | 72.96 | 50.00 | 0.34 | 38.45 | 0.00 | 4.00 | 31.42 | 50.00 | 0.00 | 35.28 | 50.00 | 0.00 | 49.65 | 41.67 | 0.83 |

Table 5: Evaluation Results on T2I-CompBench++. Self-improvement enhances MLLMs in generation, understanding, and unification, achieving results comparable to or even surpassing those of baselines that leverage external rewards.

benchmark. In addition, we adopt the definition of unification from Section 3, namely

$$\text{unification} := 1 - \text{non-unification score}.$$

For evaluating understanding capability, we introduce the win rate metric. Specifically, the win rate (excluding ties) is defined as the proportion of samples where the understanding prediction changes after self-improvement and agrees with the score of stronger judge—Qwen2.5-VL-72B-Instruct. We let $\pi_{\text{pre}}$ and $\pi_{\text{self}}$ denote the pre-trained and self-improved MLLMs, respectively. We define generations by pre-trained MLLMs as $\mathbf{x}_{\text{pre}} = \pi_{\text{pre}}^{\text{gen}}(\mathbf{y})$ for the prompt $\mathbf{y}$. Win rate is:

$$\text{Win rate} := \frac{\sum_{\mathbf{y}} \mathbb{I}\left[\pi_{\text{pre}}^{\text{und}}(\mathbf{x}_{\text{pre}}, q(\mathbf{y})) \neq \pi_{\text{self}}^{\text{und}}(\mathbf{x}_{\text{pre}}, q(\mathbf{y})) \wedge \pi_{\text{self}}^{\text{und}}(\mathbf{x}_{\text{pre}}, q(\mathbf{y})) = s_{\text{Qwen}}\right]}{\sum_{\mathbf{y}} \mathbb{I}\left[\pi_{\text{pre}}^{\text{und}}(\mathbf{x}_{\text{pre}}, q(\mathbf{y})) \neq \pi_{\text{self}}^{\text{und}}(\mathbf{x}_{\text{pre}}, q(\mathbf{y}))\right]} \quad (4)$$

where $s_{\text{Qwen}}(\mathbf{x}_{\text{pre}}, q(\mathbf{y})) \in \{0, 1\}$ is oracle label provided by Qwen. We introduce the win rate metric, which enables the simultaneous quantification of generation, understanding, and unification within the same task, thereby providing a better depiction of the synchronous changes between generation and understanding. In addition, we evaluate the understanding performance of MLLMs on dedicated benchmarks and provide illustrative examples for both before and after self-improvement.

| Model | Single Obj. | | | Two Obj. | | | Counting | | | Colors | | | Position | | | Color Attri. | | | Overall | | |
|---|---|---|---|---|---|---|---|---|---|---|---|---|---|---|---|---|---|---|---|---|---|
| | Gen.↑ | Und.↑ | Non.↓ | Gen.↑ | Und.↑ | Non.↓ | Gen.↑ | Und.↑ | Non.↓ | Gen.↑ | Und.↑ | Non.↓ | Gen.↑ | Und.↑ | Non.↓ | Gen.↑ | Und.↑ | Non.↓ | Gen.↑ | Und.↑ | Non.↓ |
| *Gen. and Und.* | | | | | | | | | | | | | | | | | | | | | |
| Janus-Pro-7B$_{(Baseline)}$ | 98.75 | 50.00 | 3.75 | 85.86 | 50.00 | 4.04 | 61.50 | 50.00 | 2.50 | 84.04 | 50.00 | 2.13 | 75.00 | 50.00 | 5.00 | 71.00 | 50.00 | 20.00 | 79.36 | 50.00 | 6.24 |
| + *SFT* | 96.25 | 100.00 | 2.50 | 87.88 | 0.00 | 7.07 | 65.00 | 50.00 | 5.00 | 87.23 | 66.67 | 1.06 | 78.00 | 40.00 | 5.00 | 65.00 | 50.00 | 13.00 | 79.89 | 51.11 | 5.61 |
| + *C-SFT* | 98.75 | 100.00 | 6.25 | 88.89 | 0.00 | 5.05 | 66.25 | 0.00 | 6.25 | 88.30 | 100.00 | 8.51 | 79.00 | 40.00 | 6.00 | 64.00 | 66.67 | 15.00 | 80.87 | 51.11 | 7.84 |
| + *DPO* | 98.75 | 50.00 | 2.50 | 89.90 | 0.00 | 6.06 | 56.25 | 50.00 | 6.25 | 88.30 | 50.00 | 3.19 | 73.00 | 50.00 | 6.00 | 69.00 | 100.00 | 13.00 | 79.20 | 50.00 | 6.17 |
| + *C-DPO* | 97.50 | 100.00 | 4.25 | 85.86 | 0.00 | 5.10 | 60.00 | 50.00 | 5.25 | 88.30 | 100.00 | 1.06 | 82.00 | 50.00 | 1.00 | 69.00 | 43.33 | 20.00 | 80.44 | 57.22 | 6.11 |
| T2I-R1$_{(External)}$ | 98.75 | 50.00 | 7.50 | 86.87 | 0.00 | 8.08 | 58.75 | 0.00 | 7.50 | 87.23 | 100.00 | 1.06 | 83.00 | 60.00 | 5.00 | 70.00 | 50.00 | 22.00 | 80.77 | 43.30 | 8.52 |
| *Gen. and Und.* | | | | | | | | | | | | | | | | | | | | | |
| Show-o$_{(Baseline)}$ | 97.50 | 50.00 | 1.25 | 80.81 | 50.00 | 2.02 | 76.25 | 50.00 | 2.50 | 85.11 | 50.00 | 0.00 | 28.00 | 50.00 | 2.00 | 53.00 | 50.00 | 0.00 | 70.11 | 50.00 | 1.30 |
| + *SFT* | 97.50 | 50.00 | 1.25 | 91.92 | 50.00 | 0.00 | 61.25 | 50.00 | 0.00 | 78.72 | 50.00 | 0.00 | 37.00 | 50.00 | 2.00 | 62.00 | 50.00 | 0.00 | 71.40 | 50.00 | 0.54 |
| + *C-SFT* | 96.25 | 50.00 | 1.25 | 86.87 | 50.00 | 0.00 | 67.50 | 50.00 | 0.00 | 78.72 | 50.00 | 1.06 | 44.00 | 50.00 | 1.00 | 66.00 | 50.00 | 1.00 | 73.22 | 50.00 | 0.72 |
| + *DPO* | 97.25 | 50.00 | 1.25 | 84.85 | 50.00 | 0.00 | 71.25 | 50.00 | 0.00 | 84.04 | 50.00 | 0.00 | 38.00 | 50.00 | 1.00 | 52.00 | 50.00 | 0.00 | 71.23 | 50.00 | 0.38 |
| + *C-DPO* | 97.50 | 50.00 | 1.25 | 84.85 | 50.00 | 0.00 | 70.00 | 50.00 | 0.00 | 86.17 | 50.00 | 0.00 | 37.00 | 50.00 | 1.00 | 59.00 | 50.00 | 0.00 | 72.42 | 50.00 | 0.38 |
| Hermsflow$_{(External)}$ | 96.25 | 50.00 | 1.25 | 83.84 | 50.00 | 1.01 | 66.25 | 50.00 | 1.25 | 80.85 | 50.00 | 1.06 | 35.00 | 50.00 | 2.00 | 46.00 | 50.00 | 0.00 | 68.03 | 50.00 | 1.10 |

Table 6: Evaluation Results on Geneval. Self-improvement enhances MLLMs in generation, understanding, and unification, achieving results comparable to or even surpassing those of baselines that leverage external rewards.

| Model | Physics | | | Chemistry | | | Biology | | | Overall | | |
|---|---|---|---|---|---|---|---|---|---|---|---|---|
| | Gen.↑ | Und.↑ | Non.↓ | Gen.↑ | Und.↑ | Non.↓ | Gen.↑ | Und.↑ | Non.↓ | Gen.↑ | Und.↑ | Non.↓ |
| *Gen. and Und.* | | | | | | | | | | | | |
| Janus-Pro-7B$_{(Baseline)}$ | 25.37 | 50.00 | 3.39 | 25.57 | 50.00 | 2.04 | 22.54 | 50.00 | 5.00 | 24.49 | 50.00 | 3.48 |
| + *SFT* | 25.47 | 33.33 | 3.39 | 26.85 | 100.00 | 0.00 | 22.90 | 75.00 | 3.33 | 25.07 | 69.44 | 2.24 |
| + *C-SFT* | 25.48 | 25.00 | 1.69 | 26.66 | 100.00 | 2.04 | 23.41 | 80.00 | 6.67 | 25.18 | 68.33 | 3.47 |
| + *DPO* | 25.72 | 50.00 | 1.69 | 25.37 | 100.00 | 0.00 | 23.49 | 0.00 | 3.33 | 24.86 | 50.00 | 1.67 |
| + *C-DPO* | 25.72 | 50.00 | 1.39 | 25.44 | 50.00 | 1.16 | 22.76 | 66.67 | 5.00 | 24.64 | 55.56 | 2.52 |
| T2I-R1$_{(External)}$ | 25.52 | 0.00 | 2.54 | 25.28 | 100.00 | 2.04 | 22.64 | 66.67 | 5.00 | 24.48 | 55.56 | 3.19 |
| *Gen. and Und.* | | | | | | | | | | | | |
| Show-o$_{(Baseline)}$ | 25.56 | 50.00 | 5.93 | 26.13 | 50.00 | 0.00 | 22.48 | 50.00 | 0.00 | 24.72 | 50.00 | 1.98 |
| + *SFT* | 26.57 | 60.00 | 1.69 | 26.62 | 50.00 | 0.00 | 22.48 | 50.00 | 0.00 | 25.22 | 53.33 | 0.56 |
| + *C-SFT* | 27.12 | 60.00 | 0.85 | 27.63 | 50.00 | 0.00 | 23.38 | 50.00 | 0.00 | 26.04 | 53.33 | 0.28 |
| + *DPO* | 26.05 | 0.00 | 5.08 | 25.76 | 50.00 | 0.00 | 21.53 | 50.00 | 0.00 | 24.44 | 33.33 | 1.69 |
| + *C-DPO* | 25.93 | 50.00 | 5.93 | 25.71 | 50.00 | 0.00 | 22.51 | 50.00 | 0.00 | 24.72 | 50.00 | 1.98 |
| HermesFlow$_{(External)}$ | 25.61 | 54.00 | 5.46 | 26.47 | 50.00 | 0.00 | 21.91 | 50.00 | 0.00 | 24.66 | 51.33 | 1.82 |

Table 7: Evaluation Results on Science-T2I-S. Self-improvement enhances MLLMs in generation, understanding, and unification, achieving results comparable to or even surpassing those of baselines that leverage external rewards.

## B.2 FULL RESULTS

**Full Results on Self-Improvement.** Table 5, Table 6, and Table 7 report the improvements in generation, understanding, and unification of self-improvemed MLLMs across three benchmarks. Results of self-improvemed MLLMs are comparable to, and even surpass, two baselines, T2I-R1 and Hermesflow, which rely on external rewards. Taking Janus-Pro under SFT as an example, self-improvement boosts its generation and unification performance on T2I-CompBench++ by an average of 8% and 10%, respectively. Moreover, compared to pretrained Janus-Pro, its understanding capability

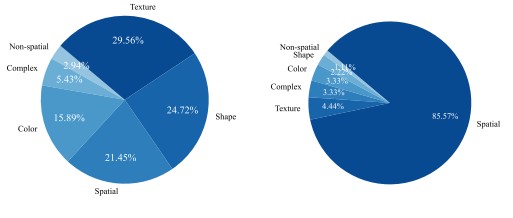

(a) SFT, Janus-Pro-7B  (b) SFT, Show-o

Figure 12: Building self-improvement data based on internal gaps yields more samples from large gap tasks, thus guiding more gains on such tasks.

is enhanced with win rate greater than 50%. Improvement also observed on GenEval and Science-T2I. These experiemnts verify the effectiveness of our proposed approach.

Additionally, MLLMs with larger internal gaps (e.g., Janus-Pro-7B) and larger gap subtasks (e.g., Texture) exhibit greater gains after self-improvement. We claim that this may be because tasks with larger internal gaps encourage more samples from those subtasks in the post-training data, thereby benefiting the learning of those specific subtasks. Figure 12 demonstrates that subtasks with larger internal gaps constitute a higher proportion of the post-training data, which contributes to their greater performance gains, supporting our hypothesis.

**Improved Understanding: Additional Results on Understanding Benchmarks and Examples**  For image understanding evaluation, we consider the benchmarks POPE (Li et al., 2023b), MMB (Liu et al., 2024), SEED (Li et al., 2023a), GQA (Hudson & Manning, 2019), and MMMU, and conduct the evaluation using VLMEvalKit. Since all these benchmarks are in a multiple-choice format, we compute accuracy using exact matching. Table 8 presents the results of the pre-trained Janus-Pro and the self-improved Janus-Pro on various understanding benchmarks, showing that the MLLM's understanding ability is further enhanced after self-improvement, with gains up to 3%. Table 8 also provides the self-improvement results for Show-o. We observe that SFT-based self-improvement not only enhances generation but also improves understanding ability, for example, POPE increases by nearly 2%.

We further present examples of self-improvement for Janus-Pro and Show-o under SFT (Figure 14) and DPO (Figure 15), which clearly demonstrate that after self-improvement, the models not only generate images that better satisfy the prompts, but also more accurately identify misalignments between the original image and the prompt, thereby providing correct evaluation scores (from score 1 to score 0). The improvements observed on understanding benchmarks, together with these concrete examples, further support the co-improvement conclusion in Section 4.2.2: generation-targeted self-improvement can also enhance understanding.

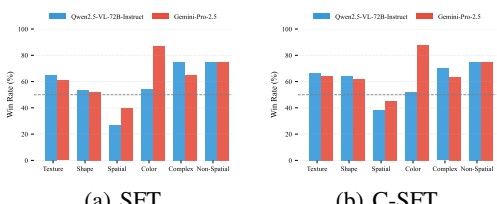

(a) SFT                    (b) C-SFT

Figure 13: Introducing an additional evaluator Gemini-Pro-2.5. Under Gemini's evaluation, self-improved model demonstrates better understanding ability with win rate greater than 0.5.

| Model | POPE↑ | MMB ↑ | SEED ↑ | GQA↑ | MMMU ↑ |
|---|---|---|---|---|---|
| Janus-Pro-7B | 89.04 | 76.23 | 70.09 | 56.02 | 32.86 |
| + *SFT* | 88.45 | 76.97 | 70.44 | 56.12 | **35.24** |
| + *DPO* | 89.06 | 76.41 | 70.10 | **56.26** | 33.71 |
| + *C-SFT* | 89.03 | **77.18** | 70.48 | 56.02 | **35.24** |
| + *C-DPO* | **89.10** | 76.47 | **70.86** | 56.17 | 34.33 |
| Show-o | 64.05 | 30.91 | 52.86 | 56.82 | 23.33 |
| + *SFT* | 65.27 | 31.92 | 54.14 | 57.22 | **24.00** |
| + *DPO* | 64.71 | 30.82 | 52.73 | 57.03 | 23.33 |
| + *C-SFT* | **65.82** | **32.34** | **54.32** | **57.33** | 23.33 |
| + *C-DPO* | 64.97 | 31.14 | 52.90 | 57.09 | 23.33 |

Table 8: The self-improved MLLMs demonstrated improvements on understanding benchmarks.

**Improved Understanding: Additional Results on External Evaluator**  In Section 6, we use Qwen2.5-VL-72B-Instruct as the external evaluator. To further validate our findings regarding changes in the model's understanding capability, we replace Qwen with the closed-source Gemini-Pro-2.5 (Team, 2025b) to compute the win rate defined in Equation (4). Using Janus-Pro's SFT and C-SFT results as an illustrative example, Figure 13 shows the outcomes obtained with Gemini as the external evaluator. In most cases, the win rate exceeds 50%, indicating that Janus-Pro's understanding ability improves after self-improvement, as reflected by its scores becoming closer to those of Gemini-Pro-2.5.

## C  ADDITIONAL DETAILS AND FULL RESULTS ON CURRICULUM-LEARNING-BASED SELF-IMPROVEMENT

In this section, we present the training details and full experimental results of the curriculum learning–based self-improvement method.

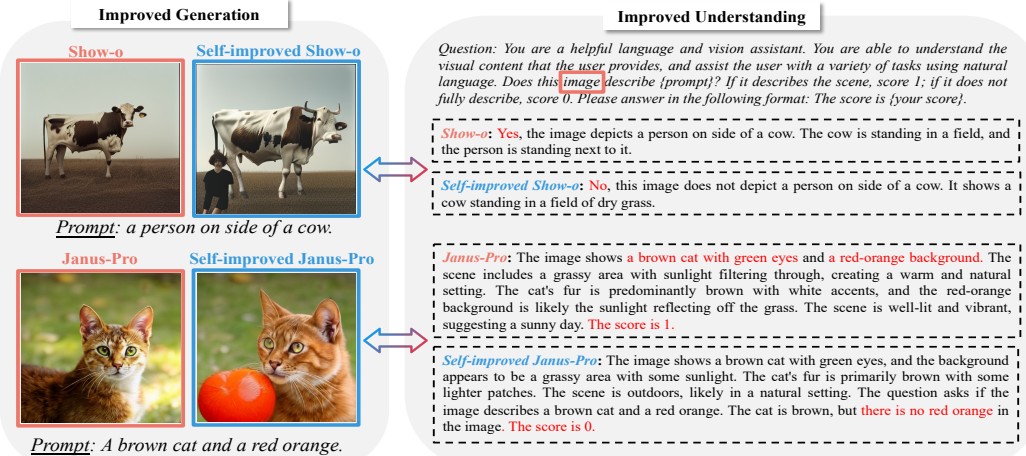

Figure 14: Examples of co-improvements in generation and understanding of self-improved Janus-Pro and Show-o under SFT. We observe that, after self-improvement, Show-o and Janus-Pro generate images that align prompts and accurately identify when images produced by the pre-trained MLLM are misaligned with the prompts.

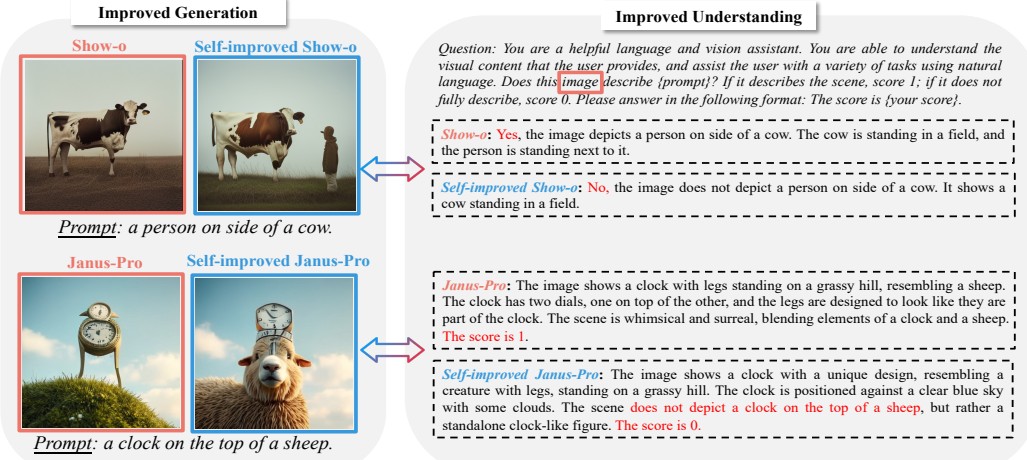

Figure 15: Examples of co-improvements in generation and understanding of self-improved Janus-Pro and Show-o with DPO. We observe that, after self-improvement, Show-o and Janus-Pro generate images that align prompts and accurately identify when images produced by the pre-trained MLLM are misaligned with the prompts.

## C.1  Additional Details

**Implementation Details.** Following Section 6, we leverage the improved generation–understanding model to revisit prompts that were not utilized by the pre-trained MLLM due to weak generation or weak understanding capabilities (see details in Alg 2). This process can be regarded as a form of curriculum learning based on prompt complexity (Li & Zhang, 2025). We follow the training configurations in Table 4 and perform curriculum replay for both SFT-based and DPO-based self-improvement at epoch 10. In Appendix F.4, we conduct an ablation study to discuss the choice of epoch for curriculum learning. Table 9 shows the data expansion for Janus-Pro and Show-o with curriculum learning, which increases sample size by up to 50%.

| MLLM | Self-improvement Strategy | Curriculum Epoch | Original Data | Expansion Data |
|---|---|---|---|---|
| Janus-Pro-7B | SFT | 10 | 2265 | +1091 |
| | DPO | 10 | | +359 |
| Show-o | SFT | 10 | 226 | +64 |
| | DPO | 10 | | +59 |

Table 9: Expansion of post-training data with introducing curriculum learning.

**Evaluation.** Consistent with the evaluation in Appendix B.1, we employ the same metrics to measure MLLMs in generation, understanding and unification.

## C.2 FULL RESULTS

As shown in Table 5, Table 6, and Table 7, the self-improvement with curriculum learning (denoted as *C-SFT* and *C-DPO*) demonstrates that the increased post-training data benefiting from curriculum learning further enhances self-improvement MLLMs' performance and unification, particularly in understanding and generation.

## D  UNDERSTANDING CO-IMPROVEMENT IN SELF-IMPROVEMENT

Section 5.2 explains why co-improvement occurs when self-improvement is performed with SFT and provides empirical evidence based on Janus-Pro. In this section, we first detail the computation of the empirical evidence in Figure 7, then additionally present empirical evidence on Show-o with SFT to further support the theoretical analysis in Section 5.1.

### D.1  FULL THEORETICAL ANALYSIS UNDER SFT

**Details on Empirical Evidence.** Figure 7(a) explains that samples from the false positive correction group $(\mathbf{y}_0, \mathbf{x}_0)$, i.e., the primary source of improvement in comprehension capability, exhibit higher similarity to their corresponding post-training samples $(\mathbf{y}_u, \mathbf{x}_u)$. Specifically, we separately compute text similarity and image similarity as proxies for eNTK term $\mathcal{K}$: for each $\mathbf{y}_0$, we first identify its nearest neighbor $\mathbf{y}_u$ in the post-training data, then compute the similarity between the corresponding images $\mathbf{x}_0$ and $\mathbf{x}_u$ . For text similarity, we use pre-trained model all-MiniLM-L6-v2 [8] to encode each prompt into a 384-dimensional vector and compute the cosine similarity between vector pairs. For image similarity, we use an equal-weighted combination of MSE and SSIM (Wang et al., 2004) to measure both pixel-level and structural similarity. To evaluate whether false positive correction group indeed exhibits higher similarity, we randomly sample *random group* $(\mathbf{y}, \mathbf{x})$ (with the same size as false positive correction group) and calculate same. Figure 7(a) shows false positive correction group demonstrates significantly higher similarity in $(\mathbf{y}_u, \mathbf{x}_u)$, particularly in text. For Figure 7(a), the difference in text similarity between the False Positive Correction group and the random group becomes more pronounced: the prompt similarity between the False Positive Correction group (0,1,0) and the training data has a mean of approximately 0.85, whereas the random group has a much lower mean of around 0.65, indicating a clear distinction. The reason we focus more on prompt similarity is that the similarity shown Figure 7 is in fact a proxy for eNTK $\mathcal{K}_{k,r}^t(\mathcal{Y}_0, \mathcal{Y}_u)$ in Proposition 1, where $\mathcal{Y} := [y \mid x]$ is formed by concatenating the token embeddings of the prompt $y$ and the image $x$ (see Appendix E for details). Since Proposition 1 shows that the learning dynamics for both understanding and generation accumulate token by token (see Proposition 1), the prompt tokens, which appear at the beginning of the concatenated sequence, tend to contribute more substantially to the resulting eNTK than the later image tokens. Therefore, variations in prompt similarity can provide a more sensitive and reliable indicator of targeted eNTK.

Figure 7(b) shows the Frobenius norm $\left\|\mathcal{K}^t(\mathcal{Y}_0, \mathcal{Y}_u)\right\|_F$ exceeds $\left\|\mathcal{K}^t(\mathcal{Y}_i, \mathcal{Y}_u)\right\|_F$. This indicates that at iteration $t$, the training dynamics of the understanding branch are primarily driven by $\mathcal{K}^t(\mathcal{Y}_0, \mathcal{Y}_u)$, which tends to align $\Delta U_t$ in Equation (3) and $\Delta G_t$ in Equation (2). To substantiate this, we use data similarity as a proxy for the eNTK. Specifically, for each sample $(\mathbf{y}_0, \mathbf{x}_0)$ in the false positive correction group, we first identify its closest $(\mathbf{y}_u, \mathbf{x}_u)$ based on the most similar prompt and compute text and image similarities using the same metrics as in Figure 7(a); this serves as the proxy for $\left\|\mathcal{K}^t(\mathcal{Y}_0, \mathcal{Y}_u)\right\|_F$. For $\left\|\mathcal{K}^t(\mathcal{Y}_i, \mathcal{Y}_u)\right\|_F$, we compute the text and image similarity between each non-$(\mathbf{y}_0, \mathbf{x}_0)$ sample $(\mathbf{y}_i, \mathbf{x}_i)$ and $(\mathbf{y}_u, \mathbf{x}_u)$, and average these similarities over all $(\mathbf{y}_i, \mathbf{x}_i)$ as the proxy.

Figure 7(c) shows, for samples in the false positive correction group, the probability of prompt-misaligned generation, i.e., the prompt-misaligned probability $\pi_\theta(\mathbf{x}_0 \mid \mathbf{y}_0)$ decreases. To quantify this change, for each validation prompt $\mathbf{y}_0$, we first use the pre-trained MLLM to generate $\mathbf{x}_0$ and record its image token sequence and the log-probability of that sequence as $\log \pi_{\theta_0}(\mathbf{x}_0 \mid \mathbf{y}_0)$. We then use the self-improved MLLMs to re-evaluate the conditional log-probability of the same token

---

[8]https://huggingface.co/sentence-transformers/all-MiniLM-L6-v2

sequence, obtaining $\log \pi_{\theta_t}(\mathbf{x}_0 \mid \mathbf{y}_0)$. Following the definition of the generation-branch learning dynamics in Section 5.1, we compute $\Delta G_t = \log \pi_{\theta_t}(\mathbf{x}_0 \mid \mathbf{y}_0) - \log \pi_{\theta_0}(\mathbf{x}_0 \mid \mathbf{y}_0)$.

**More Empirical Evidence on Show-o.** Section 5.2 explains why co-improvement occurs when post-self-improvement is performed with SFT and provides empirical evidence based on Janus-Pro. In this section, we additionally present empirical evidence on Show-o with SFT to further support the theorical analysis in Section 5.1. Figure 16(a) shows that, for Show-o under supervised fine-tuning (SFT), the primary gains in understanding still come from false positive correction, i.e., (Label, Pre-trained, Self-improved) = (0, 1, 0). Moreover, there exists post-training data similar to the false positive correction group, with an average cosine similarity of 0.8 (see Figure 16(b)). Figure 16(c) indicates that the high sample similarity makes $\left\|\mathcal{K}^t(\mathcal{Y}_0, \mathcal{Y}_u)\right\|_F$ the dominant term, encouraging alignment between the training dynamics of generation and understanding. Together with Figure 16(d), which shows $\Delta G_t < 0$, this suggests $\Delta U_t < 0$, meaning the model identifies false positives and achieves joint improvement. Empirical evidence for Show-o under SFT further corroborates the theoretical explanation in Section 5.2.

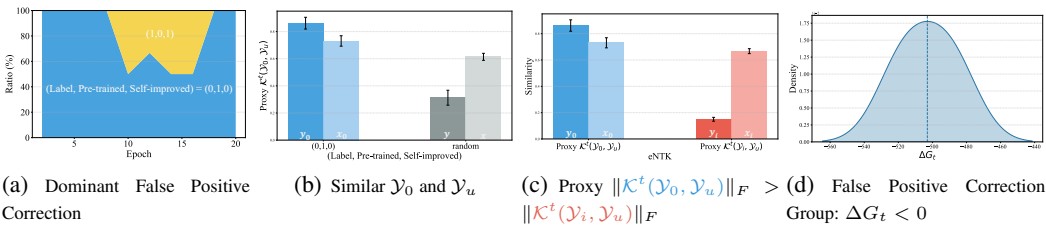

(a) Dominant False Positive Correction

(b) Similar $\mathcal{Y}_0$ and $\mathcal{Y}_u$

(c) Proxy $\|\mathcal{K}^t(\mathcal{Y}_0, \mathcal{Y}_u)\|_F > \|\mathcal{K}^t(\mathcal{Y}_i, \mathcal{Y}_u)\|_F$

(d) False Positive Correction Group: $\Delta G_t < 0$

Figure 16: Empirical Evidence from Self-Improvement with Show-o and SFT. (a) On T2I-CompBench++, understanding gains primarily arise from the false positive correction group. (b) Compared to random samples, those in the false positive correction group are more likely to be matched with highly similar post-training pairs $(\mathbf{y}_u, \mathbf{x}_u)$ (average cosine similarity 0.8). (c) Such high similarity makes $\mathcal{K}^t_{k,r}(\mathcal{Y}_0, \mathcal{Y}_u)$ be the dominant term in Equation (3), thereby promoting aligned learning dynamics between understanding in Equation (3) and generation in Equation (2). (d) With aligned dynamics, $\Delta G_t < 0$ implies $\Delta U_t < 0$: both the probability of misaligned generation $\pi_\theta(\mathbf{x}_0 \mid \mathbf{y}_0)$ and misjudging $\pi_\theta(\mathbf{y}_0 \mid \mathbf{x}_0)$, are reduced. This manifests as false positive correction and jointly as co-improvement.

## D.2 FULL THEORETICAL ANALYSIS UNDER DPO

**Proposition.** In this section, we discuss why DPO-based self-improvement also leads to co-improvement (see Table 5, Table 6, Table 7, and Table 8). For DPO, we define a post-training preference pair $(\mathbf{y}_u, \mathbf{x}_u^+, \mathbf{x}_u^-)$ where the chosen image $\mathbf{x}_u^+$ and the rejected image $\mathbf{x}_u^-$ share the same prompt $\mathbf{y}_u$. The DPO loss is

$$\mathcal{L}_{\text{DPO}}(\mathbf{y}_u, \mathbf{x}_u^+, \mathbf{x}_u^-) = -\mathbb{E}_{(\mathbf{y}_u, \mathbf{x}_u^+, \mathbf{x}_u^-)}\left[\log \sigma\left(\beta \log \frac{\pi_\theta(\mathbf{x}_u^+ \mid \mathcal{Y}_u^+)}{\pi_{\text{ref}}(\mathbf{x}_u^+ \mid \mathcal{Y}_u^+)} - \beta \log \frac{\pi_\theta(\mathbf{x}_u^- \mid \mathcal{Y}_u^-)}{\pi_{\text{ref}}(\mathbf{x}_u^- \mid \mathcal{Y}_u^-)}\right)\right],$$
(5)

where $\mathcal{Y}_u^+$ denotes the concatenation obtained by appending the embedding of $\mathbf{y}_u$ to the embedding of $\mathbf{x}_u^+$, and $\mathcal{Y}_u^-$ denotes the concatenation obtained by appending the embedding of $\mathbf{y}_u$ to the embedding of $\mathbf{x}_u^-$. Then, we have the following proposition:

**Proposition 2** (Learning Dynamics of Generation and Understanding under DPO)**.** *Consider self-improvement proposed in Section 4 with DPO.*

*At epoch $t$, the one-step learning dynamics of **generation** is*

$$\Delta G_t(\mathbf{x}_0 \mid \mathcal{Y}_0)$$
$$= -\eta\beta\sigma(-\alpha)\sum_{k=1}^{M}\sum_{r=1}^{M}(\mathbf{e}_{x_{0,k}} - \pi_k^0)^\top\left[\mathcal{K}^t_{k,r}(\mathcal{Y}_0, \mathcal{Y}_u^+)(\pi_r^{u,+} - \mathbf{e}_{x_{u,r}^+}) - \mathcal{K}^t_{k,r}(\mathcal{Y}_0, \mathcal{Y}_u^-)(\pi_r^{u,-} - \mathbf{e}_{x_{u,r}^-})\right]$$
$$+ \mathcal{O}(\eta^2),$$
(6)

*where the margin* $\alpha := \beta \log \frac{\pi_\theta(\mathbf{x}_u^+|\mathcal{Y}_u^+)}{\pi_{\mathrm{ref}}(\mathbf{x}_u^+|\mathcal{Y}_u^+)} - \beta \log \frac{\pi_\theta(\mathbf{x}_u^-|\mathcal{Y}_u^-)}{\pi_{\mathrm{ref}}(\mathbf{x}_u^-|\mathcal{Y}_u^-)}$ *and* $\pi_r^{u,+} = \mathrm{softmax}(\mathbf{z}_r^{u,+})$ *and* $\mathbf{z}_r^{u,+} = [h_\theta(\mathcal{Y}_u^+)]_r$ *are the logits at image position* $r$ *obtained by running* $h_\theta$ *on* $\mathcal{Y}_u^+$. *The neural tangent kernel* $\mathcal{K}_{k,r}^t(\mathcal{Y}_0, \mathcal{Y}_u^+) := \nabla_\theta \mathbf{z}_k^0 (\nabla_\theta \mathbf{z}_r^{u,+})^\top$ *and* $\mathcal{K}_{k,r}^t(\mathcal{Y}_0, \mathcal{Y}_u^-) := \nabla_\theta \mathbf{z}_k^0 (\nabla_\theta \mathbf{z}_r^{u,-})^\top$.

*The one-step learning dynamics of **understanding** is*

$$\Delta U_t(\mathbf{y}_0 \mid \mathcal{X}_0)$$
$$= -\eta\beta\sigma(-\alpha) \sum_{k=1}^{M} \sum_{r=1}^{M} \sum_{\mathbf{y}_i \neq \mathbf{y}_0} w_{\theta_t}(\mathbf{y}_i \mid \mathbf{x}_0) \left( (\mathbf{e}_{x_{0,k}} - \pi_k^0)^\top \underbrace{\left( \mathcal{K}_{k,r}^t(\mathcal{Y}_0, \mathcal{Y}_u^+)(\pi_r^{u,+} - \mathbf{e}_{x_{u,r}^+}) - \mathcal{K}_{k,r}^t(\mathcal{Y}_0, \mathcal{Y}_u^-)(\pi_r^{u,-} - \mathbf{e}_{x_{u,r}^-}) \right)}_{\text{Term I}} \right.$$
$$\left. - (\mathbf{e}_{x_{0,k}} - \pi_k^i)^\top \underbrace{\left( \mathcal{K}_{k,r}^t(\mathcal{Y}_i, \mathcal{Y}_u^+)(\pi_r^{u,+} - \mathbf{e}_{x_{u,r}^+}) - \mathcal{K}_{k,r}^t(\mathcal{Y}_i, \mathcal{Y}_u^-)(\pi_r^{u,-} - \mathbf{e}_{x_{u,r}^-}) \right)}_{\text{Term II}} \right) + \mathcal{O}(\eta^2)$$
$$(7)$$

*where* $\mathcal{Y}_i$ *denote the concatenation obtained by appending the embedding of* $\mathbf{y}_i$ *to* $\mathbf{U}_0$.

We can interpret Proposition 2 by analogy with Proposition 1. Specifically, when $\mathcal{Y}_0$ is more similar to the post-training data $\mathcal{Y}_u$ than any other $\mathcal{Y}_i$, that is, the Frobenius norm of Term I exceeds that of Term II, both the generation and understanding branches are dominated by the same alignment Term I, yielding consistent update signs.

**Theoretical Analysis with Empirical Evidence.** First, Figure 17(a)(b) show that under DPO, gains in understanding still primarily come from correcting false positives: across training steps, this accounts for roughly 60%–100% of the gains. Hence, we focus on $\mathbf{y}_0$ and its misaligned image $\mathbf{x}_0$ generated by pre-trained MLLMs.

For self-improved Janus-Pro with DPO, by Proposition 2, co-improvement can arise when the post-training data include pairs $(\mathbf{y}_u, \mathbf{x}_u)$ whose prompt $\mathbf{y}_u$ is more similar to $\mathbf{y}_0$ than any other prompt $\mathbf{y}_i$ (empirical evidence in Figure 18(a)(c) and Figure 19(a)(c)). In this case, the understanding update $\Delta U_t$ in Equation (7) is dominated by Term I rather than Term II (empirical evidence in Figure 18(b)(d) and Figure 19(b)(d)). Note that, because $\mathcal{K}_{k,r}^t(\mathcal{Y}_0, \mathcal{Y}_u^+)$ and $\mathcal{K}_{k,r}^t(\mathcal{Y}_0, \mathcal{Y}_u^-)$ share the same prompt $\mathbf{y}_u$, their Frobenius norms are both large, reflecting the high similarity between $\mathbf{y}_0$ and $\mathbf{y}_u$ (empirical evidence in Figure 18(a)(c) and Figure 19(a)(c)). By contrast, $\mathcal{K}_{k,r}^t(\mathcal{Y}_i, \mathcal{Y}_u^+)$ and $\mathcal{K}_{k,r}^t(\mathcal{Y}_i, \mathcal{Y}_u^-)$ are significantly smaller due to the lower similarity between $\mathbf{y}_i$ and $\mathbf{y}_u$ (also in Figure 18(b)(d) and Figure 19(b)(d)). The same Term I therefore aligns the learning dynamics of generation (Equation (6)) and understanding (Equation (7)), yielding consistent update signs $\Delta G_t$ and $\Delta U_t$.

Moreover, such similar post-training pairs $(\mathbf{y}_u, \mathbf{x}_u)$ improve generation by lowering the probability of misaligned outputs, $\pi_\theta(\mathbf{x}_0 \mid \mathbf{y}_0)$, leading to $\Delta G_t < 0$ (empirical evidence in Figure 17(c)(d)). Due to the aligned dynamics, $\Delta U_t < 0$ as well, meaning the probability of misjudging, $\pi_\theta(\mathbf{y}_0 \mid \mathbf{x}_0)$, is reduced. Consequently, false positive correction emerges, manifesting as co-improvement.

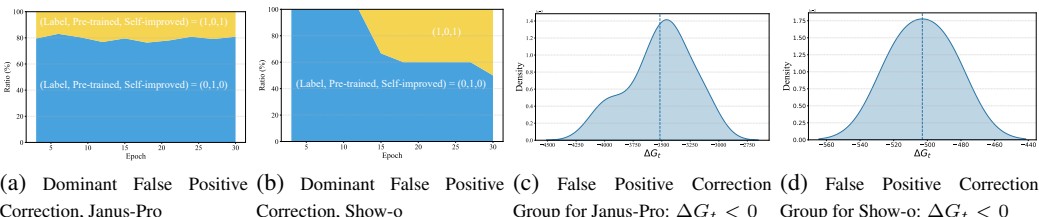

(a) Dominant False Positive Correction, Janus-Pro    (b) Dominant False Positive Correction, Show-o    (c) False Positive Correction Group for Janus-Pro: $\Delta G_t < 0$    (d) False Positive Correction Group for Show-o: $\Delta G_t < 0$

Figure 17: Empirical Evidence from DPO-based Self-Improvement with Janus-Pro and Show-o. (a)(b) On T2I-CompBench++, understanding gains primarily arise from the false positive correction group. (c)(d) For prompts $\mathbf{y}_0$ in the false positive correction group, the self-improved MLLM also reduces the probability of generating the prompt-misaligned image $\mathbf{x}_0$, i.e., $\Delta G_t < 0$.

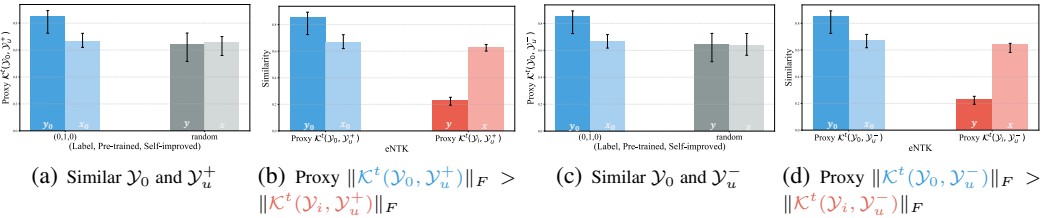

(a) Similar $\mathcal{Y}_0$ and $\mathcal{Y}_u^+$     (b) Proxy $\|\mathcal{K}^t(\mathcal{Y}_0, \mathcal{Y}_u^+)\|_F > \|\mathcal{K}^t(\mathcal{Y}_i, \mathcal{Y}_u^+)\|_F$     (c) Similar $\mathcal{Y}_0$ and $\mathcal{Y}_u^-$     (d) Proxy $\|\mathcal{K}^t(\mathcal{Y}_0, \mathcal{Y}_u^-)\|_F > \|\mathcal{K}^t(\mathcal{Y}_i, \mathcal{Y}_u^-)\|_F$

Figure 18: Empirical Evidence from Self-Improvement with Janus-Pro and DPO. (a)(c) Compared to random samples, those in the false positive correction group are more likely to be matched with highly similar post-training pairs $(\mathbf{y}_u, \mathbf{x}_u)$ (average cosine similarity 0.8). (b)(d) Such high similarity makes Term I be the dominant term in Equation (7), thereby promoting aligned learning dynamics between understanding in Equation (7) and generation in Equation (6).

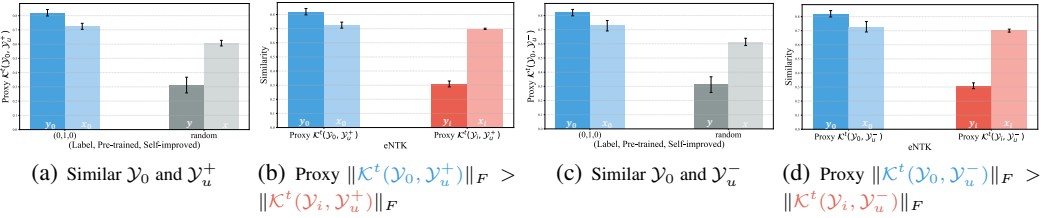

(a) Similar $\mathcal{Y}_0$ and $\mathcal{Y}_u^+$     (b) Proxy $\|\mathcal{K}^t(\mathcal{Y}_0, \mathcal{Y}_u^+)\|_F > \|\mathcal{K}^t(\mathcal{Y}_i, \mathcal{Y}_u^+)\|_F$     (c) Similar $\mathcal{Y}_0$ and $\mathcal{Y}_u^-$     (d) Proxy $\|\mathcal{K}^t(\mathcal{Y}_0, \mathcal{Y}_u^-)\|_F > \|\mathcal{K}^t(\mathcal{Y}_i, \mathcal{Y}_u^-)\|_F$

Figure 19: Empirical Evidence from Self-Improvement with Show-o and DPO. (a)(c) Compared to random samples, those in the false positive correction group are more likely to be matched with highly similar post-training pairs $(\mathbf{y}_u, \mathbf{x}_u)$ (average cosine similarity 0.8). (b)(d) Such high similarity makes Term I be the dominant term in Equation (7), thereby promoting aligned learning dynamics between understanding in Equation (7) and generation in Equation (6).

# E   DERIVATIONS AND PROOF DETAILS

**Preliminaries.** We define the unified vocabulary $\mathcal{V}$ of discrete text and image tokens, with size $V = |\mathcal{V}|$. Since fine-tuning only updates the LLM part $\pi_\theta$ of the MLLM, we work directly in the LLM input space. Let $d$ denote the input embedding dimension.

We consider the setting where both image generation and image understanding *share the same tokenizer* as the default Show-o and EMU3. This contrasts with decoupled designs such as Janus-Pro, where generation and understanding use separate tokenizers. Nevertheless, our analysis shows that results derived under the shared-tokenizer assumption continue to hold for decoupled architectures like Janus-Pro. Specifically, at inference time, for each sequence of image token IDs $\mathbf{x}_0 = (x_{0,1}, \ldots, x_{0,M})$ and text token IDs $\mathbf{y}_0 = (y_{0,1}, \ldots, y_{0,L})$, we encode them as sequences of embeddings as the inputs of LLM. The image sequence is represented by embeddings

$$\mathbf{U}_0 = [\, \mathbf{u}_{0,1} \cdots \mathbf{u}_{0,M} \,] \in \mathbb{R}^{d \times M},$$

and the evaluation prompt is represented by

$$\mathbf{V}_0 = [\, \mathbf{v}_{0,1} \cdots \mathbf{v}_{0,L} \,] \in \mathbb{R}^{d \times L}.$$

where usually $|\mathcal{V}| \gg \max(L, M)$. Similarly, the fine-tuning data pair $(\mathbf{u}_u, \mathbf{v}_u)$ yields $\mathbf{U}_u \in \mathbb{R}^{d \times M}$ and $\mathbf{V}_u \in \mathbb{R}^{d \times L}$.[9]

We consider the typical causal-masking mechanism applied in MLLMs (Wu et al., 2024a; Wang et al., 2024; Wu et al., 2025b). Under this mechanism, $\pi_\theta$ takes the full concatenation of image and text embeddings as input and predicts the next token(s) (Ren & Sutherland, 2025). We denote the concatenated inputs by

$$\mathcal{X}_0 = [\, \mathbf{U}_0 \mid \mathbf{V}_0 \,] \in \mathbb{R}^{d \times (M+L)} \quad \text{(Understanding)},$$

$$\mathcal{Y}_0 = [\, \mathbf{V}_0 \mid \mathbf{U}_0 \,] \in \mathbb{R}^{d \times (L+M)} \quad \text{(Generation)}.$$

---

[9] Across datapoints, the image length $M$ is fixed while the text length $L$ may vary; we use a common symbol $L$ for simplicity.

| Symbol | Definition |
|---|---|
| *Data-related notation* | |
| $\mathbf{y}_0 = (y_{0,1}, \ldots, y_{0,L})$ | Tokenized text prompt (index form), length $L$ |
| $\mathbf{x}_0 = (x_{0,1}, \ldots, x_{0,M})$ | Tokenized image (index form), length $M$ |
| $\mathbf{U}_0 = [\, \mathbf{u}_{0,1} \cdots \mathbf{u}_{0,M} \,]$ | Image token embedding matrix |
| $\mathbf{V}_0 = [\, \mathbf{v}_{0,1} \cdots \mathbf{v}_{0,L} \,]$ | Text token embedding matrix |
| $\mathcal{X}_0 = [\, \mathbf{U}_0 \mid \mathbf{V}_0 \,]$ | Input to understanding branch |
| $\mathcal{Y}_0 = [\, \mathbf{V}_0 \mid \mathbf{U}_0 \,]$ | Input to generation branch |
| $\mathcal{X}_u = [\mathbf{U}_u \mid \mathbf{V}_u]$ | Post-training sample (image) for SFT/DPO updates |
| $\mathcal{Y}_u = [\mathbf{V}_u \mid \mathbf{U}_u]$ | Post-training sample (prompt) for SFT/DPO updates |
| *Model-related notation* | |
| $\pi_\theta$ | Unified MLLM parameterized by $\theta$ |
| $h_\theta(\cdot)$ | Logit network producing token-wise logits before softmax |
| $V$ | Unified vocabulary size for text and image tokens |
| $\mathbf{z}_k^t(S)$ | Logits at position $k$ on sequence $S$ at epoch $t$ |
| $\pi_k^t = \text{softmax}(z_k^t)$ | Token distribution at position $k$ |
| $\mathbf{e}_x$ | One-hot vector corresponding to token $x$ |
| $\pi_\theta(\mathbf{x}_0 \mid \mathcal{Y}_0)$ | Generation likelihood of image tokens |
| $\pi_\theta(\mathbf{y}_0 \mid \mathcal{X}_0)$ | Understanding likelihood of text tokens |
| *Learning dynamic-related notation* | |
| $\Delta G_t(\mathbf{x}_0 \mid \mathcal{Y}_0)$ | One-step update of generation log-likelihood |
| $\Delta U_t(\mathbf{y}_0 \mid \mathcal{X}_0)$ | One-step update of understanding log-likelihood |
| $\mathcal{K}_{k,r}^t(\mathcal{Y}_0, \mathcal{Y}_u)$ | Empirical NTK: $(\nabla_{\theta_t} \mathbf{z}_k^0)(\nabla_{\theta_t} \mathbf{z}_r^u)^\top$ |

Table 10: Key Notations used in the learning dynamics analysis of unified MLLM.

where we omit potential special tokens (e.g., `[SOI]`) for simplicity.

Let $h_\theta$ denote the logits network with causal mask implemented. For understanding,

$$\mathbf{z}_{\text{und}}^0 := h_\theta(\mathcal{X}_0)_{[:, M+1:M+L]} \in \mathbb{R}^{V \times L}, \qquad \Pi_{\text{und}} := \text{softmax}_{\text{col}}(\mathbf{z}_{\text{und}}^0) \in \mathbb{R}^{V \times L},$$

and for generation[10],

$$\mathbf{z}_{\text{gen}}^0 := h_\theta(\mathcal{Y}_0)_{[:, L+1:L+M]} \in \mathbb{R}^{V \times M}, \qquad \Pi_{\text{gen}} := \text{softmax}_{\text{col}}(\mathbf{z}_{\text{gen}}^0) \in \mathbb{R}^{V \times M}.$$

Let $y_{0,l} \in \mathcal{V}$ and $x_{0,k} \in \mathcal{V}$ denote the scalar ground-truth token ids at text position $l$ and image position $k$, respectively. Then the modeling of understanding and generation can be factorized as

$$\log \pi_\theta(\mathbf{y}_0 \mid \mathcal{X}_0) = \sum_l \log \pi_\theta(y_{0,l} \mid \mathbf{x}_0, \mathbf{y}_{0,<l}) = \sum_{l=1}^{L} \log \left[ \Pi_{\text{und}} \right]_{y_{0,l}, l},$$

$$\log \pi_\theta(\mathbf{x}_0 \mid \mathcal{Y}_0) = \sum_k \log \pi_\theta(x_{0,k} \mid \mathbf{y}_0, \mathbf{x}_{0,<k}) = \sum_{k=1}^{M} \log \left[ \Pi_{\text{gen}} \right]_{x_{0,k}, k}.$$

At epoch $t$, we define the *one-step learning dynamics* of evaluation data pair $(\mathbf{x}_0, \mathbf{y}_0)$ after training one-step on fine-tuning data $(\mathbf{x}_u, \mathbf{y}_u)$ as

$$\Delta G_t(\mathbf{x}_0 \mid \mathcal{Y}_0) := \log \pi_{\theta_{t+1}}(\mathbf{x}_0 \mid \mathcal{Y}_0) - \log \pi_{\theta_t}(\mathbf{x}_0 \mid \mathcal{Y}_0) \qquad \text{(Generation)}$$

$$\Delta U_t(\mathbf{y}_0 \mid \mathcal{X}_0) := \log \pi_{\theta_{t+1}}(\mathbf{y}_0 \mid \mathcal{X}_0) - \log \pi_{\theta_t}(\mathbf{y}_0 \mid \mathcal{X}_0) \qquad \text{(Understanding)}$$

---

[10]Typically, generation branch includes a projector as generation head. For example, Janus-Pro uses a 2-layer MLP to map LLM outputs to generation tokenizer's codebook. In our setting, the generation head is frozen.

It is worth noting that Section 4.2.2 evaluates understanding improvement in terms of binary classification $f_\theta(\cdot)$ whereas the theory focuses on log-likelihood $\log \pi_\theta(\mathbf{y}_0 \mid \mathcal{X}_0)$. We introduce a decision rule to bridge the continuous log-likelihood with the discrete binary score:

$$f_\theta(\mathbf{y}_0 \mid \mathcal{X}_0) = \mathbf{1}\{\pi_\theta(\mathbf{y}_0 \mid \mathcal{X}_0) > \tau\},$$

where $\tau$ is a threshold. When $\Delta U_t(\mathbf{y}_0 \mid \mathcal{X}_0)$ increases, the understanding branch is encouraged to raise the log-likelihood, making it more likely to yield a score of 1 under the decision rule.

We first show the connection between the learning dynamics of generation and understanding. First, we obtain

$$\pi_\theta(\mathbf{x}_0 \mid \mathcal{Y}_0) = \prod_{k=1}^{M} \pi_\theta(x_{0,k} \mid \mathbf{y}_0, \mathbf{x}_{0,<k}) = \pi_\theta(\mathbf{x}_0 \mid \mathbf{y}_0),$$

$$\pi_\theta(\mathbf{y}_0 \mid \mathcal{X}_0) = \prod_{l=1}^{L} \pi_\theta(y_{0,l} \mid \mathbf{x}_0, \mathbf{y}_{0,<l}) = \pi_\theta(\mathbf{y}_0 \mid \mathbf{x}_0).$$

Given the prompts follows a Uniform distribution, Bayes' rule yields

$$\log \pi_\theta(\mathbf{y}_0 \mid \mathbf{x}_0) = \log \pi_\theta(\mathbf{x}_0 \mid \mathbf{y}_0) - \log \pi_\theta(\mathbf{x}_0) + C.$$

where $C := \log P(\mathbf{y}_0)$ is a constant under the uniform prompt prior. Therefore,

$$\begin{aligned}
\Delta U_t(\mathbf{y}_0 \mid \mathcal{X}_0) &= \log \pi_{\theta_{t+1}}(\mathbf{y}_0 \mid \mathcal{X}_0) - \log \pi_{\theta_t}(\mathbf{y}_0 \mid \mathcal{X}_0) \\
&= (\log \pi_{\theta_{t+1}}(\mathbf{x}_0 \mid \mathbf{y}_0) - \log \pi_{\theta_t}(\mathbf{x}_0 \mid \mathbf{y}_0)) - (\log \pi_{\theta_{t+1}}(\mathbf{x}_0) - \log \pi_{\theta_t}(\mathbf{x}_0)) \\
&= \Delta G_t(\mathbf{x}_0 \mid \mathcal{Y}_0) - \Delta \log \pi_t(\mathbf{x}_0).
\end{aligned} \tag{8}$$

Equation (8) implies that the learning dynamics of understanding $\Delta U_t(\mathbf{y}_0 \mid \mathcal{X}_0)$ and those of generation $\Delta G_t(\mathbf{x}_0 \mid \mathcal{Y}_0)$ differ only in the change of the marginal distribution $\pi_t(\mathbf{x}_0)$ between consecutive steps. We next consider the training dynamics of the generation and understanding branches under different post-training strategies, SFT and DPO. Table 10 summarizes the key notation used in the learning dynamics analysis for reference.

### E.1 LEARNING DYNAMICS UNDER SFT

Following equation 8, we first discuss the training dynamics of the generation branch, i.e., $\Delta G_t(\mathbf{x}_0 \mid \mathcal{Y}_0)$, and then provide an indirect estimation for the understanding branch $\Delta U_t(\mathbf{y}_0 \mid \mathcal{X}_0)$.

**Lemma 1** (Learning Dynamics of Generation under SFT). *Consider self-improvement proposed in Section 4 with SFT. At epoch $t$, the one-step learning dynamics of **generation** is*

$$\Delta G_t(\mathbf{x}_0 \mid \mathcal{Y}_0) = -\eta \sum_{k=1}^{M} \sum_{r=1}^{M} (\mathbf{e}_{x_{0,k}} - \pi_k^0)^\top \mathcal{K}_{k,r}^t(\mathcal{Y}_0, \mathcal{Y}_u)(\pi_r^u - \mathbf{e}_{x_{u,r}}) + \mathcal{O}(\eta^2), \tag{9}$$

*where $\pi_r^u = \mathrm{softmax}(\mathbf{z}_r^u)$ and $\mathbf{z}_r^u = [h_\theta(\mathcal{Y}_u)]_r$ are the logits at image position $r$ obtained by running $h_\theta$ on $\mathcal{Y}_u$ and $\mathcal{K}_{k,r}^t(\mathcal{Y}_0, \mathcal{Y}_u) := (\nabla_{\theta_t} \mathbf{z}_k^0)(\nabla_{\theta_t} \mathbf{z}_r^u)^\top \in \mathbb{R}^{V \times V}$ is empirical neural tangent kernel (eNTK).*

*Proof.* We first show the learning dynamic of generation, i.e., $\Delta G_t(\mathbf{x}_0 \mid \mathcal{Y}_0)$ under the SFT setting. Consider the $k$-th image token

$$\begin{aligned}
\left(\Delta G_t(\mathbf{x}_0 \mid \mathcal{Y}_0)\right)_k &:= \left[\log \pi_{\theta_{t+1}}(\mathbf{x}_0 \mid \mathcal{Y}_0)\right]_k - \left[\log \pi_{\theta_t}(\mathbf{x}_0 \mid \mathcal{Y}_0)\right]_k \\
&= \nabla_\theta \left[\log \pi_{\theta_t}(\mathbf{x}_0 \mid \mathcal{Y}_0)\right]_k^\top (\theta_{t+1} - \theta_t) + \mathcal{O}(\|\theta_{t+1} - \theta_t\|^2).
\end{aligned} \tag{10}$$

where $\left[\log \pi_\theta(\mathbf{x}_0 \mid \mathcal{Y}_0)\right]_k := \log \pi_\theta(x_{0,k} \mid \mathbf{y}_0, \mathbf{x}_{0,<k})$.

Given the post-training data $(\mathbf{x}_u, \mathbf{y}_u)$, for generation, the negative log-likelihood loss of SFT is

$$\mathcal{L}_{\mathrm{SFT}}(\mathcal{Y}_u) = -\sum_{r=1}^{M} \log \pi_\theta(x_r = x_{u,r} \mid \mathcal{Y}_u) = -\sum_{r=1}^{M} \log \left[\pi_r^u\right]_{x_{u,r}}$$

where $\pi_r^u = \text{softmax}(\mathbf{z}_r^u)$ and $\mathbf{z}_r^u = [h_\theta(\mathcal{Y}_u)]_r$ are the logits at image position $r$ obtained by running $h_\theta$ on $\mathcal{Y}_u = [\mathbf{V}_u \mid \mathbf{U}_u]$. One-step SGD yields

$$\theta_{t+1} - \theta_t = -\eta \nabla_\theta \mathcal{L}_{\text{SFT}}(\mathcal{Y}_u) = -\eta \sum_{r=1}^{M} (\nabla_\theta \mathbf{z}_r^u)^\top \mathcal{G}_r,$$

where $\mathcal{G}_r := \pi_r^u - \mathbf{e}_{x_{u,r}} \in \mathbb{R}^V$.

Then, we obtain

$$\nabla_\theta \big[\log \pi_{\theta_t}(\mathbf{x}_0 \mid \mathcal{Y}_0)\big]_k = (\nabla_{\theta_t} \mathbf{z}_k^0)^\top (\mathbf{e}_{x_{0,k}} - \pi_k^0).$$

Therefore, Equation (10) can be rewritten as

$$\big(\Delta G_t(\mathbf{x}_0 \mid \mathcal{Y}_0)\big)_k = -\eta \sum_{r=1}^{M} (\mathbf{e}_{x_{0,k}} - \pi_k^0)^\top (\nabla_\theta \mathbf{z}_k^0)(\nabla_\theta \mathbf{z}_r^u)^\top \mathcal{G}_r + \mathcal{O}(\eta^2)$$

$$= -\eta \sum_{r=1}^{M} (\mathbf{e}_{x_{0,k}} - \pi_k^0)^\top \mathcal{K}_{k,r}^t(\mathcal{Y}_0, \mathcal{Y}_u)(\pi_r^u - \mathbf{e}_{x_{u,r}}) + \mathcal{O}(\eta^2)$$

where $\mathcal{K}_{k,r}^t(\mathcal{Y}_0, \mathcal{Y}_u) := (\nabla_{\theta_t} \mathbf{z}_k^0)(\nabla_{\theta_t} \mathbf{z}_r^u)^\top \in \mathbb{R}^{V \times V}$.

Finally, we have the sequence-level one-step change as:

$$\Delta G_t(\mathbf{x}_0 \mid \mathcal{Y}_0) = \sum_k \big[\log \pi_{\theta_{t+1}}(\mathbf{x}_0 \mid \mathcal{Y}_0)\big]_k - \sum_k \big[\log \pi_{\theta_t}(\mathbf{x}_0 \mid \mathcal{Y}_0)\big]_k$$

$$= \sum_{k=1}^{M} \big(\Delta G_t(\mathbf{x}_0 \mid \mathcal{Y}_0)\big)_k$$

$$= -\eta \sum_{k=1}^{M} \sum_{r=1}^{M} (\mathbf{e}_{x_{0,k}} - \pi_k^0)^\top \mathcal{K}_{k,r}^t(\mathcal{Y}_0, \mathcal{Y}_u)(\pi_r^u - \mathbf{e}_{x_{u,r}}) + \mathcal{O}(\eta^2).$$

The proof is complete. $\qquad \square$

**Lemma 2** (Learning Dynamics of Understanding under SFT). *Consider self-improvement proposed in Section 4 with SFT. At epoch t, the one-step learning dynamics of **understanding** is*

$$\Delta U_t(\mathbf{y}_0 \mid \mathcal{X}_0) = -\eta \sum_{k=1}^{M} \sum_{r=1}^{M} \sum_{\mathbf{y}_i \neq \mathbf{y}_0} w_{\theta_t}(\mathbf{y}_i \mid \mathbf{x}_0) \Big((\mathbf{e}_{x_{0,k}} - \pi_k^0)^\top \mathcal{K}_{k,r}^t(\mathcal{Y}_0, \mathcal{Y}_u) - (\mathbf{e}_{x_{0,k}} - \pi_k^i)^\top \mathcal{K}_{k,r}^t(\mathcal{Y}_i, \mathcal{Y}_u)\Big)(\pi_r^u - \mathbf{e}_{x_{u,r}})$$
$$+ \mathcal{O}(\eta^2)$$

(11)

*where* $w_{\theta_t}(\mathbf{y} \mid \mathbf{x}_0) := \frac{\pi_{\theta_t}(\mathbf{x}_0 \mid \mathbf{y})}{\sum_{\mathbf{y}'} \pi_{\theta_t}(\mathbf{x}_0 \mid \mathbf{y}')}$ *and* $\mathcal{Y}_i$ *denotes the concatenation of prompt* $\mathbf{y}_i \neq \mathbf{y}_0$ *and* $\mathbf{x}_0$.

*Proof.* We then analyze the learning dynamics of the understanding branch. By Equation (8) and a first–order log-sum-exp expansion, we obtain

$$\Delta \log \pi_t(\mathbf{x}_0) := \log \pi_{\theta_{t+1}}(\mathbf{x}_0) - \log \pi_{\theta_t}(\mathbf{x}_0)$$

$$= \log \sum_{\mathbf{y}} \pi_{\theta_{t+1}}(\mathbf{x}_0 \mid \mathbf{y}) - \log \sum_{\mathbf{y}} \pi_{\theta_t}(\mathbf{x}_0 \mid \mathbf{y})$$

$$= \Big\langle \sum_{\mathbf{y}} w_{\theta_t}(\mathbf{y} \mid \mathbf{x}_0) \nabla_\theta \log \pi_{\theta_t}(\mathbf{x}_0 \mid \mathbf{y}), \, \theta_{t+1} - \theta_t \Big\rangle + \mathcal{O}\big(\|\theta_{t+1} - \theta_t\|^2\big)$$

where the posterior weight is

$$w_{\theta_t}(\mathbf{y} \mid \mathbf{x}_0) := \frac{\pi_{\theta_t}(\mathbf{x}_0 \mid \mathbf{y})}{\sum_{\mathbf{y}'} \pi_{\theta_t}(\mathbf{x}_0 \mid \mathbf{y}')}.$$

Following Lemma 1 and Equation (8), we obtain

$$\Delta U_t(\mathbf{y}_0 \mid \mathcal{X}_0)$$
$$= \Delta G_t(\mathbf{x}_0 \mid \mathcal{Y}_0) - \Delta \log \pi_t(\mathbf{x}_0)$$
$$= \nabla_\theta \log \pi_{\theta_t}(\mathbf{x}_0 \mid \mathcal{Y}_0)^\top (\theta_{t+1} - \theta_t) - \sum_{\mathbf{y}_i} w_{\theta_t}(\mathbf{y}_i \mid \mathbf{x}_0) \nabla_\theta \log \pi_{\theta_t}(\mathbf{x}_0 \mid \mathcal{Y}_i)^\top (\theta_{t+1} - \theta_t) + \mathcal{O}(\|\theta_{t+1} - \theta_t\|^2)$$
$$= -\eta \sum_{k=1}^{M} \sum_{r=1}^{M} \sum_{\mathbf{y}_i \neq \mathbf{y}_0} w_{\theta_t}(\mathbf{y}_i \mid \mathbf{x}_0) \left( (\mathbf{e}_{x_{0,k}} - \pi_k^0)^\top \mathcal{K}_{k,r}^t(\mathcal{Y}_0, \mathcal{Y}_u) - (\mathbf{e}_{x_{0,k}} - \pi_k^i)^\top \mathcal{K}_{k,r}^t(\mathcal{Y}_i, \mathcal{Y}_u) \right) (\pi_r^u - \mathbf{e}_{x_{u,r}})$$
$$+ \mathcal{O}(\eta^2)$$

$$(12)$$

where $\mathcal{Y}_i$ denote the concatenation obtained by appending the embedding of $\mathbf{y}_i$ to $\mathbf{U}_0$.

The proof is complete. $\qquad\square$

## E.2 LEARNING DYNAMICS UNDER DPO

**Lemma 3** (Learning Dynamics of Generation under DPO). *Consider self-improvement proposed in Section 4 with DPO. At epoch $t$, the one-step learning dynamics of **generation** is*

$$\Delta G_t(\mathbf{x}_0 \mid \mathcal{Y}_0)$$
$$= -\eta \beta \sigma(-\alpha) \sum_{k=1}^{M} \sum_{r=1}^{M} (\mathbf{e}_{x_{0,k}} - \pi_k^0)^\top \left[ \mathcal{K}_{k,r}^t(\mathcal{Y}_0, \mathcal{Y}_u^+)(\pi_r^{u,+} - \mathbf{e}_{x_{u,r}^+}) - \mathcal{K}_{k,r}^t(\mathcal{Y}_0, \mathcal{Y}_u^-)(\pi_r^{u,-} - \mathbf{e}_{x_{u,r}^-}) \right]$$
$$+ \mathcal{O}(\eta^2)$$

$$(13)$$

*where the margin $\alpha := \beta \log \frac{\pi_\theta(\mathbf{x}_u^+ \mid \mathcal{Y}_u^+)}{\pi_{\mathrm{ref}}(\mathbf{x}_u^+ \mid \mathcal{Y}_u^+)} - \beta \log \frac{\pi_\theta(\mathbf{x}_u^- \mid \mathcal{Y}_u^-)}{\pi_{\mathrm{ref}}(\mathbf{x}_u^- \mid \mathcal{Y}_u^-)}$ and $\pi_r^{u,+} = \mathrm{softmax}(\mathbf{z}_r^{u,+})$ and $\mathbf{z}_r^{u,+} = [h_\theta(\mathcal{Y}_u^+)]_r$ are the logits at image position $r$ obtained by running $h_\theta$ on $\mathcal{Y}_u^+$. The neural tangent kernel $\mathcal{K}_{k,r}^t(\mathcal{Y}_0, \mathcal{Y}_u^+) := \nabla_\theta \mathbf{z}_k^0 (\nabla_\theta \mathbf{z}_r^{u,+})^\top$ and $\mathcal{K}_{k,r}^t(\mathcal{Y}_0, \mathcal{Y}_u^-) := \nabla_\theta \mathbf{z}_k^0 (\nabla_\theta \mathbf{z}_r^{u,-})^\top$.*

*Proof.* Following equation 10, one-step SGD yields

$$\theta_{t+1} - \theta_t = -\eta \nabla_\theta \mathcal{L}_{\mathrm{DPO}}(\mathcal{Y}_u)$$
$$= -\eta \sum_{r=1}^{M} \left[ (\nabla_\theta \mathbf{z}_r^{u,+})^\top \nabla_{\mathbf{z}_r^{u,+}} \mathcal{L}_{\mathrm{DPO}} + (\nabla_\theta \mathbf{z}_r^{u,-})^\top \nabla_{\mathbf{z}_r^{u,-}} \mathcal{L}_{\mathrm{DPO}} \right]$$
$$= -\eta \beta \sigma(-\alpha) \sum_{r=1}^{M} \left[ (\nabla_\theta \mathbf{z}_r^{u,+})^\top (\pi_r^{u,+} - \mathbf{e}_{x_{u,r}^+}) - (\nabla_\theta \mathbf{z}_r^{u,-})^\top (\pi_r^{u,-} - \mathbf{e}_{x_{u,r}^-}) \right],$$

where the margin $\alpha := \beta \log \frac{\pi_\theta(\mathbf{x}_u^+ \mid \mathcal{Y}_u^+)}{\pi_{\mathrm{ref}}(\mathbf{x}_u^+ \mid \mathcal{Y}_u^+)} - \beta \log \frac{\pi_\theta(\mathbf{x}_u^- \mid \mathcal{Y}_u^-)}{\pi_{\mathrm{ref}}(\mathbf{x}_u^- \mid \mathcal{Y}_u^-)}$. And $\pi_r^{u,+} = \mathrm{softmax}(\mathbf{z}_r^{u,+})$ and $\mathbf{z}_r^{u,+} = [h_\theta(\mathcal{Y}_u^+)]_r$ are the logits at image position $r$ obtained by running $h_\theta$ on $\mathcal{Y}_u^+ = [\mathbf{V}_u \mid \mathbf{U}_u^+]$.

Then, we have

$$\left( \Delta G_t(\mathbf{x}_0 \mid \mathcal{Y}_0) \right)_k$$
$$= -\eta \beta \sigma(-\alpha) \sum_{r=1}^{M} (\mathbf{e}_{x_{0,k}} - \pi_k^0)^\top (\nabla_\theta \mathbf{z}_k^0) \left[ (\nabla_\theta \mathbf{z}_r^{u,+})^\top (\pi_r^{u,+} - \mathbf{e}_{x_{u,r}^+}) - (\nabla_\theta \mathbf{z}_r^{u,-})^\top (\pi_r^{u,-} - \mathbf{e}_{x_{u,r}^-}) \right] + \mathcal{O}(\eta^2)$$
$$= -\eta \beta \sigma(-\alpha) \sum_{r=1}^{M} (\mathbf{e}_{x_{0,k}} - \pi_k^0)^\top \left[ \mathcal{K}_{k,r}^t(\mathcal{Y}_0, \mathcal{Y}_u^+)(\pi_r^{u,+} - \mathbf{e}_{x_{u,r}^+}) - \mathcal{K}_{k,r}^t(\mathcal{Y}_0, \mathcal{Y}_u^-)(\pi_r^{u,-} - \mathbf{e}_{x_{u,r}^-}) \right] + \mathcal{O}(\eta^2)$$

where the neural tangent kernel $\mathcal{K}_{k,r}^t(\mathcal{Y}_0, \mathcal{Y}_u^+) := \nabla_\theta \mathbf{z}_k^0 (\nabla_\theta \mathbf{z}_r^{u,+})^\top$ and $\mathcal{K}_{k,r}^t(\mathcal{Y}_0, \mathcal{Y}_u^-) := \nabla_\theta \mathbf{z}_k^0 (\nabla_\theta \mathbf{z}_r^{u,-})^\top$.

Finally, we have the sequence-level one-step change as:

$$\Delta G_t(\mathbf{x}_0 \mid \mathcal{Y}_0)$$

$$= -\eta\beta\sigma(-\alpha) \sum_{k=1}^{M} \sum_{r=1}^{M} (\mathbf{e}_{x_{0,k}} - \pi_k^0)^\top \Big[ \mathcal{K}_{k,r}^t(\mathcal{Y}_0, \mathcal{Y}_u^+)(\pi_r^{u,+} - \mathbf{e}_{x_{u,r}^+}) - \mathcal{K}_{k,r}^t(\mathcal{Y}_0, \mathcal{Y}_u^-)(\pi_r^{u,-} - \mathbf{e}_{x_{u,r}^-}) \Big]$$

$$+ \mathcal{O}(\eta^2)$$

The proof is complete. $\qquad\square$

**Lemma 4** (Learning Dynamics of Understanding under DPO)**.** *Consider self-improvement proposed in Section 4 with DPO. At epoch $t$, the one-step learning dynamics of **understanding** is*

$$\Delta U_t(\mathbf{y}_0 \mid \mathcal{X}_0)$$

$$= -\eta\beta\sigma(-\alpha) \sum_{k=1}^{M} \sum_{r=1}^{M} \sum_{\mathbf{y}_i \neq \mathbf{y}_0} w_{\theta_t}(\mathbf{y}_i \mid \mathbf{x}_0) \Bigg( (\mathbf{e}_{x_{0,k}} - \pi_k^0)^\top \Big( \mathcal{K}_{k,r}^t(\mathcal{Y}_0, \mathcal{Y}_u^+)(\pi_r^{u,+} - \mathbf{e}_{x_{u,r}^+}) - \mathcal{K}_{k,r}^t(\mathcal{Y}_0, \mathcal{Y}_u^-)(\pi_r^{u,-} - \mathbf{e}_{x_{u,r}^-}) \Big)$$

$$- (\mathbf{e}_{x_{0,k}} - \pi_k^i)^\top \Big( \mathcal{K}_{k,r}^t(\mathcal{Y}_i, \mathcal{Y}_u^+)(\pi_r^{u,+} - \mathbf{e}_{x_{u,r}^+}) - \mathcal{K}_{k,r}^t(\mathcal{Y}_i, \mathcal{Y}_u^-)(\pi_r^{u,-} - \mathbf{e}_{x_{u,r}^-}) \Big) \Bigg) + \mathcal{O}(\eta^2)$$

$$(14)$$

*where $\mathcal{Y}_i$ denote the concatenation obtained by appending the embedding of $\mathbf{y}_i$ to $\mathbf{U}_0$.*

*Proof.* Following 2, for the learning dynamics of understanding under DPO, we have

$$\Delta U_t(\mathbf{y}_0 \mid \mathcal{X}_0)$$
$$= \Delta G_t(\mathbf{x}_0 \mid \mathcal{Y}_0) - \Delta \log \pi_t(\mathbf{x}_0)$$
$$= \nabla_\theta \log \pi_{\theta_t}(\mathbf{x}_0 \mid \mathcal{Y}_0)^\top (\theta_{t+1} - \theta_t) - \sum_{\mathbf{y}_i} w_{\theta_t}(\mathbf{y}_i \mid \mathbf{x}_0) \nabla_\theta \log \pi_{\theta_t}(\mathbf{x}_0 \mid \mathcal{Y}_i)^\top (\theta_{t+1} - \theta_t) + \mathcal{O}(\|\theta_{t+1} - \theta_t\|^2)$$

$$= \sum_{\mathbf{y}_i \neq \mathbf{y}_0} w_{\theta_t}(\mathbf{y}_i \mid \mathbf{x}_0) \Big( \nabla_\theta \log \pi_{\theta_t}(\mathbf{x}_0 \mid \mathcal{Y}_0)^\top - \nabla_\theta \log \pi_{\theta_t}(\mathbf{x}_0 \mid \mathcal{Y}_i)^\top \Big)(\theta_{t+1} - \theta_t) + \mathcal{O}(\|\theta_{t+1} - \theta_t\|^2)$$

$$= -\eta\beta\sigma(-\alpha) \sum_{k=1}^{M} \sum_{r=1}^{M} \sum_{\mathbf{y}_i \neq \mathbf{y}_0} w_{\theta_t}(\mathbf{y}_i \mid \mathbf{x}_0) \Bigg( \Big( (\mathbf{e}_{x_{0,k}} - \pi_k^0)^\top \mathcal{K}_{k,r}^t(\mathcal{Y}_0, \mathcal{Y}_u^+) - (\mathbf{e}_{x_{0,k}} - \pi_k^i)^\top \mathcal{K}_{k,r}^t(\mathcal{Y}_i, \mathcal{Y}_u^+) \Big)(\pi_r^{u,+} - \mathbf{e}_{x_{u,r}^+})$$

$$- \Big( (\mathbf{e}_{x_{0,k}} - \pi_k^0)^\top \mathcal{K}_{k,r}^t(\mathcal{Y}_0, \mathcal{Y}_u^-) - (\mathbf{e}_{x_{0,k}} - \pi_k^i)^\top \mathcal{K}_{k,r}^t(\mathcal{Y}_i, \mathcal{Y}_u^-) \Big)(\pi_r^{u,-} - \mathbf{e}_{x_{u,r}^-}) \Bigg) + \mathcal{O}(\eta^2)$$

$$= -\eta\beta\sigma(-\alpha) \sum_{k=1}^{M} \sum_{r=1}^{M} \sum_{\mathbf{y}_i \neq \mathbf{y}_0} w_{\theta_t}(\mathbf{y}_i \mid \mathbf{x}_0) \Bigg( (\mathbf{e}_{x_{0,k}} - \pi_k^0)^\top \Big( \mathcal{K}_{k,r}^t(\mathcal{Y}_0, \mathcal{Y}_u^+)(\pi_r^{u,+} - \mathbf{e}_{x_{u,r}^+}) - \mathcal{K}_{k,r}^t(\mathcal{Y}_0, \mathcal{Y}_u^-)(\pi_r^{u,-} - \mathbf{e}_{x_{u,r}^-}) \Big)$$

$$- (\mathbf{e}_{x_{0,k}} - \pi_k^i)^\top \Big( \mathcal{K}_{k,r}^t(\mathcal{Y}_i, \mathcal{Y}_u^+)(\pi_r^{u,+} - \mathbf{e}_{x_{u,r}^+}) - \mathcal{K}_{k,r}^t(\mathcal{Y}_i, \mathcal{Y}_u^-)(\pi_r^{u,-} - \mathbf{e}_{x_{u,r}^-}) \Big) \Bigg) + \mathcal{O}(\eta^2)$$

$$(15)$$

where $\mathcal{Y}_i$ denote the concatenation obtained by appending the embedding of $\mathbf{y}_i$ to $\mathbf{U}_0$. $\qquad\square$

# F   ABLATION STUDY AND MORE EXPLORATIONS

## F.1   SELF-IMPROVEMENTS ON ADDITIONAL MODELS

We conduct SFT-based self-improvement experiments on an additional model, Janus-Pro-1B (Chen et al., 2025b), where the training data, training pipeline, hyperparameters and evaluation metrics follow Section 4.2.1. Table 17 shows that the self-improved model exhibits improvements in generation, understanding, and unification, further confirming the effectiveness of self-improvement.

Additionally, we conducted the same self-improvement analysis on the BAGEL model (Deng et al., 2025) in Table 12 and Table 13. However, we found that improvements in generation capability provide only limited gains to BAGEL's understanding ability. This may be because BAGEL adopts a Mixture-of-Transformer-Experts (MoT) architecture (Deng et al., 2025), where the interaction between generation and understanding occurs only through shared self-attention operations, rather than through a shared LLM as in Janus-Pro, which unifies the integration of LLM and diffusion models within a single transformer. This observation may suggest that architectural choices lead to differences in the interaction of generation and understanding.

| Model | Texture | | | Shape | | | Color | | | Spatial | | | Non-Spatial | | | Complex | | | Overall | | |
|---|---|---|---|---|---|---|---|---|---|---|---|---|---|---|---|---|---|---|---|---|---|
| | Gen.↑ | Und.↑ | Non.↓ | Gen.↑ | Und.↑ | Non.↓ | Gen.↑ | Und.↑ | Non.↓ | Gen.↑ | Und.↑ | Non.↓ | Gen.↑ | Und.↑ | Non.↓ | Gen.↑ | Und.↑ | Non.↓ | Gen.↑ | Und.↑ | Non.↓ |
| *Gen. and Und.* | | | | | | | | | | | | | | | | | | | | | |
| Janus-Pro-1B$_{(Baseline)}$ | 50.85 | 50.00 | 51.33 | 43.11 | 50.00 | 46.00 | 55.81 | 50.00 | 29.83 | 7.99 | 50.00 | 42.67 | 21.34 | 50.00 | 9.00 | 23.05 | 50.00 | 16.00 | 33.70 | 50.00 | 32.47 |
| + *SFT* | 55.58 | 53.76 | 41.00 | 43.67 | 32.38 | 39.00 | 59.34 | 58.70 | 24.07 | 10.40 | 53.87 | 35.33 | 28.84 | 65.28 | 9.33 | 32.72 | 51.58 | 15.00 | 38.43 | 52.60 | 27.29 |

Table 11: Self-improvement results of Janus-Pro-1B on T2I-CompBench++, where the understanding score is evaluated using Gemini-Pro-2.5 as the external evaluator.

| Model | Texture | | | Shape | | | Color | | | Spatial | | | Non-Spatial | | | Complex | | | Overall | | |
|---|---|---|---|---|---|---|---|---|---|---|---|---|---|---|---|---|---|---|---|---|---|
| | Gen.↑ | Und.↑ | Non.↓ | Gen.↑ | Und.↑ | Non.↓ | Gen.↑ | Und.↑ | Non.↓ | Gen.↑ | Und.↑ | Non.↓ | Gen.↑ | Und.↑ | Non.↓ | Gen.↑ | Und.↑ | Non.↓ | Gen.↑ | Und.↑ | Non.↓ |
| *Gen. and Und.* | | | | | | | | | | | | | | | | | | | | | |
| BAGEL$_{(Baseline)}$ | 70.65 | 50.00 | 9.00 | 58.32 | 50.00 | 20.67 | 0.8164 | 50.00 | 7.46 | 33.47 | 50.00 | 22.67 | 0.3110 | 50.00 | 8.33 | 37.33 | 50.00 | 13.33 | 52.09 | 50.00 | 13.58 |
| + *SFT* | 71.12 | 66.67 | 8.33 | 58.33 | 42.86 | 20.33 | 81.75 | 25.00 | 8.47 | 34.53 | 58.33 | 22.00 | 31.13 | 33.33 | 9.00 | 37.25 | 50.00 | 13.00 | 52.35 | 46.02 | 13.58 |

Table 12: Self-improvement results of BAGEL on T2I-CompBench++.

## F.2 Ablation on Updated Model Components

In Section 4.2, we update only the parameters of the LLM component during self-improvement, while keeping all other components frozen. This design aligns with prior work on MLLMs (focused solely on image understanding), which suggests that optimizing the LLM alone is sufficient to improve MLLM performance, while updating other components yields limited gains (Verma et al., 2024). Table 14 supports our setting: fine-tuning only the LLM already enables the self-improved Janus-Pro-7B to achieve improvements in generation, understanding and unification. However, expanding the parameter updates to include the image aligner (a two-layer MLP projector that maps image tokens to the LLM input space), the generation head (a two-layer MLP that projects LLM output into tokenizer's codebook space), or even the vision tower, did not lead to significant performance gains in generation and slight declines were observed in both understanding and unification.

## F.3 Ablation on Image Candidates N

Table 15 reports the number of post-training samples produced under different values of image candidate $N$ (see details in Alg. 1). We observe that as $N$ increases, the number of constructed samples gradually saturates. In this paper, we adopt a large value of $N = 10$ for data construction.

## F.4 Ablation on Curriculum Learning

We introduced curriculum learning at different training epochs (4 and 10). Curriculum replay at both epochs improved self-improvement performance, though performance was better when replay was applied at epoch 10. This is likely because the model's generative and understanding capabilities had improved by that stage, enabling a more effective use of earlier samples for expanding post-training data. Accordingly, we use epoch 10 for curriculum replay in all experiments.

## F.5 Improvement with External Reward

We construct post-training data using external Qwen2.5-VL-72B-Instruct. For Janus-Pro-7B with the SFT strategy, Table 17 compares Qwen-based alignment with self-improvement. Qwen enables Janus-Pro-7B to achieve better generation and unification. Self-improvement yields slightly weaker alignment, likely due to Janus-Pro-7B's inferior image understanding capability compared to Qwen. Nevertheless, without introducing any external signals, the self-improvement method achieves results close to those obtained with Qwen-based alignment.

## G The Use of Large Language Models (LLMs)

In writing, we used LLMs (ChatGPT-5) for manuscript-wide grammar checking and sentence polishing. In coding, we leveraged LLMs (ChatGPT-5) for debugging our self-improvement pipeline and assisting with figure-visualization scripts.

| Model | POPE↑ | MMB↑ | GQA↑ | MMMU↑ |
|---|---|---|---|---|
| BAGEL | 100.00 | 88.05 | 52.56 | 64.03 |
| + *SFT* | 100.00 | 88.14 | 52.44 | 64.10 |

Table 13: Improving BAGEL's generation leads to limited improvements in its understanding.

| Model | Texture | | | Shape | | | Color | | | Spatial | | | Non-Spatial | | | Complex | | | Overall | | |
|---|---|---|---|---|---|---|---|---|---|---|---|---|---|---|---|---|---|---|---|---|---|
| | Gen.↑ | Und.↑ | Non.↓ | Gen.↑ | Und.↑ | Non.↓ | Gen.↑ | Und.↑ | Non.↓ | Gen.↑ | Und.↑ | Non.↓ | Gen.↑ | Und.↑ | Non.↓ | Gen.↑ | Und.↑ | Non.↓ | Gen.↑ | Und.↑ | Non.↓ |
| *Gen. and Und.* | | | | | | | | | | | | | | | | | | | | | |
| Janus-Pro-7B(Baseline) | 38.63 | 50.00 | 43.33 | 33.49 | 50.00 | 43.00 | 53.22 | 50.00 | 27.33 | 16.81 | 50.00 | 31.00 | 31.40 | 50.00 | 2.33 | 37.73 | 50.00 | 10.33 | 35.21 | 50.00 | 26.22 |
| + *LLM* | 53.93 | 65.22 | 29.67 | 38.63 | 53.85 | 34.00 | 73.41 | 54.62 | 10.85 | 23.73 | 26.67 | 22.00 | 31.45 | 75.00 | 1.00 | 38.57 | 75.00 | 4.33 | 43.29 | 58.39 | 16.98 |
| + *LLM and Projector* | 52.98 | 51.72 | 31.33 | 40.88 | 56.67 | 37.67 | 73.61 | 22.73 | 13.90 | 21.04 | 35.71 | 23.33 | 31.41 | 66.67 | 2.00 | 38.70 | 75.00 | 4.67 | 42.10 | 51.42 | 18.82 |
| + *LLM and Projector and Vision Tower* | 53.62 | 55.17 | 28.00 | 39.39 | 56.67 | 36.00 | 73.56 | 25.00 | 10.17 | 22.45 | 33.33 | 21.00 | 31.41 | 100.00 | 0.67 | 38.64 | 63.64 | 6.33 | 43.18 | 55.64 | 17.02 |

Table 14: Based on Janus-Pro-7B, we conducted self-improvement via SFT and observed that only fine-tuning the LLM was sufficient to achieve improvements in both performance and unification. Updating other components, such as the vision tower and projectors, yielded no significant gains.

| MLLM | N=2 | N=4 | N=6 | N=8 | N=10 |
|---|---|---|---|---|---|
| Janus-Pro-7B | 254 | 1338 | 1823 | 2088 | 2265 |
| Show-o | 80 | 160 | 192 | 208 | 226 |

Table 15: Data expansion slows down as $N$ increases.

| Model | Texture | | | Shape | | | Color | | | Spatial | | | Non-Spatial | | | Complex | | | Overall | | |
|---|---|---|---|---|---|---|---|---|---|---|---|---|---|---|---|---|---|---|---|---|---|
| | Gen.↑ | Und.↑ | Non.↓ | Gen.↑ | Und.↑ | Non.↓ | Gen.↑ | Und.↑ | Non.↓ | Gen.↑ | Und.↑ | Non.↓ | Gen.↑ | Und.↑ | Non.↓ | Gen.↑ | Und.↑ | Non.↓ | Gen.↑ | Und.↑ | Non.↓ |
| *Gen. and Und.* | | | | | | | | | | | | | | | | | | | | | |
| Janus-Pro-7B(Baseline) | 38.63 | 50.00 | 43.33 | 33.49 | 50.00 | 43.00 | 53.22 | 50.00 | 27.33 | 16.81 | 50.00 | 31.00 | 31.40 | 50.00 | 2.33 | 37.73 | 50.00 | 10.33 | 35.21 | 50.00 | 26.22 |
| + *SFT* | 53.93 | 65.22 | 29.67 | 38.63 | 53.85 | 34.00 | 73.41 | 54.62 | 10.85 | 23.73 | 26.67 | 22.00 | 31.45 | 75.00 | 1.00 | 38.57 | 75.00 | 4.33 | 43.29 | 58.39 | 16.98 |
| + *C-SFT (10)* | 56.38 | 66.67 | 28.33 | 39.86 | 64.52 | 33.67 | 73.77 | 52.14 | 12.20 | 24.87 | 38.46 | 21.67 | 31.44 | 75.00 | 2.33 | 38.78 | 70.00 | 3.33 | 44.18 | 61.13 | 16.92 |
| + *C-SFT (4)* | 55.95 | 50.00 | 28.33 | 39.23 | 60.00 | 32.67 | 74.67 | 52.73 | 10.85 | 23.42 | 26.67 | 23.00 | 31.38 | 75.00 | 0.33 | 38.49 | 77.27 | 7.67 | 43.86 | 56.94 | 17.14 |

Table 16: Curriculum learning at different epochs consistently leads to better self-improvement, and we consistently apply it at a later epoch (epoch 10).

| Model | Texture | | | Shape | | | Color | | | Spatial | | | Non-Spatial | | | Complex | | | Overall | | |
|---|---|---|---|---|---|---|---|---|---|---|---|---|---|---|---|---|---|---|---|---|---|
| | Gen.↑ | Und.↑ | Non.↓ | Gen.↑ | Und.↑ | Non.↓ | Gen.↑ | Und.↑ | Non.↓ | Gen.↑ | Und.↑ | Non.↓ | Gen.↑ | Und.↑ | Non.↓ | Gen.↑ | Und.↑ | Non.↓ | Gen.↑ | Und.↑ | Non.↓ |
| *Gen. and Und.* | | | | | | | | | | | | | | | | | | | | | |
| Janus-Pro-7B(Baseline) | 38.63 | 50.00 | 43.33 | 33.49 | 50.00 | 43.00 | 53.22 | 50.00 | 27.33 | 16.81 | 50.00 | 31.00 | 31.40 | 50.00 | 2.33 | 37.73 | 50.00 | 10.33 | 35.21 | 50.00 | 26.22 |
| + *Self-improved SFT* | 53.93 | 65.22 | 29.67 | 38.63 | 53.85 | 34.00 | 73.41 | 54.62 | 10.85 | 23.73 | 26.67 | 22.00 | 31.45 | 75.00 | 1.00 | 38.57 | 75.00 | 4.33 | 43.29 | 58.39 | 16.98 |
| + *Qwen-assisted SFT* | 56.84 | 56.00 | 25.00 | 41.53 | 59.26 | 37.33 | 76.18 | 49.63 | 11.86 | 24.14 | 31.25 | 19.33 | 31.48 | 70.00 | 1.00 | 38.53 | 66.67 | 5.33 | 44.78 | 55.47 | 16.64 |

Table 17: Constructing post-training samples with Qwen also enhances the generation, understanding, and unification of MLLMs. Without any external rewards, the self-improvement method yields slightly lower performance and unification than Qwen-based MLLMs.

