# OpenReview forum: "Turning Internal Gap into Self-Improvement: Promoting the Generation-Understanding Unification in MLLMs"
_ICLR.cc/2026/Conference — ICLR 2026 Poster_

### Official Review · Reviewer_fCxY · 2025-10-26

**Soundness:** 2
**Presentation:** 3
**Contribution:** 2
**Rating:** 2
**Confidence:** 5

**Summary:**

This paper investigates the "internal gap" in unified Multimodal Large Language Models (MLLMs), where understanding consistently outperforms generation. The authors propose a self-improvement framework that leverages stronger understanding to guide weaker generation through standard post-training (SFT/DPO), achieving up to 20% generation improvement without external signals.

**Strengths:**

The paper provides systematic empirical validation of the generation-understanding gap across multiple MLLMs with clear methodology and reproducible experiments. The theoretical framework extending learning dynamics to multimodal settings offers mathematical formalization, though it essentially explains an expected outcome—that shared parameters naturally lead to correlated improvements. The work is technically sound with detailed ablations.

**Weaknesses:**

The work is technically sound with detailed ablations, but the core insight that two capabilities in a closed system can mutually improve through iterative training is conceptually straightforward, limiting originality. The self-improvement approach applies standard techniques (SFT/DPO) without novel algorithmic contributions, and still underperforms methods using external rewards, constraining its practical significance. Critically, the theory fails to address fundamental limitations: in a closed system without external supervision, two learners cannot indefinitely improve through mutual training—the paper lacks analysis of performance ceilings or convergence bounds.  Overall, the paper competently documents a predictable phenomenon rather than introducing fundamentally new concepts.
More specifically, the theoretical analysis (Propositions 1-2) provides mathematical formalization but offers limited conceptual depth beyond expected outcomes. Experimental scope is narrow: only two models tested, consistently underperforming external reward baselines, undermining practical value. The non-unification metric relies on potentially unreliable binary judgments, and using Qwen as ground truth introduces unexamined biases. Curriculum learning adds minimal gains without principled design.

**Questions:**

no more.

---

> ### Author Response · Authors · 2025-11-21
>
> We thank the reviewer for their time and for recognizing that the work is technically sound with detailed ablations. We now address the questions raised as follows.
>
> ---
> >Q1: The core insight that two capabilities in a closed system can mutually improve through iterative training is conceptually straightforward, limiting originality. | Overall, the paper competently documents a predictable phenomenon rather than introducing fundamentally new concepts.
>
> This comment may overlook the distinct characteristics and significance of self‑improvement in unified MLLMs.
>
> - First, self-improvement in unified MLLMs is not simply a matter of `two capabilities in a closed system mutually improving`.  In unified MLLMs, the generation and understanding branches **share the same parameters** (e.g., a shared LLM). **When optimizing one branch** (e.g., generation), **whether the other branch** (e.g., understanding) **will mutually improve is entirely not guaranteed.** The coupled structure of generation and understanding is fundamentally different from the behavior of two decoupled components in a closed system, **whose behavior is inherently unpredictable**.
> **This study shows**, in a coupled unified MLLM, **only improving generation can also passively improve understanding**. We further attribute this phenomenon to the shared eNTK between generation and understanding tasks. To the best of our knowledge, prior work has not thoroughly explored the mechanisms underlying a coupled generation–understanding system, which constitutes an **original contribution of our study**.
> - Second, in explaining the interplay between generation and understanding, **we also provide the following explorations**：
>
>   - formally defining the internal gap metric and empirically validating its existence across six unified MLLMs and nine subtasks,
>   - proposing a self-improvement framework to mitigate this gap and demonstrating its effectiveness through extensive experiments,
>   - further enhancing self-improvement by explicitly leveraging this co-improvement property.
>
> These explorations go beyond merely `documenting a predictable phenomenon` and emphasize that unified MLLMs should focus on disparities in task capabilities for better unification.
>
> >Q2: The self-improvement approach applies standard techniques (SFT/DPO) without novel algorithmic contributions, and still underperforms methods using external rewards, constraining its practical significance.| consistently underperforming external reward baselines, undermining practical value
>
> There may be some misunderstanding about the scope of our work. Specifically,
> - Regarding the comment `The self-improvement approach applies standard techniques (SFT/DPO) without novel algorithmic contributions`, we would like to clarify that **the goal of this paper is not to propose a new post-training algorithm**. **Our goal** is to systematically examine whether unified models have truly achieved internal unity, and to explore how to promote the alignment between generation and understanding. **To achieve this**, we investigate the internal gap in unified MLLMs (`Section 3`), show self-improvement can alleviate this issue (`Section 4`), and analyze the unique property (i.e., co-improvement) of applying self-improvement within a coupled model (`Section 5`). While designing more advanced post-training algorithms is interesting, **it is beyond the scope of this work** and left for future research.
> - Regarding the claim that our method `still underperforms methods using external rewards`, we would like to point out that the baselines in `Table 1` (e.g., `T2I-R1`, `HermesFlow`) all rely on external reward models (see `Line 465-469`). **Our self-improvement approach** achieves performance that is **comparable to and in some cases even surpasses these external-reward methods** (see `Section 6` ). Therefore, the statement that our method `still underperforms methods using external rewards` **is not accurate**.
> Moreover, although **stronger external reward** can certainly further improve performance, **this is not in conflict with self-improvement.** In fact, self‑improvement constitutes a free bonus that requires no additional conditions, and a self‑improved model can further leverage stronger external signals for additional gains. Such hybrid approaches, however, fall outside the scope of our focus (see `Section 2` for more discussion).
> - Regarding the claim that this `constrains its practical significance`, **external rewards can be costly or difficult to obtain [1,2,3,4], whereas our approach relies solely on internal signals.** It is therefore unclear **why** self-improvement based on internal rewards **would be less practical** than methods requiring external rewards.  **If the reviewer can provide further clarification**, we would be happy to discuss this point in more detail.

---

> > ### Author Response · Authors · 2025-11-21
> >
> > >Q3: Critically, the theory fails to address fundamental limitations: in a closed system without external supervision, two learners cannot indefinitely improve through mutual training—the paper lacks analysis of performance ceilings or convergence bounds.｜ More specifically, the theoretical analysis (Propositions 1-2) provides mathematical formalization but offers limited conceptual depth beyond expected outcomes.
> >
> > We respectfully disagree with this comment, as it contains several factual misunderstandings.
> >
> > - Our setting is not `mutual training`, nor do we claim any form of `indefinitely improve`. We only optimize the generation branch (see `Section 3`). **The understanding branch is never directly trained** and is updated only passively via the shared backbone. Thus, **our work involves no mutual training** and **makes no claim of indefinite improvement**.
> > - *Why `Propositions 1–2` focus on the training dynamics of generation and understanding rather than on performance ceilings or convergence bounds.*
> > Using the learning-dynamics framework, **`Propositions 1–2` clearly explain** that the shared LLM architecture between generation and understanding induces a shared eNTK. Consequently, each generation‑focused optimization step that lowers the misgeneration probability $\pi\_{\theta}(\mathbf{x}\_0 \mid \mathbf{y}\_0)$ simultaneously decreases the misjudgment probability $\pi\_{\theta}(\mathbf{y}\_0 \mid \mathbf{x}\_0)$, resulting in co‑improvement. In contrast, analyzing `performance ceilings or convergence bounds` **cannot** characterize how the understanding ’s learning trajectory changes as generation is optimized, and is therefore **unrelated to** explaining the co-improvement phenomenon observed in `Section 4.2.2`.
> > - Regarding the statement `offers limited conceptual depth`, to the best of our knowledge, this study is among the early efforts to systematically examine the internal gap within unified MLLMs and to discuss the coupling mechanisms between generation and understanding. We would appreciate it **if the reviewer could share relevant references** supporting the view that our findings lack conceptual depth, so that we can engage in further discussion.
> >
> > >Q4: Experimental scope is narrow: only two models tested.
> >
> > We additionally report the self-improvement results for a 1B-sized model, `Janus-Pro-1B`, where the training data, training pipeline, hyperparameters, and evaluation metrics follow `Section 4.2.1`. **The table below** presents the performance improvements of the `self-improved Janus-Pro-1B` on `T2I-CompBench++`. We observe after leveraging the internal gap for self-improvement, the model achieves **consistent gains in generation** (average +5%), **understanding** (win rate > 50%), and **unification** (−5% reduction in gap) which further validate the effectiveness of our method. These results of are added in` Appendix F.1`.
> >
> >
> >   - Generation ($\uparrow$)
> >
> >  | Model | Texture | Shape |Color|Spatial | Non-Spatial|Complex |Overall|
> > | -------- | -------- | -------- |-------- |-------- |-------- |-------- |-------- |
> > |  Janus-Pro-1B   | 50.85     |   43.11  | 55.81    |  7.99  |21.34   |23.05    |33.70   |
> > |  +*SFT*    | 55.58     |   43.67 | 59.34  |  10.40  | 28.84    |32.72     |38.43    |
> >
> >   - Understanding ($\uparrow$)
> >
> >  | Model | Texture | Shape |Color|Spatial | Non-Spatial|Complex |Overall|
> > | -------- | -------- | -------- |-------- |-------- |-------- |-------- |-------- |
> > |  Janus-Pro-1B   | 50.00     |   50.00    | 50.00     | 50.00    |50.00   |50.00    |50.00    |
> > |  +*SFT*    | 53.76    |   32.38 | 58.70 |  53.87  | 65.28   |51.58   |52.60     |
> >
> >    - Non-unification ($\downarrow$)
> >
> >  | Model | Texture | Shape |Color|Spatial | Non-Spatial|Complex |Overall|
> > | -------- | -------- | -------- |-------- |-------- |-------- |-------- |-------- |
> > |  Janus-Pro-1B   | 51.33    |  46.00    | 29.83   | 42.67   |9.00   |16.00   |32.47   |
> > |  +*SFT*    | 41.00  |   39.00 | 24.07|  35.33 | 9.33  |15.00 |27.29   |

---

> > > ### Author Response · Authors · 2025-11-21
> > >
> > > >Q5: The non-unification metric relies on potentially unreliable binary judgments, and using Qwen as ground truth introduces unexamined biases.
> > >
> > > - Regarding the concern about `potentially unreliable binary judgments`, we clarify that the **non-unification metric is designed to** determine whether the generation and understanding branches are aligned **which is inherently a binary decision** (aligned vs. not aligned).  Therefore, **binary judgments are sufficient** for non-unification metric. Fine-grained scoring (e.g., a `0–2 scale`) would still collapse into the same two categories: both `level 0` and `level 1` correspond to not aligned (score $0$), and `level 2` corresponds to aligned (score $1$).
> > > - Regarding  `using Qwen as ground truth introduces unexamined biases`, we additionally employ another external evaluator, `Gemini-Pro-2.5` (which ranks 4th among more than 200 models in image understanding on [OpenCompass](https://opencompass-open-vlm-leaderboard.hf.space) ), to recompute the non-unification metric (weak generation score). **The table below** reports the results for `Janus-Pro-7B`: in the majority of subtasks (5/6), the weak generation score exceeds 50% and even reaches 100%, which is **fully consistent with the conclusions obtained using `Qwen`**. The revised manuscript further includes non-unification results for six unified models (see `Figure 9`).
> > >
> > >
> > > | External Judger | Single Obj.|Two Obj.|Color Attri.| Texture |Spatial |Complex |Physic |Chemistry|Biology |
> > > | -------- | -------- | -------- |-------- |-------- |-------- |-------- |-------- |-------- |-------- |
> > > | Gemini-Pro-2.5   | 88.89     |  77.78   |50.00   |89.23    | 92.47    |  74.19  |100.00 |100.00  |66.67  |
> > >
> > > >Q6: Curriculum learning adds minimal gains without principled design.
> > >
> > > We would like to clarify that the contribution of introducing curriculum learning is to demonstrate that **curriculum learning remains effective during self-improvement in unified MLLMs**, which is **not obvious** given the tight coupling between the generation and understanding branches. specially,
> > >
> > > - In classical settings, difficult samples are determined by an external stronger model or predefined rules and then gradually introduced during training [5,6,7].
> > > - However, unified MLLMs must rely on their own generation and understanding branches to dynamically re-evaluate previously discarded difficult samples. **This creates a new risk**: if understanding degrades while generation improves, the model may misjudge samples and reintroduce low-quality data, potentially harming performance.
> > >
> > > Our paper shows that **this risk is mitigated** because generation and understanding can improve together, **ensuring the feasibility of applying curriculum learning in unified MLLMs**. Subsequent experiments (`Table 1`) further confirm this insight. Designing more effective curriculum strategies to achieve larger improvements is an interesting direction, but it is not the core focus of this paper, and we leave it for future work.
> > >
> > >
> > > [1] Large language models can self-improve
> > >
> > > [2] Self-Instruct: Aligning Language Models with Self-Generated Instructions
> > >
> > > [3] Constitutional AI: Harmlessness from AI Feedback
> > >
> > > [4] Self-Rewarding Language Models
> > >
> > > [5] Curry-DPO: Enhancing Alignment using Curriculum Learning & Ranked Preferences
> > >
> > > [6] 2D-Curri-DPO: Two-Dimensional Curriculum Learning for Direct Preference Optimization
> > >
> > > [7] Curriculum Direct Preference Optimization for Diffusion and Consistency Models

---

> ### Author Response · Authors · 2025-11-27
>
> Dear Reviewer fCxY,
>
> We would like to express our sincere gratitude once again for the time and effort you have devoted to reviewing our work. As the discussion period draws to a close, we would be grateful if you could let us know whether our responses have adequately addressed your concerns. Should you have any further questions or comments, please do not hesitate to share them. We look forward to continuing our communication with you.
>
> Best,
>
> Authors

---

### Official Review · Reviewer_Qp5p · 2025-10-30

**Soundness:** 3
**Presentation:** 3
**Contribution:** 3
**Rating:** 8
**Confidence:** 3

**Summary:**

This paper investigates the generation–understanding internal gap in unified multimodal large language models (MLLMs), where the understanding branch consistently outperforms the generation branch.
To address this, the authors propose a simple internal gap–based self-improvement framework that leverages the model’s own understanding capability to guide and enhance generation, without any external rewards or supervision.Key contributions include:

1. Quantitative diagnosis of generation–understanding non-unification, introducing an internal Non-Unification Score to measure intra-model consistency and empirically verifying the widespread internal gap across six MLLMs (Sec. 3).

2. Internal gap–based self-improvement framework, leveraging the stronger understanding branch to score and filter generations for post-training (SFT/DPO), effectively improving generation quality and reducing non-unification without external signals (Sec. 4.1–4.2).

3. Theoretical analysis of the co-improvement effect, extending learning dynamics to multimodal models and revealing that shared empirical neural tangent kernels (eNTKs) drive aligned updates between generation and understanding (Sec. 5).

4. Curriculum-based self-improvement strategy, progressively reintroducing previously underutilized or difficult samples through curriculum replay to further enhance both branches and surpass external-reward baselines (Sec. 6).

**Strengths:**

1. The paper propose an internal gap–based self-improvement framework, which is conceptually simple but novel, requiring no external supervision or reward models.  The authors also introduce a new internal evaluation metric (Non-Unification Score) to quantify intra-model consistency.

2. The work provides strong empirical evidence through large-scale experiments on six unified MLLMs and three task difficulty levels (Figure 2).

3. Figures and algorithms are clearly presented — e.g., Algorithm 1 succinctly formalizes the self-improvement process, and Figure 5 (a) intuitively illustrates the co-improvement effect with side-by-side visual examples.

4. The discovery of a shared empirical NTK between generation and understanding offers a novel theoretical perspective on multimodal model coupling ( Eq. 2–3). The curriculum-based self-improvement strategy demonstrates a scalable path to strengthen MLLMs without external data or rewards, surpassing reward-model–based baselines such as T2I-R1 and HermesFlow.

**Weaknesses:**

1. Theory–practice gap: While the shared eNTK explanation is conceptually interesting, its empirical validation remains limited, and the theoretical section is notation-heavy, reducing accessibility for non-theoretical readers.

2. Limited model diversity: Main experiments focus on Janus-Pro and Show-o. Although six models were initially analyzed, most in-depth post-training results come from only two, limiting the generality of conclusions.

**Questions:**

1. How stable is the self-improvement process when the understanding branch provides noisy or incorrect judgments?

2. Can the proposed framework generalize to other modalities (e.g., video or audio) or to non-generative MLLM tasks?

3. How sensitive are the results to the number of generated candidates N and the selection threshold in the understanding branch?

---

> ### Author Response · Authors · 2025-11-21
>
> Thanks for your time in reviewing our paper and for recognizing the novelty and empirical contributions of our work. We now address the raised questions as follows.
>
> ---
> >Q1:Theory–practice gap: While the shared eNTK explanation is conceptually interesting, its empirical validation remains limited, and the theoretical section is notation-heavy, reducing accessibility for non-theoretical readers.
>
> Thanks for your comment.
>
> 1. For the **empirical validation**, we would like to use SFT-based self-improvement as an example to further clarify how the empirical results in `Section 5.2`, together with `Proposition 1`, jointly explain the co-improvement phenomenon. Specifically, the mainly observed co-improvement is:
>
> * For a given prompt $\mathbf{y}\_0$, the pre–self-improved unified MLLM generates an image $\mathbf{x}$ that is not aligned with $\mathbf{y}\_0$. However, for the understanding task, the model predicts that $\mathbf{x}$ **does satisfy** $\mathbf{y}\_0$. It means, **for pre–self-improved model, both generation and understanding are incorrect.**
> * After self-improvement, for the same prompt $\mathbf{y}_0$, the model becomes **less likely to generate an incorrect image** $\mathbf{x}$, and it also **correctly** judges that such an $\mathbf{x}$ does **not** satisfy $\mathbf{y}_0$, i.e.,  false positive correction in `Section 5.2`.
>
>
> The empirical evidence in `Figure 7` shows that the above phenomenon arises because:
> - Samples in improved group (i.e., the false positive correction group) have more similar training samples (see `Figure 7(a)`).
> - This leads the generation and understanding branches to share a large-norm eNTK term $\\|{\mathcal{K}^{,t}\_{k,r}(\mathcal{Y}\_0,\mathcal{Y}\_u)}\\|_{F}$ (as proven in `Proposition 1`).
> - Shared eNTK drives the learning dynamics of generation ($\Delta G$) and understanding ($\Delta U$) to follow similar trends (see `Figure 7(b)` and `Proposition 1`), i.e., $\Delta U$ and $\Delta G$ tend to have the same sign.
> - Consequently, when the generation branch is optimized and the misgeneration probability decreases, i.e., $\Delta G < 0$ (see `Figure 7(c)`), it also induces $\Delta U < 0$, reducing the misjudgment probability $\pi_{\theta}(\mathbf{y}_0 \mid \mathbf{x}_0)$ and thereby improving the model’s understanding performance.
>
> 2. For the **notation**, sorry for the confusion. We have added a table summarizing the key notations used in the theoretical analysis for easier reading. Please refer to `Table 10` in the revised manuscript.
>
> >Q2: Limited model diversity: Main experiments focus on Janus-Pro and Show-o. Although six models were initially analyzed, most in-depth post-training results come from only two, limiting the generality of conclusions.
>
>
> Thank you for the suggestion. We additionally report the self-improvement results for a 1B-sized model, `Janus-Pro-1B`, where the training data, training pipeline, hyperparameters, and evaluation metrics follow `Section 4.2.1`. **The table below** presents the performance improvements of the self-improved `Janus-Pro-1B` on `T2I-CompBench++`. We observe **consistent gains in generation, understanding, and unification after self-improvement**, supporting the feasibility of leveraging the internal gap for model enhancement. Results of `T2I-CompBench++` are added in `Appendix F.1`.
>
>   - Generation ($\uparrow$)
>
>  | Model | Texture | Shape |Color|Spatial | Non-Spatial|Complex |Overall|
> | -------- | -------- | -------- |-------- |-------- |-------- |-------- |-------- |
> |  Janus-Pro-1B   | 50.85     |   43.11  | 55.81    |  7.99  |21.34   |23.05    |33.70   |
> |  +*SFT*    | 55.58     |   43.67 | 59.34  |  10.40  | 28.84    |32.72     |38.43    |
>
>   - Understanding ($\uparrow$)
>
>  | Model | Texture | Shape |Color|Spatial | Non-Spatial|Complex |Overall|
> | -------- | -------- | -------- |-------- |-------- |-------- |-------- |-------- |
> |  Janus-Pro-1B   | 50.00     |   50.00    | 50.00     | 50.00    |50.00   |50.00    |50.00    |
> |  +*SFT*    | 53.76    |   32.38 | 58.70 |  53.87  | 65.28   |51.58   |52.60     |
>
>    - Non-unification ($\downarrow$)
>
>  | Model | Texture | Shape |Color|Spatial | Non-Spatial|Complex |Overall|
> | -------- | -------- | -------- |-------- |-------- |-------- |-------- |-------- |
> |  Janus-Pro-1B   | 51.33    |  46.00    | 29.83   | 42.67   |9.00   |16.00   |32.47   |
> |  +*SFT*    | 41.00  |   39.00 | 24.07|  35.33 | 9.33  |15.00 |27.29   |

---

> > ### Author Response · Authors · 2025-11-21
> >
> > >Q3: How stable is the self-improvement process when the understanding branch provides noisy or incorrect judgments?
> >
> > Thank you for this insightful question. In our framework, there are **two mechanisms** that **help mitigate the risk** of the understanding branch providing noisy or incorrect judgments:
> > - **The understanding branch is stronger for image understanding.** Our analysis in `Section 3` shows that the understanding branch can accurately judge whether the generation output satisfies the prompt with weak generation scores exceeding 50%, which provides the foundation for effective self-improvement.
> > - **Prioritize high-confidence samples during self-improvement**. In the self-improvement, we select **samples** for which the model assigns the **highest confidence** (see `Section 4.1`), which helps mitigate the potential misjudgments.
> >
> > Therefore, when applying the self-improvement framework, it is practical to perform a **small-scale pre-check** of the unified MLLM’s understanding capability, for example, by calculating the weak generation score on a small subset of samples. A low weak generation score indicates a higher risk that the understanding branch may provide noisy or incorrect judgments, in which case self-improvement should be applied cautiously. In addition, **selecting high-confidence samples** during training can further mitigate the impact of potential noise.
> >
> > >Q4: Can the proposed framework generalize to other modalities (e.g., video or audio) or to non-generative MLLM tasks?
> >
> > Thank you for the question. The proposed self-improvement framework **can generalize to other modalities** because **its effectiveness only requires the presence of an internal gap between different tasks**, e.g., the understanding branch being stronger than the generation branch. **In other modalities**, such as video generation and understanding, if the understanding branch can reliably assess generated videos, for example, by evaluating temporal consistency or event coherence, it can similarly provide internal signals to improve video generation performance.
> >
> > > Q5: How sensitive are the results to the number of generated candidates N and the selection threshold in the understanding branch?
> >
> > Thank you for the question.
> >
> > - For **generated candidates $N$**, the table below reports the number of self-improved samples obtained by `Janus-Pro-7B` under different values of $N$.  We observe that when $N$ is small (e.g., $N=2,3,4$), the number of selected samples is limited, and increasing $N$ leads to a clear gain in training size. As $N$ (e.g., $N=7,8,9$) is large, the marginal increase  gradually diminishes, indicating that the training data size become less sensitive to $N$. In our experiments, we set ($N = 10$) to retain as many training samples as possible. More experimental results on the generated candidates have been added to `Appendix F.3`.
> >
> >
> > | N=2 | N=3|N=4|N=5 |N=6 |N=7 |N=8 |N=9 |N=10 |
> > | -------- | -------- |-------- |-------- |-------- |-------- |-------- |-------- |-------- |
> >  |   254  |    918  |1338  |   1615 | 1823  | 1964 | 2088  | 2185 | 2265 |
> >
> >
> > - For the **selection threshold**, we have **no explicit threshold** in self-improvement. We store the model’s predicted probabilities for score $1$ and score $0$. Samples with the **highest predicted probability of score $1$** are selected as chosen samples for SFT and DPO, while **samples with the highest predicted probability of score $0$** are selected as rejected samples for DPO (see `Line 202–212` for details).

---

> ### Author Response · Authors · 2025-11-27
>
> Dear Reviewer Qp5p,
>
> We would like to express our sincere gratitude once again for your time and insightful comments on our work. With the discussion period nearing its end, we kindly ask whether our responses have satisfactorily addressed your concerns. If you have any further questions or comments, please feel free to share them. We look forward to further communication with you.
>
> Best,
>
> Authors

---

### Official Review · Reviewer_fxim · 2025-10-31

**Soundness:** 3
**Presentation:** 3
**Contribution:** 3
**Rating:** 6
**Confidence:** 4

**Summary:**

This paper systematically investigates the prevalent "generation-understanding non-unification" phenomenon in Multimodal Large Language Models (MLLMs), where understanding capabilities typically outperform generation capabilities. The authors propose an internal gap-based self-improvement framework that leverages the model's own stronger understanding branch to guide its weaker generation branch. Without relying on external signals, this approach significantly enhances generation quality and reduces the internal gap through post-training methods like SFT or DPO. Experiments reveal that this method also induces a "co-improvement" effect, where improvements in generation simultaneously enhance understanding, particularly in identifying misaligned generated samples. Furthermore, drawing from learning dynamics theory, the paper identifies the shared empirical Neural Tangent Kernel (eNTK) between generation and understanding as the key mechanism behind co-improvement. Based on this, a curriculum learning strategy is proposed to dynamically expand the training data, further boosting model performance and unification.

**Strengths:**

- Rigorous Problem Verification: The paper first confirms the internal gap in MLLMs where "generation is weaker than understanding" through large-scale evaluation (across 6 models and tasks of varying difficulty). To achieve this, the authors innovatively propose a "non-unification score" that does not rely on external evaluators.
- Simple and Effective Solution: An "internal gap-based self-improvement" framework is proposed, which leverages the model's own stronger understanding capability to filter generated data (for SFT or DPO). This method requires no external reward models, achieving closed-loop self-improvement.
- Discovery and Explanation of the "Co-improvement" Effect: this paper introduces that "generation-targeted training" also enhances "understanding capability" (particularly in correcting false positive samples). From a learning dynamics perspective, the paper attributes this to the shared eNTK (empirical Neural Tangent Kernel) between the generation and understanding branches.
- Clear Logical Structure Throughout the Paper: The paper exhibits a clear and coherent logical structure from beginning to end.

**Weaknesses:**

- Experimental Results Heavily Rely on a Single, Insufficiently Strong Judge Model:Two of the paper's key conclusions—that the "internal gap stems mainly from weak generation" and the "co-improvement effect"—rely heavily on using Qwen2.5-VL-72B-Instruct as the sole external judge.
    However, Qwen2.5-VL-72B is not a strong enough multimodal model to serve as a reliable evaluator, especially when dealing with complex or "hard tasks." To enhance the credibility of the experimental results, it is recommended to adopt more advanced SOTA models or incorporate multiple models as evaluators.
- Limited Innovation and Effectiveness of the Curriculum Learning Part:
    The curriculum learning (C-SFT/C-DPO) strategy proposed in Section 6, whose core idea is to reuse "difficult" samples discarded earlier due to the model's limited capability, is lacking innovation and Effectiveness.
    This method is a conventional application of curriculum learning and lacks significant methodological innovation.Based on the experimental results (Table 1 and Table 5), the improvement of C-SFT over standard SFT is very marginal. For example, on the "Overall" metric for T2I-CompBench++ (Table 5), the generation score for Janus-Pro only increased from 43.29 to 44.18, and for Show-o, only from 52.67 to 52.82. This marginal gain seems insufficient to justify the introduction of this strategy.
- Insufficient Experimental Evidence and Lack of Generalization for "Co-Improvement Effect":The "co-improvement effect" (Finding 2), a key contribution, claims that self-improvement targeted at generation also enhances understanding. As mentioned, the effect is primarily shown via the custom, Qwen-dependent "win rate" metric.The improvement on standard benchmarks is Negligible(Table 8). For example, after SFT, Janus-Pro-7B's score on POPE \textit{decreased} from 89.04 to 88.45, the improvement on MMB was only 0.74 (76.23 $\rightarrow$ 76.97), and on GQA, only 0.1 (56.02 $\rightarrow$ 56.12).This significant disconnect between the custom metric (Win Rate) and standard benchmarks (Table 8) suggests that the so-called "co-improvement" might just be overfitting to the internal understanding task or the Qwen judge's preferences, rather than a genuine, generalizable improvement in understanding ability for standard VQA or hallucination detection tasks.

**Questions:**

A critical ambiguity exists regarding potential data leakage in the T2I-CompBench++ experiments. The authors state in Section 4.2.1 that post-training data was constructed using ``about 6000 text prompts as post-training candidates'' from T2I-CompBench++. Subsequently, the model's performance is evaluated on the T2I-CompBench++ evaluation set (as shown in Table 1 and Table 5) . However, the paper fails to explicitly state whether the 6000 prompts used for post-training (SFT/DPO) were drawn from the benchmark's designatedtraining split. If any overlap exists between these post-training prompts and the prompts in the evaluation set, the reported performance gains on this benchmark would be invalid due to data contamination. This lack of clarity undermines the reliability of these key experimental results.

---

> ### Author Response · Authors · 2025-11-21
>
> We thank the reviewer for reviewing this paper and for recognizing our work as rigorous and effective. We now address the questions raised as follows.
>
> ----
>
> >Q1: Experimental Results Heavily Rely on a Single, Insufficiently Strong Judge Model... To enhance the credibility of the experimental results, it is recommended to adopt more advanced SOTA models or incorporate multiple models as evaluators.
>
> Thank you for the suggestion. We additionally include `Gemini-2.5-Pro` as an external evaluator. On the public VLM leaderboard [OpenCompass](https://opencompass-open-vlm-leaderboard.hf.space), `Gemini-2.5-Pro` ranks **4th among more than 200 models** in image understanding. We present the additional experimental results below.
>
>
> - **Internal gap stems mainly from weak generation**
>
> Using `Janus-Pro-7B` as an example, we recompute the weak generation score defined in `Section 3` and observe that in most subtasks (8/9) the weak generation score is above 50%, even reaches 100% (2/9). This indicates the internal gap is primarily caused by weak generation instead of misunderstanding which **is consistent with** the results obtained using `Qwen`.
>
>
> | External Judger | Single Obj.|Two Obj.|Color Attri.| Texture |Spatial |Complex |Physic |Chemistry|Biology |
> | -------- | -------- | -------- |-------- |-------- |-------- |-------- |-------- |-------- |-------- |
> | Gemini-Pro-2.5   | 88.89     |  77.78   |50.00   |89.23    | 92.47    |  74.19  |100.00 |100.00  |66.67  |
>
> The weak generation scores for **all six unified MLLMs across the nine subtasks** have also been added to the revised manuscript (see `Fig.9`). All models evaluated with `Gemini-Pro-2.5` consistently **support the conclusion** that the internal gap is primarily driven by weak generation.
>
> - **Co-improvement effect**
>
> Similarly, we use `Gemini-2.5-Pro` to compute the understanding metric, Win Rate in `Section 4.2.1`. The table below reports the Win Rate of `Janus-Pro-7B` after SFT-based self-improvement. We observe in most subtasks (5/6) the Win Rate exceeds 50%, **indicating** the model's understanding capability improves after self-improvement, providing more evidence of the co-improvement effect. The revised manuscript includes additional results supporting this phenomenon (see `Line 1223–1235` and `Fig.13`).
>
> | Model | Texture | Shape |Spatial |Color |Complex |Non-Spatial |
> | -------- | -------- | -------- |-------- |-------- |-------- |-------- |
> |  Gemini-Pro-2.5    | 60.87     |    52.31  | 40.00     |  87.18   | 65.00     |75.00     |
>
> >Q2:Limited Innovation and Effectiveness of the Curriculum Learning Part ... the improvement of C-SFT over standard SFT is very marginal. For example, on the "Overall" metric for T2I-CompBench++ (Table 5), the generation score for Janus-Pro only increased from 43.29 to 44.18, and for Show-o, only from 52.67 to 52.82. This marginal gain seems insufficient to justify the introduction of this strategy.
>
> Thank you for your comment. We would like to point out that **curriculum learning in unified MLLMs differs fundamentally from classical curriculum-learning settings**, and **its feasibility is not a priori guaranteed**. Therefore, the **key innovation of curriculum-learning part** is to demonstrate that curriculum learning is effective in unified MLLMs. Specially,
> - In *classical setup*, difficult samples are typically determined by an external stronger model or predefined rules and then gradually introduced during training [1,2,3].
> - However, *in unified MLLMs*, curriculum learning relies on the model’s own generation and understanding branches to dynamically re-evaluate difficult samples discarded earlier. **This introduces a new risk**: if the understanding  degrades while the generation improves, the understanding branch may misjudge sample and incorrectly add low-quality samples back into training, potentially harming overall performance.
>
> Therefore, the feasibility of curriculum learning in unified MLLMs **is not obvious** which **hinges on the premise** that **optimizing the generation branch within a structurally coupled unified model does not degrade understanding**, i.e., that the co-improvement effect holds (as discussed in `Sections 4` and `5`). The results (`Table 1`) further confirm the validity of incorporating curriculum learning in our setting.
>
> Additionally, designing more effective curriculum strategies to achieve larger improvements is indeed an interesting direction, which we leave for future work.

---

> ### Author Response · Authors · 2025-11-21
>
> > Q3: Insufficient Experimental Evidence and Lack of Generalization for "Co-Improvement Effect":...suggests that the so-called "co-improvement" might just be overfitting to the internal understanding task or the Qwen judge's preferences, rather than a genuine, generalizable improvement in understanding ability for standard VQA or hallucination detection tasks.
>
> Thank you for the question.
>
> - First,  during self-improvement, **models do not see `Qwen’s` judgment labels, nor is the understanding branch optimized.** Therefore, **the improvement in understanding cannot be due to overfitting to Qwen judge’s preferences or to the internal understanding task.**   Specifically, Win Rate is computed on a fixed set of images ($\mathbf{x}_0$) generated before self-improvement: we first obtain 0/1 judgments from the pre-trained understanding branch ($\mathbf{y}_0$), the self-improved understanding branch ($\mathbf{y}_1$), and an external evaluator such as `Qwen` ($\mathbf{y}$).  Then, treating `Qwen’s` output ($\mathbf{y}$) as an oracle label, Win Rate simply measures whether $\mathbf{y}_1$ or $\mathbf{y}_0$ is closer to $y$ (see `Line 238–248`).  This process introduces no supervision from `Qwen` and involves no loss applied to the understanding branch.
>
>      Moreover, **recomputing Win Rate** using additional evaluators such as `Gemini-2.5-pro` **still shows consistent improvements** (see **[Q1]**), indicating that **the gain is not tied to any specific judge’s preference**.
>
> - Second, **it would be inappropriate to directly compare Win Rate’s magnitude or variability with Accuracy** (i.e., standard understanding metrics). Win Rate follows the win rate without tie metric used in LLM-as-judge works [4,5]. **It is more sensitive** because it is computed only on samples where pre- and post-self-improved predictions differ. The smaller denominator than the full data makes subtle but genuine improvements in understanding easier to detect.
> - Finally, we point out that on standard understanding benchmarks, **the self-improved (e.g., SFT) model improves on four out of five benchmarks, with a particularly notable gain on MMMU of about 3%** (32.86 → 35.24). This consistent, cross-benchmark improvement further indicates that the enhancement in understanding is not accidental.
>
> > Q4: A critical ambiguity exists regarding potential data leakage in the T2I-CompBench++ experiments.
>
> Sorry for missing details on the data split for `T2I-CompBench++`, but we can confirm that **there is no data leakage**.
>
> - First, we strictly followed **the official data split** defined in the `T2I-CompBench++` paper [6] to obtain the training and evaluation sets. Specifically, we downloaded the training and validation data directly from the [official repository](https://github.com/Karine-Huang/T2I-CompBench/tree/main/examples/dataset) for our experiments.
> - Second, we further verified data integrity by **performing string-level matching** between used training data and validation data. The matched ratio was $0$, confirming no contamination exists.
>
> We have added this details to the revised manuscript (see `Line 1052-1054`).
>
> [1] Curry-DPO: Enhancing Alignment using Curriculum Learning & Ranked Preferences
>
> [2] 2D-Curri-DPO: Two-Dimensional Curriculum Learning for Direct Preference Optimization
>
> [3] Curriculum Direct Preference Optimization for Diffusion and Consistency Models
>
> [4] Judging llm-as-a-judge with mt-bench and chatbot arena.
>
> [5] Mllm-as-a-judge: Assessing multimodal llm-as-a-judge with vision-language benchmark.
>
> [6] T2I-CompBench++: An Enhanced and Comprehensive Benchmark for Compositional Text-to-image Generation

---

> ### Author Response · Authors · 2025-11-27
>
> Dear Reviewer fxim,
>
> We sincerely thank you once again for your time and constructive comments on our work. With the discussion period nearing its end, we would like to kindly check whether our responses have resolved your concerns.  Please feel free to share any remaining questions or comments. We look forward to continuing our discussion with you.
>
> Best,
>
> Authors

---

### Official Review · Reviewer_iJU7 · 2025-10-31

**Soundness:** 3
**Presentation:** 4
**Contribution:** 3
**Rating:** 8
**Confidence:** 3

**Summary:**

The authors let the understanding branch of MLLMs judge the correctness of images generated by the generation branch, as a reflection of the model's internal gap, across multiple models and tasks. They leverage this gap to improve the models' generation ability using SFT and DPO with data generated by the generation branch and verified by the understanding branch, resulting in improved generation, understanding, and internal unification. They explain the enhancement of the understanding ability with a learning dynamics theory and support it with empirical analysis. They further propose to incorporate data that could not be utilized initially in their data-collection pipeline into the post-training process, after the model's ability has improved, thereby further enhancing the training outcome.

**Strengths:**

1. The self-improvement approach, which uses the understanding ability to guide the generation branch, is intuitive and intriguing in concept and effective in practice.
2. The work is generally solid and logically rigorous, with a motivation validated across multiple models, comprehensive experiments conducted on two models evaluated by both the authors’ proposed metrics and existing benchmarks, theoretical interpretation, and empirical validation. One of the relatively unconvincing and inelegant aspects—using Qwen as an external judge—is compensated for by a human verification experiment.
3. The learning dynamics analysis provides theoretical insights into the mechanism of the co-improvement effect.
4. The paper is generally well written, and most details are explained clearly.

**Weaknesses:**

The empirical evidence for some of the paper's claims is less conclusive than stated, which lacks further clarification:

1. In Fig.2(c), the claimed "trend of increasing with task difficulty" for the non-unification score is not obvious or monotonic. The variation between models seems to dominate any difficulty-based trend.
2. In Fig.7, the difference in similarity between improved samples and random samples is also not clear, especially for image pairs.

**Questions:**

1. What was the detailed setup of the human evaluation in Fig. 9? The appendix mentions the human check but omits details on the number of annotators, the interface used, the specific instructions given.
2. Could the authors evaluate the self-improved Show-o model on the understanding benchmarks, similar to Janus-Pro-7B in Tab. 8?

---

> ### Author Response · Authors · 2025-11-21
>
> Thanks for your time reviewing this paper and for recognizing our work as solid and rigorous. We now address your questions as follows.
>
> ---
> > Q1: In Fig.2c, the claimed "trend of increasing with task difficulty" for the non-unification score is not obvious or monotonic. The variation between models seems to dominate any difficulty-based trend.
>
> Sorry for the confusion. What we intended to convey in `Fig.2c` is that all six evaluated MLLMs exhibit largest non-unification scores on the hard (5/6 MLLMs) and medium (1/6 MLLMs) tasks. As a result, if the evaluation considers only easy tasks, the non-unification scores may appear lower, which is not because the internal gap is (nearly) absent, but simply because the tasks are too easy to reveal it. Therefore, including tasks of different difficulty levels offers a more complete assessment of an MLLM’s non-unification. We have revised the manuscript to clarify this point (see `Line 180–183`).
>
> > Q2: In Fig.7, the difference in similarity between improved samples and random samples is also not clear, especially for image pairs.
>
> Thank you for the question. In `Fig.7`, we observe that the similarity difference between the improved and random groups is  small for image pairs, but there is **a clear and substantial difference in prompt similarity** (i.e., compare the bar $y_0$ and $y$): improved group has a mean similarity of $0.85$, while the random group is only $0.65$. For the improved group, the **advantage in prompt similarity matters more**, since prompt similarity contributes more to the eNTK term in `Equation 2` and `3` than image similarity.
>
> Specially, `Equation 2` and `3` shows that the learning dynamics for both understanding and generation accumulate token by token. Consequently, the prompt tokens which appear **at the beginning of the input concatenated sequence** $\mathcal{Y}\_0 = [\mathbf{y}\_0 \mid \mathbf{x}\_0]$ (see `Line 1507-1510`, `Appendix E`), make a **substantially contribution to the resulting eNTK** than the later image tokens, especially for the outputs of the early tokens. Therefore, compared with the random group, the improved group, **which has much higher prompt similarity while exhibiting similar image similarity**, is more likely to have the learning dynamics of both generation and understanding branches dominated by shared eNTK (i.e., $\mathcal{K}^t_{k,r}(\mathcal{Y}_0,\mathcal{Y}_u)$)(see `Section 5.2` for details), which in turn encourages the observed co-improvement.
>
> We have clarified this point in the revised manuscript (See `Line 1325-1335`).
>
> > Q3: What was the detailed setup of the human evaluation in Fig. 9? The appendix mentions the human check but omits details on the number of annotators, the interface used, the specific instructions given.
>
> Thank you for your question. Using the human check for `Janus-Pro-7B` on the `Texture` subtask as an example, the procedure is as follows:
>
> - **Step 1**: Obtain the understanding branch scores.
>
>     Following the definition of the non-unification score in `Section 3`, we first compute the understanding branch’s judgment of whether each generated image satisfies its prompt.
>
> - **Step 2**: Select samples predicted as incorrect ($score = 0$).
>
>     We collect all samples for which the understanding branch outputs $score = 0$ (misaligned image–prompt pairs). In the `Texture` subtask, this yields $130$ samples.
>
> - **Step 3**: Perform human re-evaluation of these $score = 0$ samples.
>
>     All $130$ samples are manually checked, where annotators decide whether each image satisfies the corresponding prompt ($score = 0$ or $1$). The evaluation is conducted by two PhD-level annotators: one performs the annotation, the other performs double-check, ensuring accurate understanding of both prompts and images.
>
> - **Step 4**: Compute the human-evaluated weak generation score.
>
>    As defined in `Line 191`, this score measures the probability that humans agree with the understanding branch conditional on the understanding branch predicting $score = 0$. For the `Texture` subtask: human-evaluated weak generation score is
>
>    $\mathbb{P}\big( \pi^{\text{und}}\_{\theta}(\mathbf{x},\ q(\mathbf{y})) = S\_{\text{human}}(\mathbf{x},\ q(\mathbf{y})) \,\big|\, \pi^{\text{und}}\_{\theta}(\mathbf{x},\ q(\mathbf{y})) = 0 \big) = \frac{112}{130} = 0.8615$
>
>     where $S\_{\text{human}}$ denotes the human score.
>
> We have added the human check procedure in the revised manuscript (see `Line 1017-1034`) and included the annotation interface in `Fig.10`.

---

> > ### Author Response · Authors · 2025-11-21
> >
> > > Q4: Could the authors evaluate the self-improved Show-o model on the understanding benchmarks, similar to Janus-Pro-7B in Tab. 8?
> >
> > Thank you for the suggestion. We provide the performance of `Show-o` and `self-improved Show-o` on the understanding benchmarks as follows. We observe that the `self-improved Show-o` (e.g.,*SFT* and *C-SFT*) consistently outperforms the original `Show-o` across all five benchmarks. We have incorporated this experimental result into the revised manuscript (see updated `Table 8` and `Line 1199–1201`).
> >
> > | Model | POPE | MMBench |MMMU |GQA |SEED|
> > | -------- | -------- | -------- |-------- |-------- |-------- |
> > | Show-o     |  64.05    |    30.91  |  23.33  |  56.82     |  52.86   |
> > | +*SFT*     |  65.27    |  31.92    |  **24.00**   |  57.22    |  54.14  |
> > | +*DPO*    |    64.71  |  30.82  |    23.33 |  57.03  |   52.73 |
> > | +*C-SFT*     |  **65.82**    |   **32.34**  |   23.33  |  **57.33**    |   **54.32** |
> > | +*C-DPO*     |   64.97   |    31.14 |  23.33   |   57.09  |    52.90 |

---

> ### Author Response · Authors · 2025-11-27
>
> Dear Reviewer iJU7,
>
> We sincerely appreciate your time and constructive comments on our manuscript again. As the discussion period draws to a close, we would like to verify whether our responses have adequately addressed your concerns. Please feel free to share any remaining questions or comments. We would be glad to discuss them further.
>
> Best,
>
> Authors

---

### Meta-Review · Area_Chair_YQiL · 2026-01-07

**Summary:**

Two major concerns were raised:
- The reproducibility and scalability of the co-improvement effect. The co-improvement effect, specifically the improvement on the understanding branch, is demonstrated on multiple metrics, including a proposed win-rate metric. However, the performance gain is only noticeable on the proposed win-rate metric and is less noticeable on the other public benchmarks.
- Experiments. The judgement of self-improvement and co-improvement is done mostly with the help of Qwen models. A key question is whether Qwen is reliable enough.

**Reviewer Concerns:**

A human-in-the-loop check procedure is provided to enhance the strength of the experimental results, which partially address the problem in the experiments. However, the authors have not provided enough evidences to verify the co-improvement effect.

**Reviewer Scores:**

No change of score is expected.

---

### Decision · Program_Chairs · 2026-01-26

Accept (Poster)